# Modelling Silicate – Nitrate - Ammonium co-limitation of algal growth and the importance of bacterial remineralisation based on an experimental Arctic coastal spring bloom culture study

Tobias R. Vonnahme[1], Martial Leroy[2], Silke Thoms[3], Dick van Oevelen[4], H. Rodger Harvey[5], Svein Kristiansen[1], Rolf Gradinger[1], Ulrike Dietrich[1], Christoph Voelker[3]

[1] Department of Arctic and Marine Biology, UiT – The Arctic University of Norway, Tromsø, Norway
[2] Université Grenoble Alpes, Grenoble, France
[3] Alfred-Wegener Institute for Polar and Marine Research, Bremerhaven, Germany
[4] Department of Estuarine and Delta Systems, NIOZ Royal Netherlands Institute for Sea Research, and Utrecht University, Texel, Yerseke, Netherlands
[5] Department of Ocean and Earth Sciences, Old Dominion University, Norfolk, USA

*Correspondence to*: Tobias R. Vonnahme (T.r.vonnahme@gmail.com) and Christoph Voelker (christoph.voelker@awi.de)

**Abstract.** Arctic coastal ecosystems are rapidly changing due to climate warming. This makes modelling their productivity crucially important to better understand future changes. System primary production in these systems is highest during the pronounced spring bloom, typically dominated by diatoms. Eventually the spring blooms terminate due to silicon or nitrogen limitation. Bacteria can play an important role for extending bloom duration and total $CO_2$ fixation through ammonium regeneration. Current ecosystem models often simplify the effects of nutrient co-limitations on algal physiology and cellular ratios and simplify nutrient regeneration. These simplifications may lead to underestimations of primary production. Detailed biochemistry- and cell-based models can represent these dynamics but are difficult to tune in the environment. We performed a cultivation experiment that showed typical spring bloom dynamics, such as extended algal growth via bacterial ammonium remineralisation, reduced algal growth and inhibited chlorophyll synthesis under silicate limitation, and gradually reduced nitrogen assimilation and chlorophyll synthesis under nitrogen limitation. We developed a simplified dynamic model to represent these processes. Overall, model complexity in terms of the number of parameters is comparable to the phytoplankton growth and nutrient biogeochemistry formulations in common ecosystem models used in the Arctic while improving the representation of nutrient co-limitation related processes. Such model enhancements that now incorporate increased nutrient inputs and higher mineralization rates in a warmer climate will improve future predictions in this vulnerable system.

# 1 Introduction

Marine phytoplankton are responsible for half of the $CO_2$ fixation on Earth (Field et al., 1998; Westberry et al., 2008). In high latitude oceans, diatoms are an important group contributing 20-40% of the global $CO_2$ fixation (Nelson et al., 1995; Uitz et al., 2010). Marine primary production can be bottom-up limited by light and/or nutrients like nitrogen (N), phosphorous (P), silicon (Si), and iron (Fe). Their availability is affected by pronounced geographical and seasonal variations (Eilertsen et al., 1989; Loebl et al., 2009;

Iversen and Seuthe, 2011; Moore et al., 2013). Arctic coasts are one of the fastest changing systems due to climate change. Thus, modelling their dynamics is difficult but crucial for predictions of primary production with climate change (e.g., Slagstad et al., 2015; Fritz et al., 2017; Lannuzel et al., 2020). In Arctic coastal ecosystems, primary production is typically highest in spring. In spring, previous winter mixing supplied fresh nutrients and a stratified surface layer with sufficient light is facilitated by

increasing temperatures and potentially sea ice melt (Sverdrup, 1953; Eilertsen et al., 1989; Eilertsen and Frantzen, 2007; Iversen and Seuthe, 2011). With increasing temperatures and runoff, stratification in coastal Arctic systems is expected to increase (Tremblay and Gagnon, 2009). This will lead to decreased mixing and nutrient upwelling in autumn and winter and an earlier stratified surface layer in spring, which may lead to an earlier spring bloom (Tremblay and Gagnon, 2009). However, at the same time,

brownification and increased sediment resuspension is already leading to light inhibition in spring, which may lead to a delayed spring bloom (Opdal et al., 2019). The spring bloom typically consists of chain-forming diatoms and is terminated by Si or N limitation (Eilertsen et al., 1989; Iversen and Seuthe, 2011). Zooplankton grazing is typically of low importance for terminating blooms (e.g., Saiz et al., 2013), while inorganic nutrients are considered to drive bloom termination (Krause et al. 2019, Mills et al. 2018).

Heterotrophic bacteria remineralisation of organic matter may supply additional N and Si (Legendre and Rassoulzadegan, 1995; Bidle and Azam, 1999; Johnson et al., 2007). N regeneration has been described as a mostly bacteria-related process (Legendre and Rassoulzadegan, 1995), while Si dissolution is mainly controlled by abiotic dissolution of silica (Bidle and Azam, 1999). Zooplankton may also release some ammonium and urea after feeding on phytoplankton, but we suggest that this process is likely far less

important than bacterial regeneration (e.g. Saiz et al., 2013). Previously measured ammonium excretion of Arctic mesozooplankton is typically low compared to bacterial remineralization (Conover and Gustavson, 1999), with the exception for one study in summer in a more open ocean setting (Alcaraz et al., 2010). In some Arctic systems urea, excreted by zooplankton may be an important N source for regenerated algae production (Conover and Gustavson, 1999). A warmer climate will increase both

bacteria-related remineralisation rates (Legendre and Rassoulzadegan, 1995; Lannuzel et al., 2020) and abiotic silica dissolution (Bidle and Azam, 1999). However, the magnitude is not well understood. Phytoplankton blooms may be dominated by a single or a few algal species, often with a similar physiology during certain phases of the bloom (e.g., Eilertsen et al., 1989; Degerlund and Eilertsen, 2010; Iversen and Seuthe, 2011). Chain-forming centric diatoms share physiological needs and responses to

nutrient limitations (e.g., Eilertsen et al., 1989; von Quillfeldt, 2005) and typically dominate these blooms. In some Arctic and sub-Arctic areas the Arctic phytoplankton species chosen for this model, *Chaetoceros socialis*, can be dominant during spring blooms (Rey and Skjoldal, 1987; Eilertsen et al., 1989; Booth et al., 2002; Ratkova and Wassmann, 2002; von Quillfeldt, 2005; Degerlund and Eilertsen, 2010). Such spring phytoplankton blooms are accompanied by heterotrophic bacterioplankton blooms also showing

typical succession patterns and distinct re-occurring taxa that dominate the community (Teeling et al., 2012; Teeling et al., 2016). The importance of bacterial nutrient recycling for regenerated production has been recognized in several ecosystem models (e.g. van der Meersche et al., 2004; Vichi et al., 2007; Weitz et al., 2015) and algae bioreactor models focusing on nutrient conversions (e.g. Zambrano et al., 2016). However, these models are typically highly simplified or omitted in more sophisticated dynamic multi-

nutrient, quota based models (e.g. Flynn and Fasham, 1997b.; Wassmann et al., 2006; Ross and Geider, 2009). These latter models have been often developed and tuned based on cultivation experiments in which microbial remineralization reactions were assumed to be absent (e.g. Geider et al., 1998; Flynn, 2001) despite the fact that most algae cultures, likely including Geider et al., (1998) and Flynn (2001) are not axenic. Parameters estimated by fitting axenic models on non-axenic experiments may be misleading,

mostly by an inflated efficiency of DIN uptake. Additional positive effects of bacteria include vitamin synthesis (Amin et al., 2012), trace metal chelation (Amin et al., 2012), the scavenging of oxidative stressors (Hünken et al., 2008), and exchange of growth factors (Amin et al., 2015). Especially in the stationary algal growth phase, Christie-Oleza et al. (2017) found that marine phototrophic cyanobacteria cultures are dependent on heterotrophic bacteria contaminants mainly due to their importance in

degrading potentially toxic DOM exudates and regenerating ammonium. The current study aimed to bridge the gap between detailed representations of algae physiology and the role of microbial activity in an accurate way while keeping model complexity low.

Most ecosystem models consider only a single limiting nutrient to control primary production after Liebig's Law of the minimum (Wassmann et al., 2006; Vichi et al., 2007). Yet we know that nutrient co-

limitation is more complex. For example, ammonium and glutamate can inhibit nitrate uptake (Morris, 1974; Dortch, 1990; Flynn et al., 1997), C and N uptake is reduced under Fe limitation, while Si uptake continues (Werner, 1977; Firme et al., 2003), and the effects on photosynthesis differs for nitrogen and silicon limitations and for different algal groups (Werner, 1977; Flynn, 2003; Hohn et al., 2009). Complex interaction models considering intracellular biochemistry ($NH_4$-$NO_3$ co-limitation, Flynn et al., 1997),

transporter densities and mobility (Flynn et al., 2018), and cell cycles (Si limitation, Flynn, 2001) can accurately describe these dynamics (Flynn, 2003), but are ultimately too computationally expensive to be integrated and parameterized in large scale ecosystem models. Some models (Hohn et al., 2009, Le Quéré et al., 2016) implemented multi-nutrient (Hohn et al., 2009) and heterotrophic bacterial dynamics (Le Quéré et al., 2016) in Southern Ocean ecosystem models, but have their limitations in representing

bacterial remineralisation (Hohn et al., 2009), or ammonium and silicate co-limitations (Le Quéré et al., 2016). In contrast to Antarctica, DIN is the primary limiting nutrient for phytoplankton growth while iron is not limiting in most Arctic systems (Tremblay and Gagnon, 2009; Moore et al., 2013).

While simple lab experiments cannot represent all nutrient dynamics found in the environment (e.g. N excretion by zooplankton), they can focus on the quantitatively most important dynamics, to facilitate the

development of simple multi-nutrient models, which are scalable to larger ecosystem models. The present study investigated the impact of silicate, ammonium - nitrate co-limitation and bacterial nutrient regeneration on photosynthesis, nitrogen assimilation, and cellular quotas based on data from a culture based Arctic spring bloom system. The culture consisted of an axenic isolate of *Chaetoceros socialis*, dominating a phytoplankton net haul of a Svalbard fjord. The cultivation experiment was conducted either

under axenic conditions or after inoculation with mostly free-living bacteria cultures, isolated beforehand from the non-axenic culture. Parametrization and insights from these incubations were then used to

develop and parameterize a simple Carbon quota based dynamic model (based on Geider et al., 1998), aiming to keep the number of parameters, and computational costs as low as possible to allow for its implementation in large-scale ecosystem models.

The aims of the study were I) to study the bloom dynamics of a simplified Arctic coastal pelagic system in a culture experiment consisting of one Arctic diatom species and co-cultured bacteria, II) to develop a simple dynamic model representing the observed interactions, and III) to discuss the importance of more complex bloom dynamics for their accurate representation in ecosystem models.

We hypothesize that: I) Bacterial regeneration extends a phytoplankton growth period and gross carbon
fixation; II) Diatoms continue photosynthesis under silicate limitation at a reduced rate if DIN is available; III) Cultivation experiments are powerful for understanding the major spring bloom dynamics.

## 2 Methods

### 2.1 Cultivation experiment

The most abundant phytoplankton species from a net haul (20 µm mesh size) in April 2017 in van
Mijenfjorden (Svalbard) *Chaetoceros socialis* was isolated via the dilution isolation method (Andersen et al., 2005) on F/2 medium (Guillard, 1975). Bacteria were isolated on LB-medium (evaluated by Bertani, 2004) Agar plates using the algae culture as inoculum and sequenced at GENEWIZ LLC using the Sanger method and standard 16S rRNA primers targeting the V1-V9 region (Forwards 5'- AGAGTTTGATCCTGGCTCAG -3', Reverse 5'- ACGGCTACCTTGTTACGACTT -3') provided by
GENEWIZ LLC for identification via blastn (Altschul et al., 1990). Two strains of *Pseudoalteromonas elyakovii*, a taxon previously isolated from the Arctic (Khudary et al., 2008) and known to degrade algae polysaccharides (Ma et al., 2008) and to excrete polymeric substances (Kim et al., 2016), were successfully isolated and used for the experiments. Before the start of the experiment, all bacteria in the algae culture were killed using a mixture of the antibiotics penicillin and streptomycin. The success was
confirmed via incubation of the cultures on LB-Agar plates and bacterial counts after DAPI staining (Porter et al., 1980). The axenic cultures were diluted in fresh F/2 medium lacking nitrate addition (Guillard, 1975) using sterile filtered seawater of Tromsø sound (Norway) as basis. The algae cultures were transferred into 96 200 ml sterile cultivation bottles with three replicates for each treatment. Half of the incubations were inoculated with bacteria cultures (BAC+), while the other half was kept axenic
(BAC-). The cultures were incubated at 4 °C and 100 µE m$^{-2}$ s$^{-1}$ continuous light and mixed 2-3 times a day to keep the algae and bacteria in suspension. We ensured sterile conditions during the experiment by keeping the cultivation bottles closed until sampling. However, endospores may survive the antibiotic treatment in low numbers and start growing especially towards the end of the experiment. Over 16 days three axenic and three BAC+ bottles were sacrificed daily for measurements of chlorophyll a (Chl),
particulate organic carbon (POC) and nitrogen (PON), bacterial and algal abundances, nutrients (nitrate, nitrite, ammonium, phosphate, silicate), dissolved organic carbon (DOC), and the maximum quantum yield (QY) of PSII (Fv/Fm) as a measure of healthy photosystems. Due to technical problems not all replicates could be measured on all days and an overview of replication is given in Table S2.

Chlorophyll a was extracted from a GF/F (50 ml filtered at 200 mbar) filter at 4 °C for 12-24 h in 98%
methanol in the dark before measurement in a Turner Trilogy™ Fluorometer (evaluated by Jacobsen and
Rai, 1990). POC and PON were measured after filtration onto precombusted (4 h at 450 °C) GF/F
(Whatman) filters (50 ml filtered at 200 mbar), using a Flash 2000 elemental analyser (Thermo Fisher
Scientific, Waltham, MA, USA) and Euro elemental analyser (Hekatech) following the protocol by Pella
and Colombo (1973) after removing inorganic carbon by fuming with saturated HCl in a desiccator.
Bacteria were counted after fixation of a water sample for 3-4 h with 2% Formaldehyde (final
concentration), filtration of 25 ml on 0.2 µm pore size Polycarbonate filter, washing with filtered Seawater
and Ethanol, DAPI staining for 7 minutes after Porter et al. (1980), and embedding in Citifluor-
Vectashield (3:1). Bacteria were counted in at least 20 grids under an epifluorescence microscope (Leica
DM LB2, Leica Microsystems, Germany) at 10x100 magnification. In the same sample the average
diameter of diatom cells at the start and end of the experiment was measured. Algae were counted in 2ml
wells under an inverted microscope (Zeiss Primovert, Carl Zeiss AG, Germany) at 20x10 magnification
after gentle mixing of the cultivation bottle. Algae cells incorporated in biofilms after day 9 in the BAC+
cultures were counted after sonication in a sonication bath until all cells were in suspension. Nutrient and
DOC samples were sterile filtered (0.2µm) and stored at -20°C before measurements. Nutrients were
measured in triplicates after using standard colorimetric on a nutrient analyser (QuAAtro 39, SEAL
Analytical, Germany) using the protocols No. Q-068-05 Rev. 12 for nitrate (detection limit = 0.02 µmol
$L^{-1}$), No. Q-068-05 Rev. 12 for nitrite (detection limit = 0.02 µmol $L^{-1}$), No. Q-066-05 Rev. 5 for silicate
(detection limit = 0.07 µmol $L^{-1}$), and No. Q-064-05 Rev. 8 for phosphate (detection limit = 0.01 µmol $L^{-1}$). The data were analysed using the software AACE. The nutrient analyzer was calibrated with reference
seawater (Ocean Scientific International Ltd., United Kingdom). Ammonium was measured manually
using the colorimetric method after McCarthy et al., (1977) on a spectrophotometer (Shimadzu UV-1201,
detection limit = 0.01 µmol $L^{-1}$). Ammonium values > 100 µmol $L^{-1}$ were removed as outliers caused by
too high filtration pressure. DOC was measured by high temperature catalytic oxidation (HTCO) using a
Shimadzu TOC-5000 total C analyser following methods for seawater samples (Burdige and Homstead,
1994). The photosynthetic quantum yield was determined using an Aquapen PA-C 100 (Photon Systems
Instruments, Czech Republic).
Certain factors, such as grazing, settling out of the euphotic zone, and bacterial and algae succession were
not included into the experimental set-up to reduce complexity, and focus on nutrient dynamics. Trace
metals, phosphate, and Vitamin B12 in coastal systems are assumed to be not limiting in Arctic coastal
systems and were supplied in excess to the culture medium. Realistic pre-bloom DOC concentrations
were present in the medium as it was prepared with sterilized seawater from the Fjord outside Tromsø
before the onset of the spring bloom (March 2018).
The f-ratio as indication for the importance of regenerated production (Eppley, 1981) was calculated
based on the average PON fixation in the last three days of the experiment (Eq C1). Here, nitrogen
assimilation in the BAC- culture was assumed to be based on new (nitrate based) production, while
fixation in the BAC+ experiment was assumed to also be based on regenerated (ammonium based)
production.

## 2.2 Model structure

This section outlines the overall model structure followed by a description of the chosen parametrization
approach for each relevant process. Details regarding model equations are provided in the Appendix
(Tables A6 and A7) and a schematic representation of the models is given in Figure 1. We used a dynamic
cell quota model by Geider et al. (1998) to describe the BAC- experiment (G98). We then extended the
G98 model to represent the role of silicate limitation, bacterial regeneration of ammonium, and different
kinetics for ammonium and nitrate uptake (EXT) and fitted it to the BAC+ experiment while retaining
the parameter values previously estimated for G98.

The Geider et al. (1998) model (G98) describes the response of phytoplankton to different nitrogen and
light conditions and is based on both intracellular quotas and extracellular dissolved inorganic nitrogen
(DIN) concentrations, allowing decoupled C and N growth (Fig. 1). Within this model, light is a
controlling factor on photosynthesis and chlorophyll synthesis. C:N ratios and DIN concentrations control
nitrogen assimilation, which is coupled to chlorophyll synthesis and photosynthesis. Chl:N ratios are
controlling photosynthesis and chlorophyll synthesis. G98 has been used in a variety of large scale
ecosystem models with some extensions representing the actual conditions in the environment or
mesocosms (e.g. Moore et al., 2004; Schartau et al., 2007; Hauck et al., 2013). Photoacclimation dynamics
in Geider type models have been evaluated as quick and robust (Flynn et al., 2001), while the N-
assimilation component has some shortcomings in regard to ammonium-nitrate interactions. The original
model of Geider et al. (1998) for C and N was corrected for minor typographical errors (see Ross and
Geider, (2009); Appendix Tables A6 A7).

One aim of the study was to develop a model (EXT) with simplified dynamics of nutrient co-limitation,
which is suitable for future implementation in coupled biogeochemistry-circulation models. The EXT
model keeps all formulations of the G98 and adds dynamics and interactions of silicate, nitrate and
ammonium uptake, carbon and nitrogen excretion and bacterial remineralisation (Fig. 1). The aim of the
model was to describe the response in photosynthesis, chlorophyll synthesis and nitrogen assimilation
with a minimal number of parameters. Hence, dynamics in silicate cycling and bacterial physiology were
highly simplified. The limitations of these simplifications and the potential need for more complex models
are discussed later.

Silicate uptake was modelled using Monod kinetics after Spilling et al. (2010). The effect of silicate
limitation on photosynthesis and chlorophyll synthesis was implemented after findings by Werner (1978),
Martin-Jézéquel et al. (2000), and Claquin et al. (2002). Werner (1978) found that silicate limitation can
lead to a 80% reduction in photosynthesis and a stop of chlorophyll synthesis in diatoms within a few
hours. Hence, we added a parameter for the reduction of photosynthesis under silicate limitation ($Si_{PS}$)
and formulated a stop of chlorophyll synthesis under silicate limitations.

N and Si metabolism have different controls and intracellular dynamics, with N uptake driven by
photosynthesis (as $Pc^{ref}$ in G98) and Si mainly linked to algal respiration (Martin-Jezequel et al., 2000).
Besides earlier cultivation studies, the reduction of photosynthesis after Si limitation has been shown via
photophysiological (inhibited PSII reaction center, Lippemeier et al., 1999) and molecular (down-
regulated photosynthetic proteins, Thangaraj et al., 2019) approaches. In general, we assume that nitrogen
metabolism is not directly affected by silicate limitation (Hildebrand 2002, Claquin et al., 2002), but we

expect cellular ratios to be affected by reduced photosynthesis and chlorophyll synthesis under Si limitation (Hildebrand, 2002; Gilpin, 2004).

The algal respiration term included both respiration and excretion of dissolved organic nitrogen and carbon as a fraction of the carbon and nitrogen assimilated. For testing the importance of DON excretion, we also ran the EXT model without DON excretion (EXT$_{-excr}$). Dissolved organic nitrogen (DON) was recycled into ammonium via bacterial remineralisation. It was assumed that this process is faster for freshly excreted DON compared to DON already present in the medium. Thus, we implemented a labile

DON pool (DON$_l$) for freshly excreted DON and a refractory (DON$_r$) DON pool with the respective remineralization rates rem and rem$_d$. We also assumed that excreted DON and DOC do not coagulate as extracellular polymeric substances (EPS) during the course of the experiment. After Tezuka (1989), net bacterial regeneration of ammonium occurs at DOM C/N molar ratios below 10 and is proportional to bacterial abundances. Higher thresholds up to 29 have been found (e.g., Kirchmann, 2000), but we

selected a lower number to stay conservative. DOM C/N ratios are assumed to be proportional to algae C/N ratios (van der Meersche et al., 2004), with algal molar C/N ratios below 10 representing substrate (DOM) molar C/N ratios below 10.5. Hence, we assumed net bacterial ammonium regeneration to occur at molar POC/PON ratios below 10, while higher ratios lead to bacteria retaining more N for growth than they release. Bacteria abundance change was estimated using a simple logistic growth curve as a function

of DOM since the number of parameters is low (2) and the fit sufficient for the purpose of modelling algae physiology. Michaelis-Menton kinetics based on bacteria growth on DOM with different labilities kinetics could give a more accurate representation of bacterial growth but would not change the fit of the other model parameters aiming for the best fit of the model output to algal PON, POC, Chl, and DIN. Algal nitrate uptake was modelled after the original model by Geider et al. (1998) and ammonium

assimilation was based on the simplified SHANIM model by Flynn and Fasham (1997b), excluding the internal nutrient and glutamine concentrations. Ammonium uptake is preferred over nitrate (lower half saturation constant) and reduces nitrate assimilation if available above a certain threshold concentration of ammonium (Dortch, 1990; Flynn, 1999). Ammonium is the primary product of bacterial regeneration N-compound after remineralization of DON. Nitrification was assumed to be absent, since the bacteria in

our experiment are not known to be capable of nitrification.

## 2.3 Model fitting

The model was based on a set of ordinary differential equations (ODEs) and was written in R. All model equations are provided in the Appendix (Table A6 and A7) and the newest version of the R code is available on GitHub (https://github.com/tvonnahm/Dynamic-Algae-Bacteria-model) and the version

used in this manuscript archived at zenodo (doi.org:10.5281/zenodo.4459550). The ODEs were solved using the ode function of the deSolve package (Soetart et al., 2010) with the 2nd-3rd order Runge-Kutta method with automated step size control.

The parameters of the G98 model were fitted to the BAC- experiment data and those of the EXT model were fitted to the BAC+ experiment data. The model fitting started with data from day 1 in order to avoid

artifacts during acclimation of the cultures after transfer to a new medium. Si and NO$_X$ were not measured at day 1 and the mean of day 0 and day 2 was used. The G98 parameter values were fitted first and retained without changes for the EXT model fitting. The maximum Chl:N ratio ($\theta^N_{max}$), minimum and maximum

N:C ratios ($Q_{min}$, $Q_{max}$), and irradiance (I) were available as experimental data and needed no further fitting (Table A2). The start values and constraints for the remaining six variables ($\zeta$, $R^C$, $\alpha_{Chl}$, n, $K_{no3}$,

$P^C_{ref}$, Table A3) were based on model fits of G98 to other diatom cultures in previous studies (Geider 1998, Ross and Geider 2009). The parameters were first fitted manually via graphical comparisons with the experimental data (POC, PON, Chl, DIN, Fig. 5 and 5), and via minimizing the model cost calculated as the root of the sum of squares normalized by dividing the squares with the variance (RMSE Eq. C2, Stow et al., 2009). The initial manual tuning approach allowed control of the model dynamics, considering

potential problems with known limitations of the G98 model (e.g. lag phase not modelled; Pahlow, 2005). The manual tuning also allowed obtaining good start parameters for the automated tuning approach and sensitivity/ collinearity analyses, which are sensitive to the start parameters.

After the manual tuning, an automated tuning approach was used to optimize the fits. The automated tuning was done using the FME package (Soetart et al., 2010b), a package commonly used for fitting

dynamic and inverse models based on differential equations (i.e., deSolve) to measured data. The automated analyses were based on minimizing the model cost calculated as the sum of squares of the residuals (SSR, Fitted vs measured data). The experimental data were normalized so that all normalized data were in a similar absolute range of values. This involved increasing Chl and PON values by an order of magnitude while decreasing DIN ($NH_4$ + $NO_3$) data by one order of magnitude. The data were not

weighted, assuming equal data quality and importance. Prior to the automated fitting, parameters were tested for local sensitivity (SensFun) and collinearity, or parameter identifiability (collin; e.g., Wu et al., 2014). sensFun tests for changes in output variables at each time point based on local perturbations of the model parameter. The sensitivity is calculated as L1 and L2 norms (Soetart et al. 2009; Soetart et al., 2010b). The sensFun output is further used as input for the collinearity, or parameter identifiability

analyses. Parameters were considered collinear and not identifiable in combination with a collinearity index higher than 20 (Brun et al., 2001). In this case, only the more sensitive parameter was used for further tuning. Eventually, $R^C$, $K_{no3}$, n, and $\alpha_{Chl}$ were subject to the automated tuning approach using the modfit function, based on minimizing the SSR within the given constraints. Parameters were first fitted using a Pseudorandom search algorithm (Price, 1977) to ensure a global optimum. The resulting

parameters were then fine-tuned using the Nelder-Mead algorithm (Soetart et al., 2010b ) for finding a local optimum. A model run with the new parameters was then compared to the initial model via graphical comparisons of the model fit to the experimental data, and via the RMSE value.

The parameter values obtained for the G98 fit to the BAC- experiment were retained without changes or further fitting in the EXT model. The additional parameters of the EXT model were then fitted to the

BAC+ experimental data (POC, Chl, PON, DIN). The model was only fitted to total DIN, due to the potential uncertainties related to ammonium immobilization in the biofilm, which could be released during filtration and be part of the measured data. In fact, a test run, fitting the EXT model to $NO_3$ and $NH_4$ separately lead to a substantially worse overall fit (RMSE=3.49). Otherwise, the data were not weighted. Since the aim of the study was to model the effects of silicate and bacteria on algae growth and

not to develop an accurate model for bacteria biomass and silicate concentrations, the parameters $\mu_{bact}$, $bact_{max}$, $K_{si}$, and $V_{max}$ were only fitted to the corresponding data (Bacteria, Silicate) prior to fitting the other parameters of the EXT model. Bacterial growth parameters ($\mu_{bact}$, $bact_{max}$) were fitted to the bacterial growth curve using common bacterial carbon conversion units (20 fg C per cell as described by Lee and Fuhrman, 1987). Silicate related parameters ($K_{si}$, $V_{max}$) were constrained by the study of Werner (1978)

and fitted to the measured silicate concentrations. The remaining parameters were subject to the tuning approach described for G98. Ammonium related parameters ($K_{nh4}$, $nh4_{thres}$) were constrained by measured ammonium concentrations, and constants available for other diatom taxa described by Eppley et al. (1969). Remineralization parameters for excreted (rem) and background ($rem_d$) DOM were constrained by the data with the limitation of rem > $rem_d$, assuming that the excreted DOM is more labile. The
parameters related to the effect of silicate limitation on photosynthesis and chlorophyll production ($s_{min}$, $Si_{PS}$) were constrained by the study of Werner (1978) and fitted as described for G98. None of the added parameters were collinear/ unidentifiable or given by the measured data and thus retained for the automated tuning approach. Eventually, the 15 parameters (Table A3) were fitted against 160 data points (Table A1).
Due to the biofilm formation in the stationary phase of the BAC+ experiment, we tested two additional modelling approaches representing different dynamics in biofilms: i) DOC coagulation to EPS as part of the POC pool, which was assumed absent in our EXT model (Schartau et al., 2007), and ii) Increased DOM excretion in the stationary phase (e.g. Christie-Oleza et al., 2017). However, we suggest that the photosynthesis reduction term $Si_{PS}$ can give very similar model outputs, while being similarly or more
sensitive. Thus, we tested the sensitivity of the added parameters of the two extended biofilm models in comparison to $Si_{PS}$ by testing the magnitude of perturbations of $Si_{PS}$ needed to reverse the effects of the added biofilm parameter (Fig. S1-3). The effects could be mostly reversed with similar or less perturbations of $Si_{PS}$. However, DOC coagulation to EPS can yield in a better overall better fit than $Si_{PS}$ alone. The main effect of the biofilm that we could not model with the available data appears to be
ammonium immobilization in the biofilm, potentially by adsorption, accumulation in pockets, or conversion to ammonia due to the locally reduced pH caused by increased bacterial respiration. Model stability was estimated by extending the model run for 120 days, to test for unrealistic model dynamics (Fig. S3).

# 3 Results

## 3.1 Cultivation experiment

The concentrations of nitrate and silicate declined rapidly over the course of the experiment (Fig. 2). After eight days, silicate decreased to concentrations below 2 µmol $L^{-1}$ a threshold known to limit diatom dominance in phytoplankton (Egge and Aksnes, 1992), while inorganic nitrogen (nitrate, nitrite, and ammonium) became limiting (<0.5 µmol $L^{-1}$ , POC:PON >8-9 DIN:DIP<16) only in the BAC- culture.
DIN:DIP ratios far below 16, or DIN concentrations below 2 µmol $L^{-1}$ have been described as indication for DIN limitation (Pedersen and Borum, 1996), as well as POC:PON ratios >9 (Geider and La Roche, 2002). Phosphate was not potentially growth limiting with molar DIN to $PO_4$ ratios consistently far below 16 (Redfield, 1934) and concentrations around 15 µmol $L^{-1}$. Typically, phosphate concentrations below 0.3 µmol $L^{-1}$ are considered limiting (e.g., Haecky and Andersson, 1999). Regeneration of ammonium
and phosphate were important after eight days as seen by increasing concentrations of both nutrients and showed higher concentrations in the BAC+ experiments compared to the BAC- cultures (Fig. 2a,b). Ammonium concentrations were consistently higher, and nitrate was removed more slowly in the

presence of bacteria, especially during the exponential phase. With the onset of the stationary phase in the BAC+ experiment, $PO_4$ and $NH_4$ concentrations doubled within 2 to 4 days and stayed high with variations in phosphate concentrations, while they stayed low in BAC-. With depletion of $NO_3$ in BAC+, $NH_4$ concentrations remained high, while $PO_4$ concentrations dropped. While not all ammonium measured is also available for algae growth, discussion of the dynamics (decrease in the start, increase with the onset of the stationary phase), especially if also shown in the EXT model, are still useful to understand multi-nutrient dynamics (e.g. regeneration). Considering the overall higher concentrations of $NO_3$, compared to $NH_4$, discussions of total DIN dynamics, DIN:DIP ratios, and limitations are also meaningful. DOC values were very high from the start (approx. 2-4 mmol $L^{-1}$) and remained largely constant throughout the experiment (Table A8).

The diatom *Chaetoceros socialis* grew exponentially in both treatments until day 8 before reaching a stationary phase with declining cell numbers (Fig. 3). The growth rate of the BAC- culture (0.36 $d^{-1}$) was slightly lower than in the treatment with bacteria present (0.42 $d^{-1}$) during the exponential phase. Algal cellular abundance was higher in the BAC+ cultures. Towards the end of the exponential phase, the diatom started to form noticeable aggregates in cultures with bacteria present, but only to a limited extent in the BAC- cultures. Such aggregate formation with associated EPS production is typical for *C. socialis*. With the onset of the stationary phase in the BAC+ cultures about 30% of the cells formed biofilms on the walls of the cultivation bottles (estimated after sonication treatment). Bacteria (Fig. 3) continued to grow throughout the entire experiment, but growth rates slowed down from 0.9 to 0.6 after day 8. In the BAC- cultures, bacterial numbers increased after 8 days, but abundances remained two order of magnitude below the BAC+ cultures and effectively BAC- over the experimental incubation period. The maximum photosynthetic quantum yield (Fv/Fm) is commonly used as a proxy of photosynthetic fitness (high QY), indicating the efficiency of energy transfer after adsorption in photosystem II. Low values are typically related to stress, including for example nitrogen (Cleveland and Perry, 1987), or silicate (Lippemeier et al., 1999) limitation. We found an increase in QY from approx. 0.62 to 0.67 in the exponential phase and a decrease to approx. 0.62 in the BAC+ treatment after 8 days and to approx. 0.58 in the BAC- treatment (Table A8).

During algal exponential growth, POC and PON concentrations followed changes in algal abundances increasing four, seven, and 19-fold respectively, within 8 days (Fig. 3a, 4). Interestingly, with the beginning of the stationary phase, POC and PON continued to increase in the BAC+ cultures, while their concentrations stayed constant (POC), or decreased due to maintenance respiration (PON) in BAC- cultures. POC and PON concentrations were consistently higher (1.2 times POC, 1.4 times PON) in BAC+ cultures during the exponential phase. POC:PON ratios decreased in both cultures, but increased again after 11 days in the BAC- culture. Chlorophyll *a* concentrations also increased exponentially over the first eight days in both treatments, and thereafter decreased within the stationary phase in the BAC- cultures. In contrast, cell numbers remained nearly constant in the BAC+ cultures, before declining at the last sampling day.

Overall, both experimental cultures showed similar growth dynamics until day 8, with silicate becoming limiting for both treatments and nitrogen only limiting in BAC- cultures. Algal growth with bacteria present was slightly, but consistently higher during this phase. After eight days, algae growth stopped in both treatments, but nitrogen and carbon were continuously assimilated in BAC+ cultures. BAC- cultures started to degrade chlorophyll, while it stayed the same in BAC+ cultures. Algal abundances in the BAC+

treatment at the end of the experiment were ca 30% higher due to biofilm formation, and considerably more carbon (2x total POC, or 10-20% per cell) and nitrogen (3x total PON) per cell had been assimilated, and considerably more chlorophyll (2-3x total chlorophyll) produced at day 16. Cell size differences were not detectable (ca 4μm diameter, Table A8). POC to PON ratios increased after 11 days in BAC- cultures but showed no change in BAC+ cultures. POC to Chl ratios were comparable in both treatments (Fig. 5).

Assuming BAC- N fixation was mostly based on new production (nitrate as N source), while the algal N fixation in bacterial enriched treatments was based on new and regenerated (ammonium as N source) production, two-thirds of the production was based on regenerated production (f-ratio = 0.31).

## 3.2 Modelling

A comparison of the traditional G98 model with the EXT model allowed an estimate of importance of
bacterial DIN regeneration and Si co-limitations for describing the experimental growth dynamics. The EXT model led to a slightly worse fit to the BAC- experiment ($RMSE_{G98}$ = 2.79 $RMSE_{EXT}$ = 3.38, Fig. 5 & 6). The real strength of the EXT model was in representing growth dynamics with bacteria present (Fig. 5 & 6). Here, the RMSE was reduced by 47% from $RMSE_{G98}$ = 4.31 down to $RMSE_{EXT}$ = 2.31. Both, the G98 and EXT model fits of the BAC- experiment were similarly good for POC and PON with
a slightly lower modelled growth rate. PON in the BAC+ experiment was poorly modelled without consideration of silicate limitation or regenerated production specifically towards the end of the exponential phase and during the stationary phase. Maximum PON values were about 3 times lower using the G98 model (Fig. B3). In addition, the start of the stationary phase in the BAC+ experiment was estimated 3 days too late via G98, even though modelled DIN was depleted three days too soon (Fig. B3).
Under BAC- conditions, where silicate limitation does not play a major role the G98 model appears sufficient.

The EXT model allowed representing detailed dynamics in a bacteria influenced system such as the responses to silicate limitation with a decrease in POC production, continued PON production, and the stagnation of Chl synthesis (Fig. 5). Apart from the lag phase, the mass ratios of gC:gN and gC:gChl were
represented accurately (Fig. 5). The model fits of POC, PON and Chl without the separate carbon excretion term ($x_f$) were lower compared to the model with excretion, indicating the importance of excreted dissolved organic matter (DOM) concentrations ($RMSE_{EXT-exr}$ of 5.73).

DIN dynamics caused by ammonium – nitrate interactions were represented well (Fig. 6a). However, at the onset of the stationary phase, ammonium concentrations of the model were one order of magnitude
lower than in the experiment, showing a major weakness (Fig. 6c). Increased weighting of ammonium during the model fitting led to a slightly better fit to ammonium, but a substantially worse fit of the model to POC, PON, and Chl ($RMSE_{EXT}$=3.49). This indicates that the problem lies with the ammonium data, which include immobilized ammonium in the biofilm released during the filtration, but unavailable for diatom growth. Other potential differences in biofilms were tested by means of different model extensions
(DOC and DON aggregation to EPS, increase DOM excretion). After including the DOC and DOM aggregation to the model, the overall fit was improved ($RMSE_{EXT}$=2.31, $RMSE_{EXT+eps}$=2.19, Fig. S1). However, in the absence of EPS data, we used the EXT model for the main discussion. Increased DOM excretion could be explained by the SiPS term of the EXT model (Fig. S2). The silicate uptake estimation was highly simplified using simple Monod kinetics, leading to too high modelled values in the stationary

phase and a too quick depletion in the start (Fig. 6d). Carbon excretion ($x_f$) lead to $NO_X$ depletion after 8 days.

The sensitivity analysis (Table A4) revealed that the sensitivity of the added parameters in EXT is overall comparable to the sensitivity of the original parameters in G98. The model outputs were most sensitive to $P_C^{Ref}$ (L1=1.27, L2=2.05), which is a parameter in both G98 and EXT. The most sensitive added

parameters in EXT were the DON excretion rate ($x_f$, L1=0.17, L2=0.24), the bacterial growth rate ($\mu_{bact}$, L1=0.18, L2=0.36), and the remineralisation rate of refractory DON ($rem_d$, L1=0.09, L2=0.21), and the inhibition of photosynthesis under Si limitation ($Si_{PS}$, L1=0.07, L2=0.18), which was overall comparable to other sensitive parameters of the G98 model ($Q_{max}$, $R^C$, $\alpha_{Chl}$, $\zeta$, $n$, $I$, $\Theta_N^{max}$, Table A1). Small perturbations of the parameters only indirectly related to the fitted output variables did not lead to changes

in POC, PON, Chl, or DIN.

## 4 Discussion

The experimental incubations showed that in the presence of bacteria both the growth period and gross carbon fixation were extended (Hypothesis I). The diatoms were able to continue photosynthesis under silicate limitation at a reduced rate as long as inorganic nitrogen was present (Hypothesis II). Overall, the

experimental incubations represented typical spring bloom dynamics for coastal Arctic systems, including an initial exponential growth phase terminated by N and Si limitation (Hypothesis III) and the potential for an extended growth period via regenerated production. Our model incorporating these results was able to reflect these dynamics by adding $NH_4$-$NO_3$-$Si(OH)_4$ co-limitations and bacterial $NH_4$ regeneration to the widely used G98 model. In addition, bacteria-algae interactions, DOC and biofilm dynamics were

important in the experiment, but those were not crucial for quantitatively modelling algal C:N:Chl quotas. While *C. socialis* may not be the dominant species in all coastal Arctic phytoplankton blooms, we argue that it is representative for chain-forming diatoms typically dominating these systems due to their shared needs and responses to nutrient limitations (e.g., Eilertsen et al., 1989; von Quillfeldt, 2005).

### 4.1 Silicon-nitrogen regeneration

Spring phytoplankton dynamics in Arctic and sub-Arctic coastal areas is typically characterized by an initial exponential growth of diatoms, followed by peaks of other taxa (like *Phaeocystis pouchetii*) soon after the onset of silicate limitation (Eilertsen et al. 1989). Thus, a shift in species composition for the secondary bloom is linked to silicate limitation prior to final bloom termination caused by inorganic nitrogen limitation. As suggested by our second hypothesis, photosynthesis was reduced by approx. 70%

after silicate became limiting, which is comparable to earlier experimental studies (Tezuka, 1989). However, as suggested by our first hypothesis, the secondary bloom was extended in time by bacterial regeneration of ammonium, allowing regenerated production to contribute about 69% of the total production (f-ratio=0.31) even during the diatom dominated scenario in our experimental incubation. With the start of the stationary phase, $NH_4$ and $PO_4$ concentrations doubled, presumably due to decreased

assimilation by the silicate-starved diatoms and increased regeneration by bacteria, supplied with increasing labile DOM (doubled remineralisation rate in EXT) excreted by the stressed algae. However, $NH_4$ concentrations doubled within four days, while $PO_4$ concentrations doubled in only one day,

indicating some unexplained internal dynamics, potentially via different bacterial uptake and release of N and P. After $NO_3$ depletion at day 15, also the $PO_4$ concentrations dropped, indicating a coupling of
N:P metabolism, but not of $NH_4$:P metabolism. Thus, the sudden drop may also indicate dynamics of bottle experiments, not accounted for, showing potential limitations of these experiments. The presence of bacteria and thus regenerated production allowed diatom growth to continue 8 days after silicate became limiting (Figs. 2, 3 & 4), nearly doubling the growth period similar to observations in the field (e.g. Legendre and Rassoulzadegan, 1995; Johnson et al., 2007), which supports our Hypotheses I and
III.
The G98 model has its most severe limitation in the modelling of PON, simply due to the lack of the ammonium pool, supplied via bacterial regeneration. The substantially better fit of PON in the EXT model shows therefore clearly that bacterial remineralisation is crucial to successfully model spring bloom dynamics, especially near bloom termination. Many biogeochemical models used in the Arctic include
remineralisation but rely on fixed or temperature dependent rates and do not consider them bacteria-dependent (MEDUSA, LANL, NEMURO, NPZD, see Table 1). While this simplification allows modelling regenerated production, using bacteria-independent remineralisation rates does have limitations under spring bloom scenarios, where bacteria biomass can vary over orders of magnitudes (e.g. Sturluson et al., 2008) as also seen in our experimental study.
While we do not expect the f-ratio in our bottle experiment to be directly comparable to open ocean system  due to the higher biodiversity of field communities, a comparison can aid to identify limitations in our experiment and model. The f-ratio can be used as a measure to check how representative the cultivation study was for typical spring bloom dynamics (Hypothesis III). Regenerated production is significant in polar systems and our estimated experimental f-value of 0.31 is slightly below the average
for polar systems (Harrison and Cota, 1990, mean f-ratio=0.54). Nitrification is a process supplying about 50% of the $NO_3$ used for primary production in the oceans, which may lead to a substantial underestimation of regenerated production (Yool et al., 2007), inflating the f-ratio interpreted as estimate for new production, potentially also in the study by Harrison and Cota (1990). The absence of vertical PON export in our experiment may be another explanation for the above average fraction of regenerated
production. In the ocean environment, regenerated production is also affected by vertical export (sedimentation) and grazing which are not represented in the experimental incubations. Via sedimentation, a fraction of the bloom either in the form of direct algal sinking of fecal pellets is typically exported to deeper water layers, reducing the potential for N regeneration within the euphotic zone (e.g. Keck and Wassmann, 1996). Larger zooplankton grazing can lead to increased export of PON via fecal
pellet aggregation, or diel vertical migration (Banse, 1995), but may also release ammonium and urea (Conover and Gustavson, 1999, Saiz et al., 2013).
In contrast, bacterial death by microflagellate grazing and viral lysis may supply additional nutrients, or DON available for N regeneration in the euphotic zone (e.g. Goldman and Caron, 1985), which potentially leads to an overestimation of regenerated production. Another potentially important N source for
regenerated production may be urea (Harrison et al., 1985), which would lead to an even higher importance of regenerated production as suggested by our study. Hence, ecosystem scale models will need to consider these dynamics regarding bacterial abundances, microbial networks and particle export in addition to bacterial remineralization in order to model realistic ammonium regeneration in the euphotic

zone. Overall, our cultivation experiment was powerful to represent major aspects of spring bloom dynamics, but has its limitations, thereby confirming our third hypothesis only to some extent.

Bacteria-mediated silicate regeneration is absent from the modelling approach, as indicated by the identical silicate concentrations in both treatments and models (Fig. 2d). In the environment silicate dissolution is, in fact, mostly described as an abiotic process with temperature as the main control, and a minor contribution by bacterial remineralisation (Bidle and Azam, 1999). Our experiment indicates that silicate dissolution for *Chaetoceros socialis* was negligible at cold temperatures and the time scale of the incubations and typical for bloom durations and residence times of algae cells in the euphotic zone (Eilertsen et al., 1989, Keck and Wassmann, 1996). We conclude that silicate dissolution in coastal Arctic systems happens most likely in the sediment or deeper water layers and is only supplied via mixing in winter. In Antarctica substantial silicate dissolution has been observed but not in the upper 100 m, which has been related to the low temperatures (Nelson and Gordon, 1981) in agreement with our conclusion. Hence, modelling silicate regeneration in the euphotic zone is not necessary in these systems.

## 4.2 Algal growth response to Si and N limitation

The response of diatoms to Si or N limitation is based on different dynamics and different roles of N and Si in diatom growth. N is needed for proteins and nucleic acids and its uptake is mainly driven by phototrophic reactions (Martin-Jézéquel et al., 2000). Si is only needed for frustule formation and cell division, mostly during a specific time in the cell cycle (G2 and M phase, Hildebrand, 2002) and the assimilation is mostly driven by heterotrophic reactions (Martin-Jézéquel et al., 2000). Once N is limiting, growth rapidly stops (Geider et al. 1998). In the case of Si limitation, however, growth can continue with a slower rate if N is still available (Werner, 1978; Gilpin et al., 2004). Several studies found a reduced growth rate with weaker silicified cell walls (Hildebrand, 2002; Gilpin, 2004), but unaffected nitrogen assimilation under silicate limitation (Hildebrand 2002, Claquin et al., 2002) in accordance with our study. Claquin et al. (2002) found variable Si:C and Si:N ratios and highly silicified cells under nitrogen limitation, indicating uncoupled Si and N:C metabolism.

Nitrogen is a crucial element as part of amino acids and nucleic acids, which are necessary for cell activity and growth. If N becomes limiting major cellular processes are affected and growth or chlorophyll synthesis is not possible. Photosynthesis can continue for a while leading to carbon overconsumption (Schartau et al., 2007), which is well modelled by G98 for both BAC+ and BAC-. A part of the excess carbon can be stored as intracellular reserves, and a part is excreted as DOC, which may aggregate as EPS, contributing to the total POC pool. The excess carbon can potentially be toxic for the algae and excretion and extracellular degradation by bacteria may be crucial for algal survival (Christie-Oleza et al., 2017). Quantitatively, N limitation is well modelled by G98 under BAC- conditions, if only one nitrogen source plays a role. However, under longer nitrogen starvation times or higher light intensities, alternative models that include carbon excretion and aggregation (Schartau et al., 2007) or intracellular storage in reserve pools (Ross & Geider 2009) might be needed. Our growth experiment shows clearly, that C:N ratios are not fixed and variable quotas are needed. Vichi et al. (2007) estimated that Carbon based models that do not consider variable C/N ratios may underestimate net primary production (NPP) by 50%, arguing for the importance of quota based models (Fransner et al., 2018). However, most ecosystem scale models are simplified by using fixed C:N ratios (Table 1). The next step towards quota

based-models is the consideration of more detailed cell based characteristics, such as transporter density,
cell size, and mobility, including sedimentation (Aksnes and Egge, 1991). Flynn et al. (2018) discuss a model with detailed uptake kinetics showing that large cells are overall disfavored over small cells due to higher half saturation constants, but that they may still have competitive advantages due to lower investment in transporter production. Also increased sedimentation in larger cells increases the mobility and may offset the disadvantage of a larger size. While this extension is too complex for our aim of a
simplified model, the dynamics may become important when modelling different algae taxa.

The type of inorganic nitrogen available also affects nitrogen uptake. Due to the metabolic costs related to intracellular nitrate reduction to ammonium, ammonium uptake is preferred over nitrate, potentially leaving more energy for other processes (Lachmann et al., 2019). Ammonium can even inhibit or reduce nitrate uptake over certain concentrations (Morris, 1974). The dynamics are mostly controlled by
intracellular processes, such as glutamate feedbacks on nitrogen assimilation, cost for nitrate conversion to ammonium, or lower half saturation constants of ammonium transporters (Flynn et al., 1997). The most accurate representation of these dynamics is given in the ANIM model by Flynn et al. (1997), but the model is too complex for implementations in larger ecosystem models. The number of parameters is difficult to tune with the typically limited availability of measured data and its complexity makes it also
computationally costly to scale up the models. Typically, modelling ammonium-nitrate interactions by different half-saturation constants and inhibition of nitrate uptake by ammonium appears sufficient (e.g., BFM, LANL, NEMURO, Table 1) and has been adapted in our model.

Studies on the coupling of silicate limitation on C, N, and Chl show inconclusive patterns, including a complete decoupling (Claquin et al., 2002), a relation of N to Si (Gilpin et al., 2004) and reduction of
photosynthesis without new chlorophyll production (Werner, 1978; Gilpin et al., 2004). Cell size is limited by the frustules, but cells may become more nutritious (higher N:C ratio), or simply excrete more DOM, which may aggregate and contribute to the PON and POC pools. A detailed cell-cycle based model has been suggested by Flynn (2001), but the number of parameters (30) makes the model too complex for ecosystem scale models. In ecosystem scale models Si limitation is modelled in various simplifications,
such as thresholds triggering a stop (MEDUSA), or reduction (e.g. BFM, MEDUSA, SINMOD) of the Si dependent production (Table 1), or Si:N ratio scaled production (NEMURO, Table 1).

Our cultivation study shows i) that a threshold value in the model, leading to a stop or solely Si dependent photosynthesis has its limitations, since DIN controlled photosynthesis continues at lower rates, and ii) that coupling of Si:N:C:Chl is present. We do not expect a direct Si:N coupling, due to different controls
of Si and N metabolism (Martin-Jézéquel et al., 2000), but suggest indirect coupling via reduced photosynthesis. In fact, detailed photophysiological and molecular approaches under Si limitation found inhibited PSII reaction centers (Lippemeier et al., 1999) similar to the decreased QY in our experiment, and down-regulated photosynthetic proteins (Thangaraj et al., 2019) under Si limitation. Thus, we modelled the response of diatom growth to silicate limitation by reducing photosynthesis through a
parameterized fraction (Si$_{PS}$) and a stop of chlorophyll synthesis below a certain threshold, based on experimental studies (Werner, 1978, Lippemeier et al., 1999, Gilpin et al., 2004, Thangaraj et al., 2019) and in accordance to other ecosystem scale approaches. We suggest that this extension is more accurate than the typical threshold-based dynamics, with one limiting nutrient controlling the growth equally for POC and Chl production (e.g. SINMOD by Wassmann et al., 2006; BFM by Vichi et al., 2007), while

still keeping the number of parameters low compared to very detailed cell-cycle based models (e.g. Flynn, 2001, Flynn et al., 2018).

## 4.3 Importance of algae-bacteria interactions and DOC excretion

As described above, N or Si limitation can lead to excretion of DON and DOC, which can aggregate as EPS and be available for bacterial regeneration of ammonium. For accurately including EPS dynamics in
the model additional data would be needed. However, the importance of EPS formation is evident in the end of the BAC+ experiment. Firstly, a biofilm was clearly visible containing about 30% of the algae cells. While we would not expect biofilms in the open ocean, aggregation of algae cells, facilitated by EPS is common towards the end of spring blooms, increasing vertical export fluxes (e.g. Thornton, 2002). *Chaetoceros socialis* is in fact a colony forming diatom building EPS-rich aggregates in nature (Booth et
al., 2002). Secondly, POC and PON concentrations increased, while cell numbers and sizes stayed constant, showing that the additional POC and PON was most likely part of an extracellular pool. Silicate limitation could be one trigger for enhanced exudation. In fact, the biofilm dynamics evaluated (DOC aggregation, increased excretion) showed similar dynamics as the $Si_{PS}$ term. Since the biofilm formation corresponds with silicate limitation, it is difficult to untangle the direct effects of the biofilm, or the
indirect effects of silicate limitation, without additional data or experiments (e.g. EPS measurements, DOM characterization). However, our model run, including EPS aggregation allowed an improved overall fit, while reducing the $Si_{PS}$ term to values more similar to earlier studies (Werner, 1978), pointing to the importance of DOC and DON aggregation. Hence, we suggest that Si limitation and EPS aggregation are both important.
Interestingly, algae – bacteria interactions can be species specific with specific organic molecules excreted by the algae to attract specific beneficial bacteria (Mühlenbruch et al., 2018). Thereby bacteria are crucial for recycling ammonium, but also to degrade potentially toxic exudates (Christie-Oleza et al., 2017).
In the BAC- experiment, Carbon excretion after Carbon overconsumption could be expected after
Schartau et al. (2007), but no indications, such as biofilm formation, or increased POC per cell were found. In fact, a model considering EPS aggregation lead to a substantially worse fit to BAC-. This indicates that carbon overconsumption has been of minor importance likely due to the low light levels. An alternative explanation is that bacteria and potentially chemotaxis are important controls on algal carbon excretion (Mühlenbruch et al., 2018). Overall, DOM excretion and EPS dynamics appear to play
a major role in quantitatively modelling C:N:Chl quotas in our experiment, with higher errors ($RMSE_{EXT-excr}$=5.73, $RMSE_{EXT}$=2.31) for a model run without than with the excretion term $x_f$.

## 4.4 Considerations in a changing climate

Due to a rapid changing climate, especially in Arctic coastal systems, the dynamics addressed in this study will change (Tremblay and Gagnon 2009). With warmer temperatures, heterotrophic activities, and
thereby bacterial recycling will increase (Kirchman et al., 2009). Our study showed that regenerated production is crucial for an extended spring bloom. Hence, higher heterotrophic activities may lead to extended blooms (increased bacterial regeneration). At the same time, higher temperatures and increased

precipitation will lead to stronger and earlier stratified water columns, which will lead to less nutrients reaching the surface by winter mixing, reducing new production (decreased bacterial regeneration)(Tremblay and Gagnon, 2009; Fu et al., 2016). Consequently, the phenology of Arctic coastal primary production in a warmer climate will likely be increasingly driven by bacterial remineralization, showing the necessity to include this process into biogeochemical models. An earlier temperature driven water column stratification may also lead to an earlier bloom. However, due to increasing river and lake brownification and sediment resuspension, the spring bloom may also be delayed (Opdal et al., 2019). With decreased light, carbon overconsumption as described by Schartau et al. (2007) may become less important due to decreased photosynthesis. An earlier or later phytoplankton bloom can lead to a mismatch with zooplankton grazers (Durant et al., 2007; Sommer et al., 2007). Reduced zooplankton production would decrease the fecal pellet driven vertical export and thereby increase the residence time of POM in the euphotic zone and the potential for ammonium regeneration. Thus, the incorporation of bacterial recycling into ecosystem models may be even more important under this scenario. In fact, global climate change models agree that vertical carbon export is decreasing overall (Fu et al., 2016). Silicate regeneration is thought to be mostly controlled abiotically by temperature (Bidle and Azam, 1999). Thus, increasing temperature and a stronger stratification will allow recycling of silicate in the euphotic zone before sinking out and thus could cause a shift in the algal succession observed during spring with prolonged contributions of diatoms (Kamatani, 1982). Thus, a temperature dependent silica dissolution may need to be included for models in a substantially warmer climate in further model developments. Increased precipitation will also lead to increased runoff and allochthonous DOM inputs, increasing the importance of terrestrial DOM degradation and decreasing the relative importance of algal exudate regeneration (Jansson et al., 2008). The, low light levels, and the absence of grazing and export fluxes are simplifications of our study, which are, however, expected to be realistic scenarios under climate change. Hence, we suggest that our experiment and model are well suited as a baseline for predictive ecosystem models investigating the impacts of climate change on coastal Arctic spring blooms. However, climate change may lead to shifts in algae communities with non-silicifying algae dominating over diatoms (e.g. Falkowski and Oliver, 2007), reducing the importance of silicate limitation. Thus, conducting similar experiments and modelling exercises with a wider range of algal taxa and different temperature and nutrient regimes is suggested.

## Acknowledgements

The project was supported by ArcticSIZE - A research group on the productive Marginal Ice Zone at UiT (project number 01vm/h15). We want to thank Paul Dubourg and Elzbieta Anna Petelenz-Kurdziel for the help with Nutrient and POC/PON analyses. DOC analyses were supported through a Fulbright Distinguished Scholar Award to HRH.

## Authors contributions

TRV designed the experiment with contributions by RG and ML. TRV isolated and identified the cultures. ML performed the experiment with contributions of TRV and UD. RH measured DOC and SK measured

the Nutrients. The other parameters where measured by ML and TRV. TRV programmed the model with contributions of CV, ST and DvO. TRV wrote the manuscript with contributions from all co-authors.

## Data availability

The experimental data are archived at DataverseNO under the doi number doi.org/10.18710/VA4IU9. The Rscripts for the model used in this publication are archived at zenodo
(doi.org/10.5281/zenodo.4459550) and available at github under https://github.com/tvonnahm/Dynamic-Algae-Bacteria-model.

## Competing interests

The authors declare that they have no conflict of interest.

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

 **Figures**

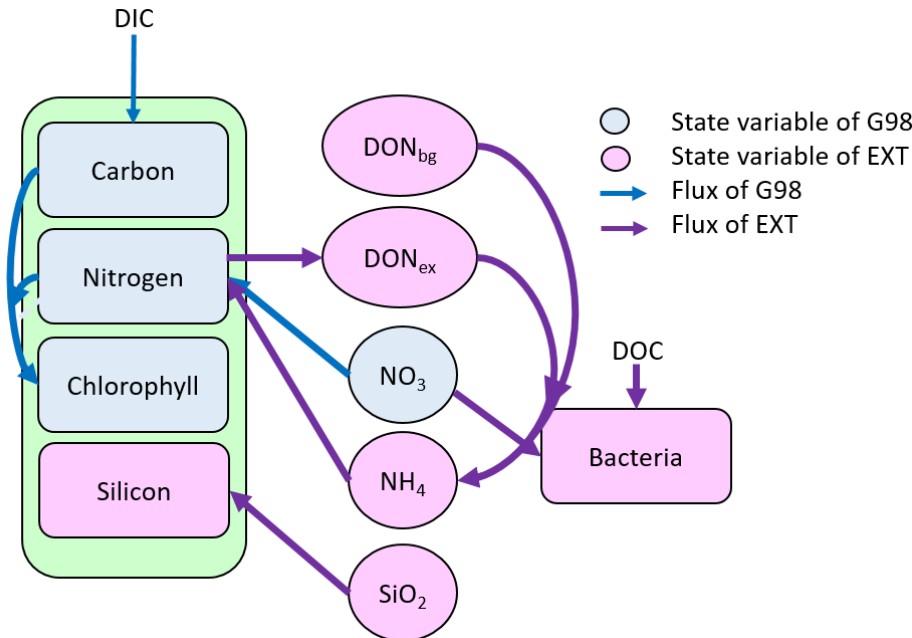

Figure 1. Schematic representation of the state variables and connections and controls in the G98 model (blue) and EXT model (purple). The EXT model has the same formulations as G98 with the additions shown in purple.

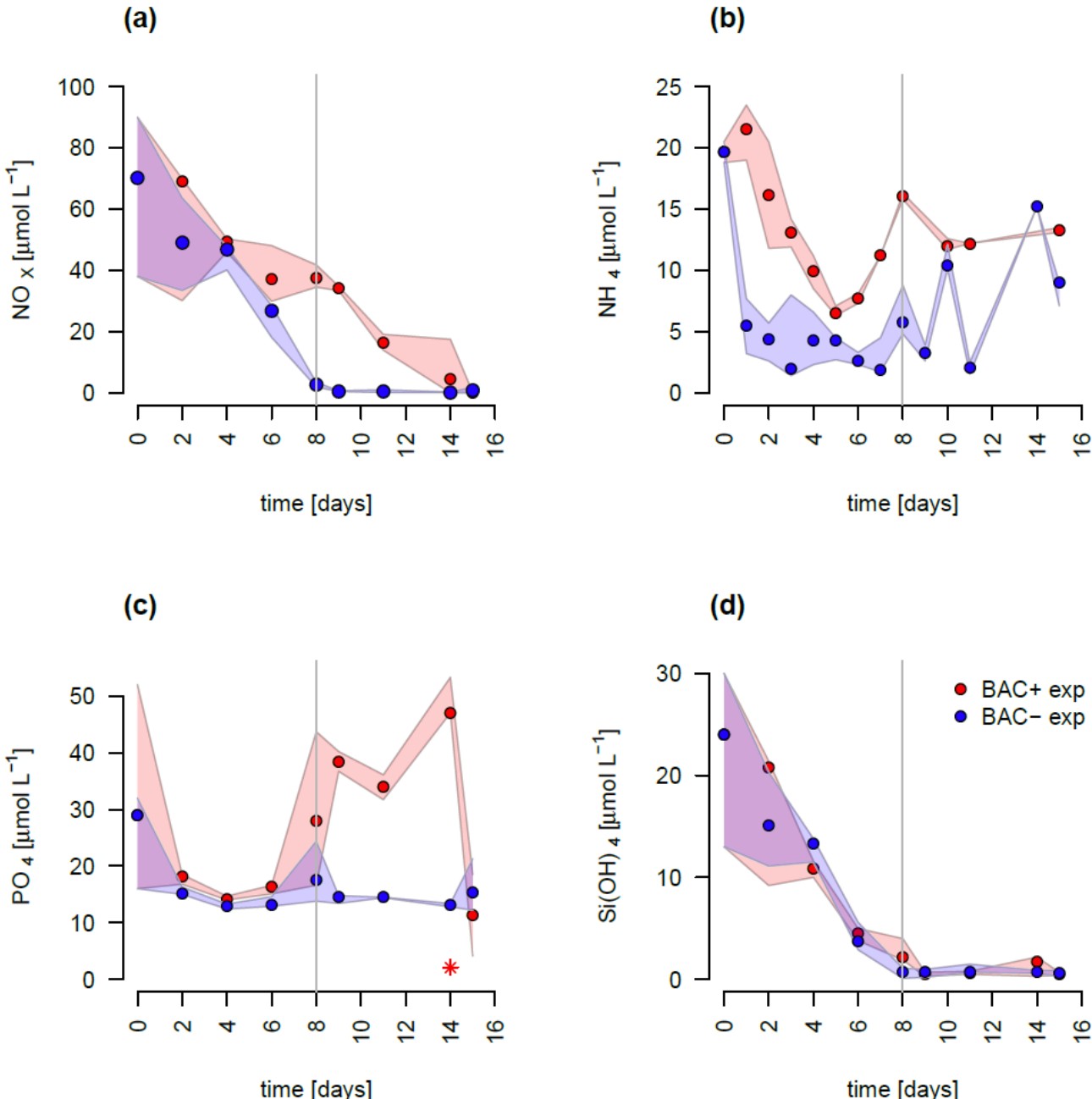

Figure 2. Nutrient measurements over the experimental incubations of a) $NO_x$, ($NO_3^- + NO_2^-$) b) $NH_4^+$, c) $PO_4^{2-}$ (The asterisk indicates a presumed measurement error), d) Silicate, blue circles are BAC-cultures and red symbols are BAC+ cultures. Circles show median values (blue = BAC-, red = BAC+) and the colored polygons show maximum and minimum of measured data (n=1-3, Table S2). The grey line shows the beginning of the stationary growth phase of *Chaetoceros socialis*.



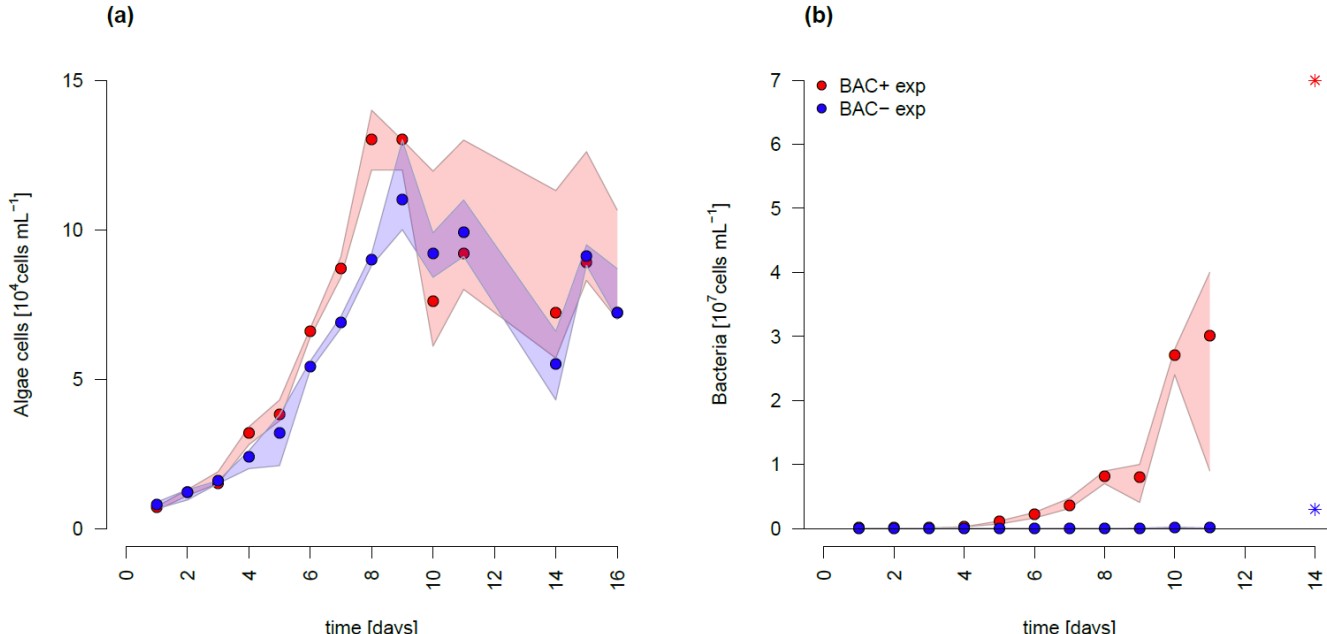

Figure 3. Abundances of a) *Chaetoceros socialis* and b) bacteria over the 14 day experimental period. Blue data are from BAC- cultures and red from BAC+ cultures. The asterisks at day 14 indicate potential outliers as based on only one replicate. Circles represent median values (blue= BAC-, red = BAC+) and the colored polygons show maximum and minimum of measured data (n=1-3, Table S2, Not visible for bacteria counts in BAC- cultures due to very small range). The maximum values of the BAC+ experiment includes algae cells in the biofilm (after day 9).

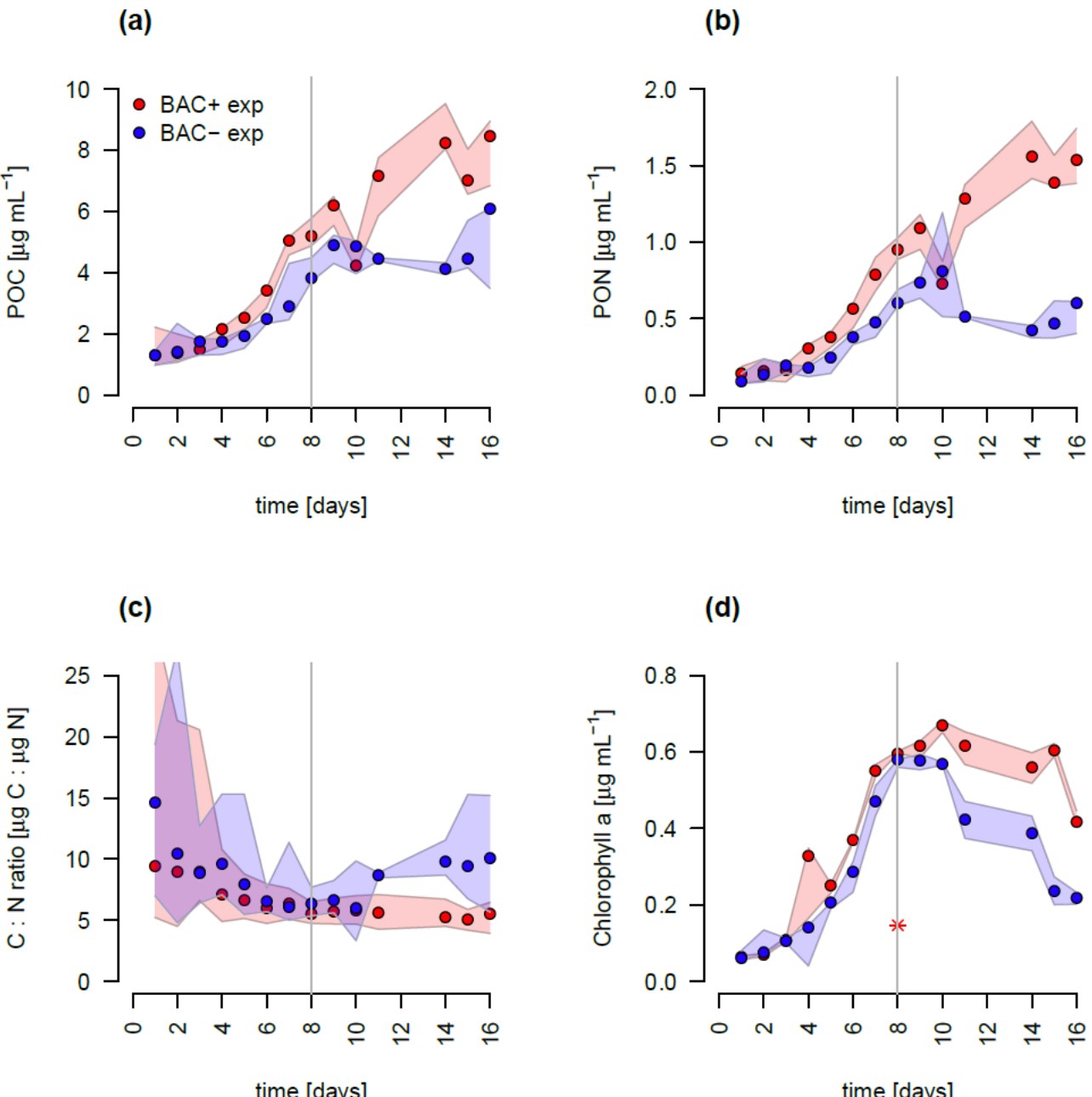

Figure 4. Total particulate organic a) Carbon (POC) b) Nitrogen (PON), c) C : N ratios, and d) Chlorophyll a concentration in experimental cultures (The asterisk indicates a presumed measurement error). Blue symbols are BAC- cultures and red show BAC+ cultures. Circles show median values (blue = BAC-, red = BAC+) and the colored polygons show the maximum and minimum of measured data (n=2-3, Table S2). The grey line indicates the start of the stationary phase.

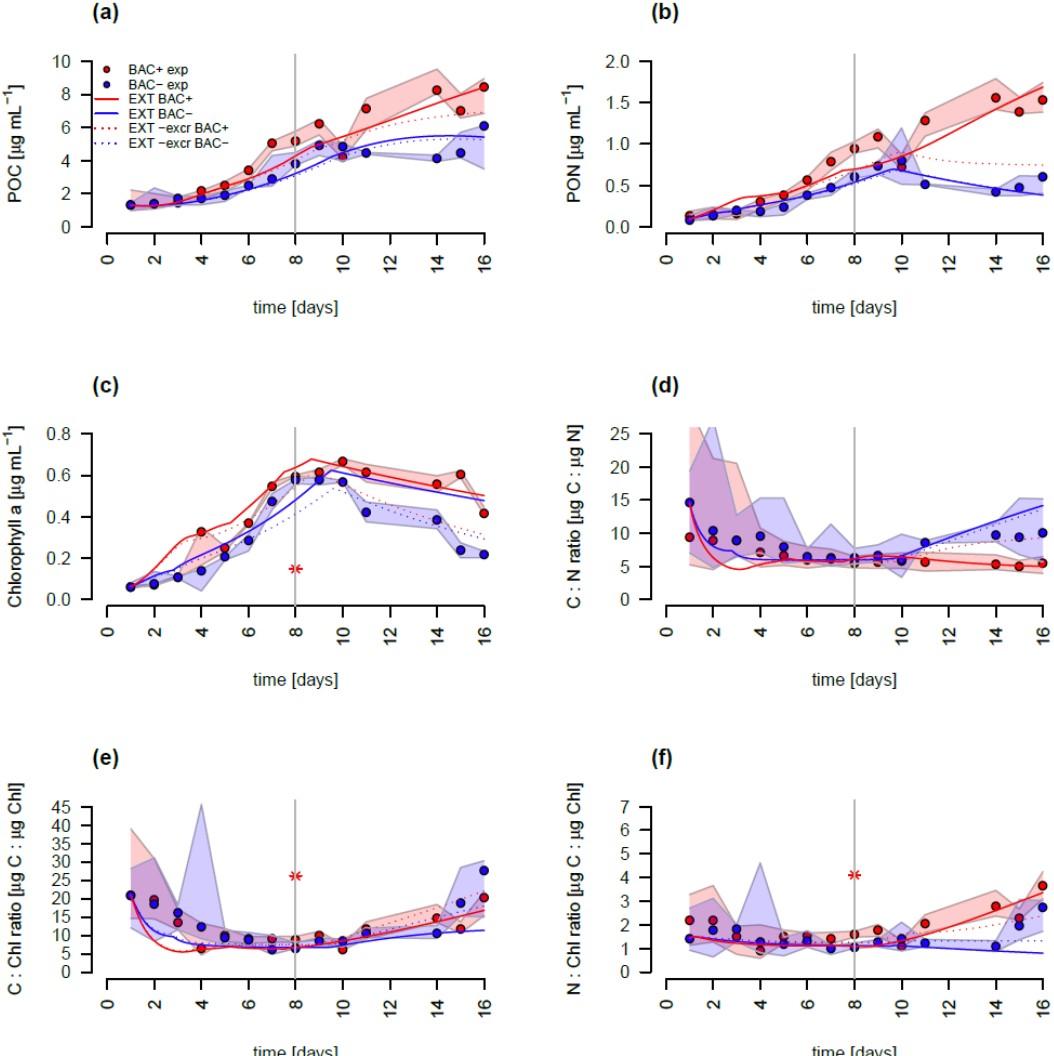

Figure 5. Model fit of the EXT model to the BAC- (blue) and BAC+ (red) experiment. Circles show median values and the colored polygons show the maximum and minimum of measured data (n=2-3, Table S2). Solid lines show the model outputs of a) POC, b) PON, c) Chl (The asterisk indicates a presumed measurement error) , d) C:N, e) C:Chl, and f) N:Chl. Dotted lines show the model fit without additional Carbon excretion term $x_f$. At day 8 the threshold for silicate limitation is reached leading to reduced photosynthesis (by the factor given by $Si_{PS}$) and inhibited Chl synthesis, which is visible as sharp transitions in POC and Chl.

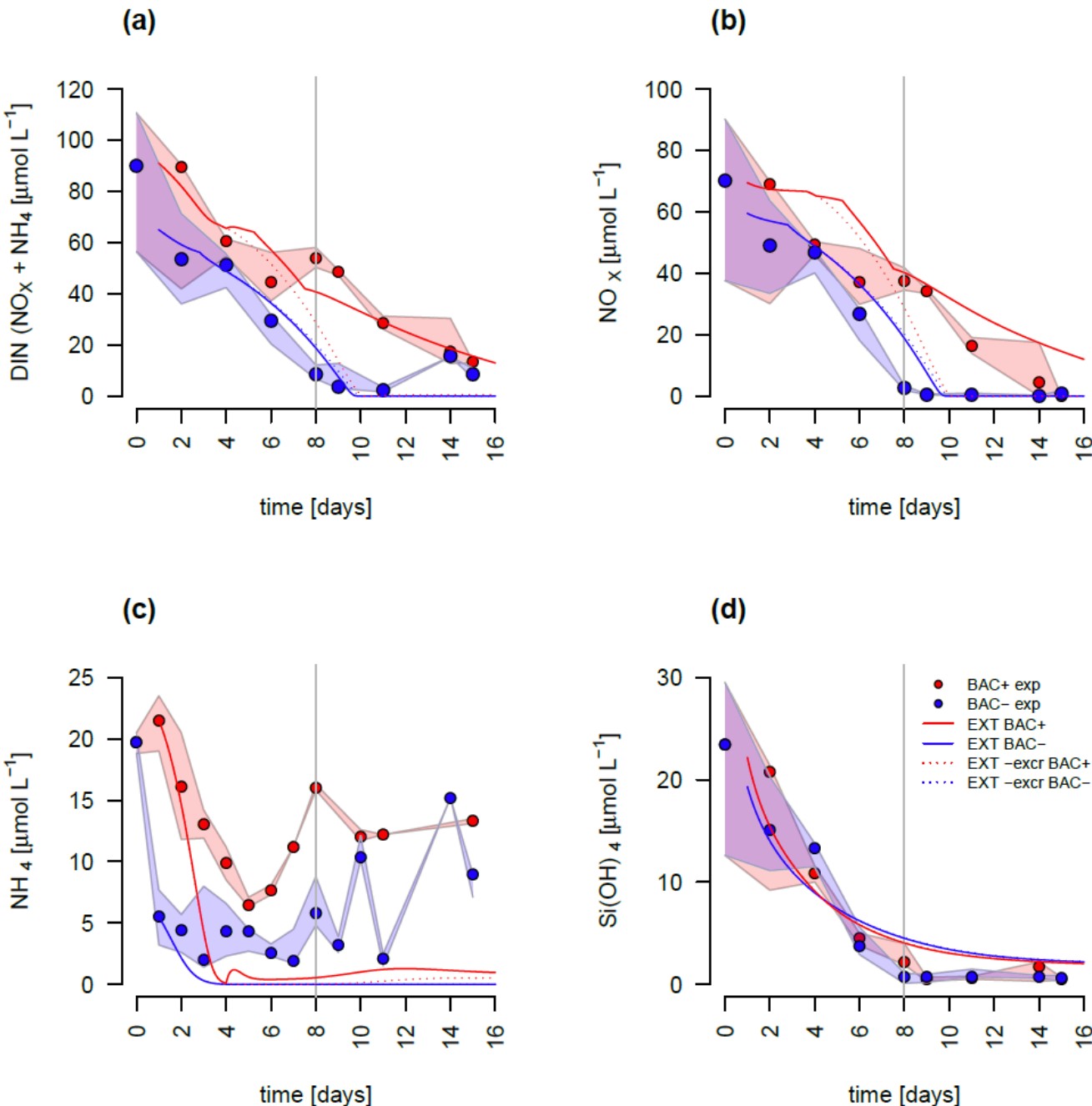

Figure 6. Model fit of the EXT model to the BAC- (blue) and BAC+ (red) experiment. Circles show median values and the colored polygons show the maximum and minimum of measured data (n=1-3, Table S2). Solid lines show the model outputs of a) DIN (NOX and NH4), b) NOx, c) NH4, and d) Si(OH)4 (All model fits overlap).

**Table**

Table 1. A comparison of major components contributing to the complexity of different models discussed. #param is the number of parameters. In case of ecosystem models (SINMOD, BFM, MEDUSA, LANL, NEMURO, NPZD only the model formulations representing the components of the current model (phytoplankton growth, remineralisation, nutrient dynamics) are considered. For the full ecosystem scale models we give the original reference to the biogeochemical compartment of the ecosystem scale models and examples for more recent versions with updated formulations of other model compartments (e.g. physical drivers). REM designates those models that include Remineralisation (Rem) marked with V is present and X is absent. Ratios shows if the stoichiometry in the model considers variable or fixed ratios of intracellular elements (C:N:Si:P:Fe). The Nutrients considered are given under Nutrients. If DIN is considered as both $NH_4$ and $NO_3$, N is shown as $N^2$. MEDUSA has Fe dependent Si:N ratios, which makes them fixed in the Arctic (fixed*).

| Model | Reference | #param | Rem | ratios | Nutrients |
|-------|-----------|--------|-----|--------|-----------|
| **Culture scale** | | | | | |
| EXT | This study | 21[*1] | V | variable | $N^2$, Si |
| G98 | Geider et al., 1998 | 10[*2] | X | variable | N |
| ANIM | Flynn, 1997 | 30 | V | variable | $N^2$ |
| SHANIM | Flynn and Fasham, 1997 | 23 | X | variable | $N^2$ |
| Flynn01 | Flynn, 2001 | 54 | X | variable | $N^2$, Si, P, Fe |
| Flynn18 | Flynn et al., 2018 | 27 | X | variable | N |
| **Ecosystem  scale** | | | | | |
| BFM | Vichi et al., 2007 | 54 | V | variable | $N^2$, Si, P, Fe |
| BFM17 | Smith et al., 2020 | 24 | V | variable | $N^2$, P |
| REcoM-2 | Hauck et al., 2013 Schourup-Kristensen et al. 2018 | 28 | X | variable | N, Si, Fe |
| MEDUSA | Yool and Popova, 2011 Henson et al., 2018 | 21 | V | fixed* | N, Si, Fe |
| LANL | Moore et al., 2004 | 15 | V | fixed | $N^2$, Si, P, Fe |
| NEMURO | Kishi et al., 2007 Amju et al., 2020 | 21 | V | fixed | $N^2$, Si |
| NPZD | Gruber et al., 2006 | 9 | V | fixed | $N^2$ |
| SINMOD | Wassmann et al., 2006 Alver et al., 2016 | 12 | X | fixed | $N^2$, Si |

Degrees of freedom after constraints by the measured data are [*1]14 and [*2]6

**Appendix**

**Tables**

Table A1. State variables of the G98 model and the EXT model (marked with V if present and X if absent) with units and designation if these state variables had been measured in the experiment.

| variable | Description | G98 | EXT | Measured | Unit |
|---|---|---|---|---|---|
| DIN | Dissolved inorganic nitrogen | V | V | V | $mgN\ m^{-3}$ |
| C | Particulate organic carbon | V | V | V | $mgC\ L^{-1}$ |
| N | Particulate Nitrogen | V | V | V | $mgN\ L^{-1}$ |
| Chl | Chlorophyll a | V | V | V | $mgChl\ L^{-1}$ |
| $Si_d$ | Dissolved Silicate | X | V | X | $\mu mol\ L^{-1}$ |
| $Si_p$ | Particulate/biogenic Silicon | X | V | V | $mgSi\ L^{-1}$ |
| Bact | Bacteria cells | X | V | V | $mgC_{bac}\ L^{-1}$ |
| DOC | Dissolved organic carbon | X | V | V | $mgC\ L^{-1}$ |
| DONr | refractory dissolved organic nitrogen | X | V | X | $mgN\ L^{-1}$ |
| DONl | labile dissolved organic nitrogen | X | V | X | $mgN\ L^{-1}$ |
| NH4 | Ammonium | X | V | V | $\mu mol\ L^{-1}$ |
| NO3 | Nitrate | X | V | V | $\mu mol\ L^{-1}$ |
| Q | Particulate N : C ratio | X | V | X | $gN\ gC^{-1}$ |
| $\theta^C$ | Chl to POC ratio | X | V | X | $gChl\ gC^{-1}$ |
| $\theta^N$ | Chl : phytoplankton nitrogen ratio | X | V | X | $gChl\ gN^{-1}$ |



Table A2. Parameters of the original G98 model and the model extension with associated units.

| parameter | | Unit |
|---|---|---|
| | G98 | |
| $\zeta$ | cost of biosynthesis | gC gN$^{-1}$ |
| $R^C$ | The carbon-based maintenance metabolic rate | d$^{-1}$ |
| $\theta^N_{max}$ | Maximum value of Chl:N ratio | gChl gN$^{-1}$ |
| $Q_{min}$ | Min. N:C ratio | gN gC$^{-1}$ |
| $Q_{max}$ | Max. N:C ratio | gN gC$^{-1}$ |
| $\alpha^{Chl}$ | Chl-specific initial C assimilation rate | gC m$^2$ (gChl μmol photons)$^{-1}$ |
| I | Incident scalar irradiance | μmol photons s$^{-1}$ m$^{-2}$ |
| n | Shape factor for $V^N_{max}$ max photosynthesis | - |
| $K_{no3}$ | Half saturation constant for nitrate uptake | μmol L$^{-1}$ |
| $P^C_{ref}$ | Value of max C specific rate of photosynthesis' | d$^{-1}$ |
| | Extension | |
| $x_f$ | Carbon excretion fraction | - |
| $K_{si}$ | Half saturation constant for Si uptake | μmol L$^{-1}$ |
| $V_{max}$ | maximum Si uptake rate | μmol Si d$^{-1}$ mg C$^{-1}$ |
| $s_{min}$ | minimum Si required for uptake | μmol L$^{-1}$ |
| rem | remineralisation rate of excreted don | mgC$_{bac}^{-1}$ d$^{-1}$ |
| $rem_d$ | remineralisation rate of refractory don | mgC$_{bac}^{-1}$ d$^{-1}$ |
| $\mu_{bact}$ | bacteria growth rate | mgC$_{bac}$ L$^{-1}$ d$^{-1}$ |
| $bact_{max}$ | Carrying capacity for bacteria | mgC$_{bac}$ L$^{-1}$ |
| $K_{nh4}$ | Half saturation constant for ammonium uptake | μmol L$^{-1}$ |
| $nh4_{thres}$ | threshold concentration for ammonium uptake | μmol L$^{-1}$ |
| $Si_{PS}$ | Fraction of photosynthesis possible after Si lim. | - |
| | Constants | |
| RR | Redfield ratio | molC molN$^{-1}$ |
| $M_N$ | Molar mass of nitrogen | g mol$^{-1}$ |

Table A3. Parameters and constants of the original G98 model and the EXT model with initial values used in the model and the lower and upper value constraints used for model fitting, unless the parameter was already defined by the data (measured). The constraints are either based on G98 fits to other diatom species, to present experimental data, or to typical values found in the literature.

| parameter | value | lower | upper | constrained by |
|---|---|---|---|---|
| **G98** | | | | |
| $\zeta$ | 1 | 1 | 2 | G98 |
| $R^C$ | 0.096 | 0.01 | 0.1 | G98 |
| $\theta^N_{max}$ | 1.7 | measured | | Data |
| $Q_{min}$ | 0.05 | measured | | Data |
| $Q_{max}$ | 0.3 | measured | | Data |
| $\alpha^{Chl}$ | 0.049 | 0.075 | 1 | G98 |
| $I$ | 100 | measured | | Data |
| $n$ | 2.572 | 1 | 4 | G98 |
| $K_{no3}$ | 1 | 1 | 10 | G98 |
| $P^C_{ref}$ | 0.8 | 0.5 | 3.5 | G98 |
| **Extension** | | | | |
| $x_f$ | 0.0546 | 0.01 | 0.3 | Schartau et al., 2017 |
| $K_{si}$ | 7.6 | 0.5 | 8 | Werner 1978 |
| $V_{max}$ | 0.1 | 0.05 | 0.9 | Data |
| $s_{min}$ | 1.82 | 1.5 | 6 | Werner 1978 |
| rem | 5.391 | 0.1 | 10 | open (rem > $rem_d$) |
| $rem_d$ | 0.00066 | 0.0001 | 0.1 | open ($rem_d$ < rem) |
| $\mu_{bact}$ | 0.8 | 0.01 | 0.8 | Data |
| $bact_{max}$ | 0.7 | 0.1 | 1 | Data |
| $K_{nh4}$ | 4.537 | 0.5 | 9.3 | Eppley 1969 |
| $nh4_{thres}$ | 0.486 | 0.1 | 10 | open |
| $Si_{PS}$ | 0.6 | 0.01 | 0.7 | Werner 1978 |
| **Constants** | | | | |
| RR | 6.625 | constant | | Redfield 1934 |
| $M_N$ | 14 | constant | | Periodic table |

Table A4. Output of the sensitivity analysis (senFun of the FME package in R, EXT fit on BACT+) with the value for each parameter and different sensitivity indices obtained after quantifying the effects of small perturbations of the parameters on the output variables (POC, PON, Chl, DIN). The L1 and L2 norms are normalized sensitivity indices defined as $L1 = \sum \frac{|S_{i,j}|}{n}$ and $L2 = \sqrt{\frac{S_{i,j}^2}{n}}$ with $S_{i,j}$ being the the

sensitivity of parameter i for model output j.

| par | value | L1 | L2 |
|---|---|---|---|
| G98 | | | |
| $\zeta$ | 1.00 | 0.18 | 0.30 |
| $R^C$ | 0.096 | 0.60 | 0.80 |
| $\theta^N_{max}$ | 1.70 | 0.37 | 0.52 |
| $Q_{min}$ | 0.05 | 0.05 | 0.06 |
| $Q_{max}$ | 0.30 | 0.36 | 0.62 |
| $\alpha^{Chl}$ | 0.05 | 0.27 | 0.43 |
| I | 100 | 0.27 | 0.43 |
| n | 2.572 | 0.53 | 0.96 |
| $K_{no3}$ | 1.00 | 0.007 | 0.015 |
| $P^C_{ref}$ | 0.80 | 1.27 | 2.05 |
| EXT | | | |
| $x_f$ | 0.0546 | 0.17 | 0.24 |
| $K_{si}$ | 7.6 | 0.00 | 0.00 |
| $V_{max}$ | 0.1 | 0.00 | 0.00 |
| $S_{min}$ | 1.82 | 0.00 | 0.00 |
| rem | 5.391 | 0.005 | 0.007 |
| $rem_d$ | 0.00065 | 0.09 | 0.21 |
| $\mu_{bact}$ | 0.8 | 0.18 | 0.36 |
| $bact_{max}$ | 0.7 | 0.04 | 0.117 |
| $K_{nh4}$ | 4.5372 | 0.06 | 0.08 |
| $nh4_{thres}$ | 0.4863 | 0.00 | 0.00 |
| $Si_{PS}$ | 0.6 | 0.07 | 0.18 |


Table A5. Other parameters calculated and used in the model equations

| parameter | Description | Unit |
|---|---|---|
| $P^C_{phot}$ | C-specific rate of photosynthesis | $d^{-1}$ |
| $P^C_{max}$ | Maximum value of $P^C_{phot}$ at temperature T | $d^{-1}$ |
| $R^{Chl}$ | Chl degradation rate constant | $d^{-1}$ |
| $R^N$ | N remineralization rate constant | $d^{-1}$ |
| $V^C_{nit}$ | Diatom carbon specific nitrate uptake rate | gN (gC d)$^{-1}$ |
| $V^C_{ref}$ | Value of $V^C_{max}$ at temperature T | gN (gC d)$^{-1}$ |
| $p_{Chl}$ | Chl synthesis regulation term | - |
| $\mu$ | specific growth rate of algae | cells d$^{-1}$ |






Table A6. Model equations from G98 (Geider et al., 1998) corrected for typographical errors by Ross and Geider (2009) with extensions.

| 1) | Carbon synthesis | $\dfrac{dC}{dt} = \left(P^C - \zeta V_N^C - R^C\right)C = \mu C$ |
|---|---|---|
| | (C originates from unmodelled excess pool of DIC) | |
| 2) | Chl synthesis | $\dfrac{dChl}{dt} = \left(\dfrac{\rho_{Chl}V_N^C}{\Theta^C} - R^{Chl}\right)Chl$ |
| 3) | Nitrogen uptake | $\dfrac{dN}{dt} = \left(\dfrac{V_N^C}{Q} - R^N\right)N$ |
| 4) | from Eq. (1) and (2) | $\dfrac{dQ}{dt} = V_N^C - \mu Q$ |
| 5) | from Eq. (1) and (2) | $\dfrac{d\Theta^C}{dt} = V_N^C \rho_{Chl} - \Theta^C \mu$ |
| 6) | Photosynthesis | $P^C = P_{max}^C \left[1 - exp\left(-\dfrac{I}{I_K}\right)\right]$ |
| 7) | Max. N uptake | $V_N^C = V_{ref}^C \left[\dfrac{Q_{max} - Q}{Q_{max} - Q_{min}}\right]\dfrac{DIN}{DIN + K_{no3}}$ |
| 8) | with | $\rho^{Chl} = \Theta_{max}^N \left[1 - exp\left(-\dfrac{I}{I_K}\right)\right]$ |
| 9) | | $V_{ref}^C = P_{ref}^C Q_{max}$ |
| 10) | | $P_{max}^C = P_{ref}^C \dfrac{Q - Q_{min}}{Q_{max} - Q_{min}}$ |
| 11) | | $I_K = \dfrac{P_{max}^C}{\alpha^{Chl}\Theta^C}$ |

Table A7. Model equations of the EXT model based on G98

| 1a) | Carbon synthesis | $IF\ (Si_d < 2\ s_{min})$ |
|---|---|---|

*(Reduced C synthesis under Si limitation after Werner 1978)*

$$Si_{PS} = Si_{PS}$$

$$ELSE$$

$$Si_{PS} = 1$$

1b)

$$\frac{dC}{dt} = Si_{PS}\left(P^C - \zeta V_N^C - R^C - xf\right)C = \mu C$$

2)  Chl synthesis

$$IF\ (Si_d < 2\ s_{min})$$

*(Chl synthesis stops under Si limitation after Werner 1978)*

$$\frac{dChl}{dt} = -R_{Chl}\ Chl$$

$$ELSE$$

$$\frac{dChl}{dt} = \left(\frac{\rho_{Chl}V_N^C}{\Theta^C} - R_{Chl}\right)Chl$$

3)  from Eq. (1 & 2)

$$\frac{dQ}{dt} = V_N^C - \mu Q$$

4)  from Eq. (1 & 2)

$$\frac{d\Theta^C}{dt} = V_N^C \rho_{Chl} - \Theta^C \mu$$

5)  Nitrogen uptake

$$\frac{dN}{dt} = \left(\frac{V_N^C}{Q} - R^N - xf\right)N$$

6)  Bacteria biomass production

$$\frac{dBact}{dt} = Bact\ \mu_{Bact}\left(1 - \frac{Bact}{Bact_{max}}\right)$$

*(Logistic growth)*

7a)  Silicate uptake

$$\frac{dSi_d}{dt} = V_S^C = \left(V_{max}Si_d\ \frac{Si_d - S_{min}}{K_{si}\ S_{min}}\right)C$$

*(Monod kinetics after Spilling et al., 2010)*

7b)
$$\frac{dSi_p}{dt} = -\frac{dSi_d}{dt} \quad 14$$

8) Ammonium uptake and
production

*(Threshhold after
Tezuka 1989, and Gilpin
2004)*

$$IF \left(\frac{C}{N} < 10\right)$$

$$\frac{dNH4}{dt} = \frac{-\left(\frac{V_{NH4}^C}{Q}\right)N + Bact\ DONl\ rem + Bact\ DONr\ rem_d)}{M_N\ 10^3}$$

$$ELSE\ IF\ (NH4 > nh4_{thresj}$$

$$\frac{dNH4}{dt} = \frac{-\left(\frac{V_{NH4}^C}{Q}\right)N - \frac{dBact}{dt\ RR})}{M_N\ 10^3}$$

$$ELSE$$

$$\frac{dNH4}{dt} = \frac{-\left(\frac{V_{NH4}^C}{Q}\right)N)}{M_N\ 10^3}$$

9) DON uptake and
production

$$IF \left(\frac{C}{N} < 10\right)$$

$$\frac{dDONl}{dt} = \frac{-Bact\ DONl\ rem + xf\ N\ - \frac{dBact}{dt\ RR}}{M_N\ 10^3}$$

$$\frac{dDONr}{dt} = \frac{-\ Bact\ DONr\ rem_d}{M_N\ \ 10^3}$$

$$ELSE$$

$$\frac{dDONl}{dt} = \frac{xf\ N}{M_N\ 10^3}$$

$$\frac{dDONr}{dt} = 0$$

| 10) | DIN uptake | $IF\ (NH4 > nh4_{thresh})$ |
|---|---|---|

$$\frac{dDIN}{dt} = \frac{-\left(\frac{V^C_{NO3}}{Q}\right)N}{M_N\ 10^3}$$

$$ELSE$$

$$\frac{dDIN}{dt} = \frac{-0.2\left(\frac{V^C_{NO3}}{Q}\right)N - \frac{dBact}{dt\ RR}}{M_N\ 10^3}$$

| 11) | Photosynthesis | $P^C = P^C_{max}\left[1 - exp\left(-\frac{I}{I_K}\right)\right]$ |
|---|---|---|

| 12a) | Max NO3 uptake | $V^C_{NO3} = V^C_{ref}\left[\frac{Q_{max} - Q}{Q_{max} - Q_{min}}\right]\frac{NO3}{NO3 + K_{no3}}$ |
|---|---|---|

| 12b) | Max NH4 uptake | $V^C_{NH4} = (0.01\ Q)\ 0.0021\ \frac{NH4}{NH4 + K_{nh4}}$ |
|---|---|---|
| | *(based on SHANIM Eq4 by Flynn and Fasham, 1997)* | |

| 13) | Max N uptake | $IF\ (NH4 > nh4_{thresh})$ |
|---|---|---|
| | *(Based on Flynn and Fasham, 1997 and Flynn, 1999 showing no total inhibition in cold water)* | $V^C_N = V^C_{NH4} + 0.2\ V^C_{NO3}$ |

$$ELSE$$

$$V^C_N = V^C_{NH4} + V^C_{NO3}$$

| 14) | with | $\rho^{Chl} = \Theta^N_{max}\left[1 - exp\left(-\frac{I}{I_K}\right)\right]$ |
|---|---|---|

| 15) | | $V^C_{ref} = P^C_{ref}Q_{max}$ |
|---|---|---|

| 16) | | $P^C_{max} = P^C_{ref}\frac{Q - Q_{min}}{Q_{max} - Q_{min}}$ |
|---|---|---|

| 17) | | $I_K = \frac{P^C_{max}}{\alpha^{Chl}\Theta^C}$ |
|---|---|---|

| | | |
|---|---|---|
| 18) | DOC to DONr conversion | $$DONr = \frac{DOC}{RR}$$ |





Table A8. Output of the collinearity or parameter identifiability analysis using the collin function (G98 fit on BACT- data) of the FME R package (Soetart et al., 2010b). A subset of any combinations of two parameter with a collinearity above 20, indicating non-identifiable parameter combinations is given (Brun et al., 2001). Parameter combinations tested are marked with a V in the left part of the table, the collinearity output is given on the right site.

| | | | Parameter combinations | | | | | | | |
| $\zeta$ | $R^C$ | $\theta^N_{max}$ | $Q_{min}$ | $Q_{max}$ | $\alpha^{Chl}$ | $I$ | $n$ | $K_{no3}$ | $P^C_{ref}$ | collinearity |
|---|---|---|---|---|---|---|---|---|---|---|
| V | X | V | X | X | X | X | X | X | X | 27 |
| V | X | X | X | V | X | X | X | X | X | 98 |
| V | X | X | X | X | V | X | X | X | X | 31 |
| V | X | X | X | X | X | V | X | X | X | 31 |
| V | X | X | X | X | X | X | V | X | X | 93 |
| X | V | X | X | X | X | X | X | X | V | 25 |
| X | X | V | X | V | X | X | X | X | X | 34 |
| X | X | V | X | X | V | X | X | X | X | 101 |
| X | X | V | X | X | X | V | X | X | X | 101 |
| X | X | V | X | X | X | X | V | X | X | 38 |
| X | X | V | X | X | X | X | X | X | V | 32 |
| X | X | X | X | V | V | X | X | X | X | 40 |
| X | X | X | X | V | X | V | X | X | X | 40 |
| X | X | X | X | V | X | X | V | X | X | 146 |
| X | X | X | X | X | V | V | X | X | X | 455473 |
| X | X | X | X | X | V | X | V | X | X | 46 |
| X | X | X | X | X | V | X | X | X | V | 28 |
| X | X | X | X | X | X | V | V | X | X | 46 |
| X | X | X | X | X | X | V | X | X | V | 28 |


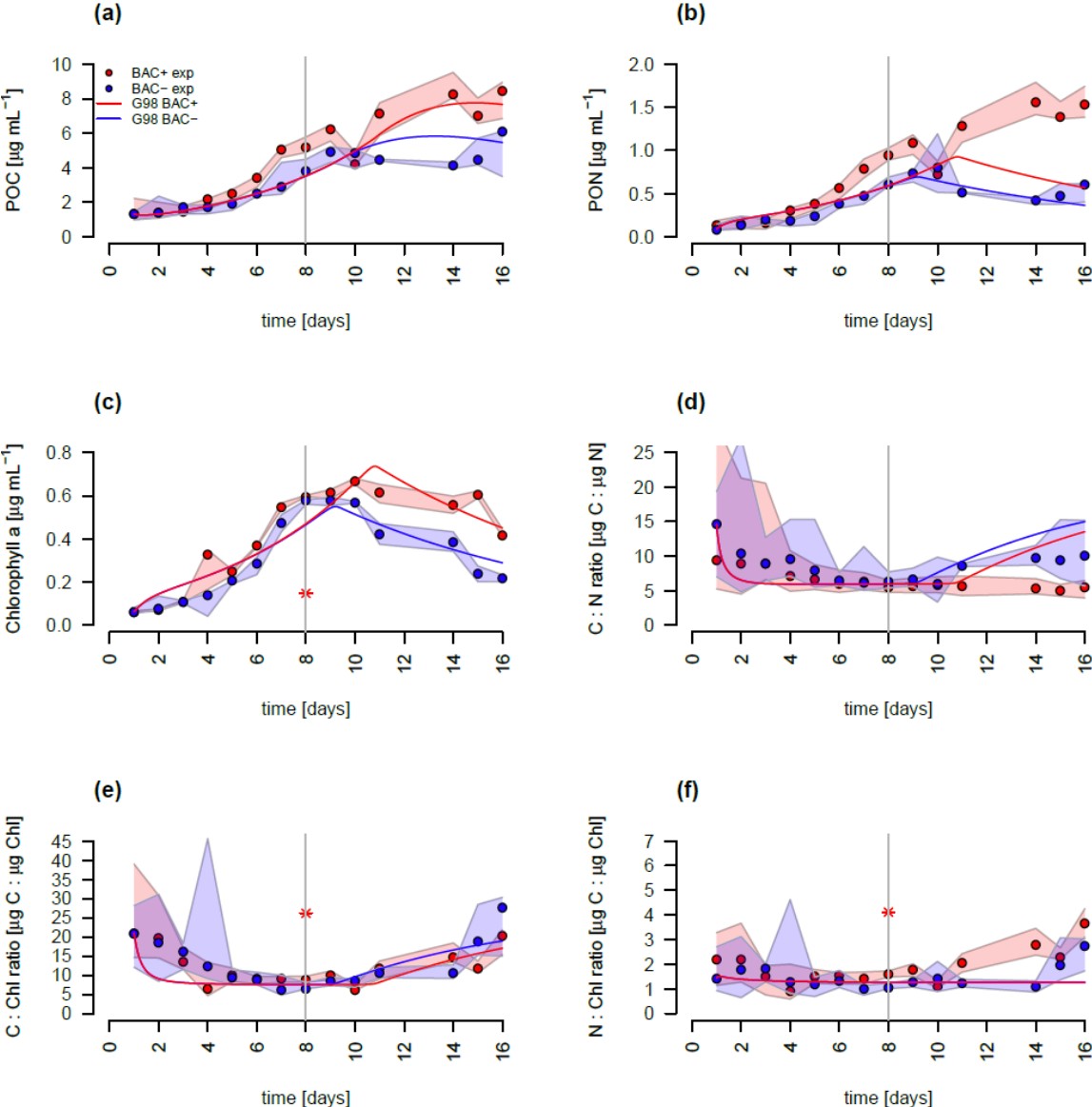

Figure B1: Model fit of the G98 model to the BAC- (blue) and BAC+ (red) experiment. Circles show median values and the colored polygons show the minimum and maximum of the measured data (n=2-3, Table S2). Solid lines show the model outputs of a) POC, b) PON, c) Chl (The asterisk indicates a presumed measurement error), d) C:N, e) C:Chl, and f) N:Chl.

**Equations**

Equation C1. F-ratio estimation in the cultivation experiments with the average PON concentrations at day 13 to 15 (PON$^{d13-15}$) for the BAC- and BAC+ treatments.

$$f-ratio = \frac{PON_{BAC-}^{d13-15}}{PON_{BAC-}^{d13-15} + PON_{BAC+}^{d13-15}}$$


Equation C2. normalized RMSE with i being the different variables (POC, PON, Chl, DIN), and j the different values of each state variable. Predicted values are given as P and observed values as O.

$$RMSE = \sqrt{\sum_{\substack{i=1 \\ j=1}}^{n,p} \frac{(P_{i,j} - O_{i,j})^2}{Var(O_i)}}$$





