# Peer review of "Modelling Silicate – Nitrate - Ammonium co-limitation of algal growth and the importance of bacterial remineralisation based on an experimental Arctic coastal spring bloom culture study"

_Biogeosciences, 2020_

## Referee Comment (RC1) · Anonymous Referee #1 · 23 Sep 2020

This study investigates the role of bacteria in nutrient remineralization during the spring bloom in an Arctic coastal ecosystem. The work combines a cultivation experiment with a simplified dynamic model to represent processes of remineralization, silicate limitation, and nitrogen limitation. The purpose of the model was to "describe the response in photosynthesis, chlorophyll synthesis and nitrogen assimilation with a minimal number of parameters."

Chaetoceros socialis may not be representative of the the most important diatom species across all of the Arctic coastal areas. How representative do you expect that it

is?

line 173: define EPS here and not on line 342

line 191: sentence needs a .

line 228: change extend to extent

line 231: units for growth rate?

line 255: two-thirds

line 257: allowed an estimate of

line 274: change was to were

line 285: parameters

line 307: delete it

line 320: in order

line 325: the time scale

line 403: export is decreasing overall

Figs. 4 and 5: the legend is missing entries for the solid red line and the dashed red line

Fig. B3: the legend is missing an entry for the red line

---

## Author Comment (AC1) · 6 Oct 2020

We want to thank reviewer 1 for the constructive feedback. All specific comments are changed in the manuscript as suggested by the reviewer. A more detailed response regarding the representativeness of Chaetoceros socialis for Arctic coastal systems is given below.

Reviewers comment: Chaetoceros socialis may not be representative of the the most important diatom species across all of the Arctic coastal areas. How representative do

you expect that it is?

We acknowledge that many different species contribute to the bloom formations in the Arctic coastal areas, including pennate sea ice algae and several pelagic centric diatoms (see below) Chaeotoceros socialis may not be the most dominant species in all coastal Arctic spring blooms, however it has been reported as dominating blooms in several areas (see below). We thus consider C. socialis to be overall a representative model organism.

Chaetoceros socialis is a widely occurring marine diatom species that has been observed from Arctic seas into warmer oceans like the Gulf of California (Hasle and Syvertsen 1997) that differ physiologically and morphologically (Degerlund et al. 2012, Huseby et al. 2012). Current research indicates several cryptic species to be within the C. socialis complex (Gaonkar et al. 2017, De Luca et al. 2019). It is frequently used in culture based experiments to evaluate for example the role of ocean acification (Li et al. 2017), and DMS (Baumann et al. 1994) and lipid production (Artamonova et al. 2017).

In Arctic waters, it has been observed as bloom forming species across the Arctic with for example bloom occurring in the North Water Polynya between July and September (Booth et al. 2002), the Barents Sea (Rey and Skjoldal 1987, Rat'kova and Wassmann 2002) and other Arctic coastal sites, often dominating phytoplankton biomass following the blooming of Thalassiosira spp. (von Quillfeldt 2005).

Besides C.socialis, coastal Arctic spring blooms are typically dominated by other chain forming diatoms, such as Thalassiosira spp., Fragillaripsis spp., Chaetoceros spp., Navicula spp., or Skeletonema spp.. All of these pennate or centric diatoms share similar requirements for inorganic nutrients and all of these groups are typically limited by silicate and/or nitrogen limitation in coastal Arctic systems. In addition, all of these groups have similar physiological opportunities to respond to nutrient limitations, can excrete EPS and interact with bacteria. Hence, we are confident that C. socialis is suitable as model organism, representative for coastal Arctic spring blooms unless silicate is limiting from the start in which case, flagellates, such as Phaeocystis may dominate (As discussed in line 302 in our manuscript).

We added a few more details and references in the manuscript to support these statements in the following way, with changes highlighted in green:

Line 53-58: Phytoplankton blooms may be dominated by a single or a few algal species, often with a similar physiology during certain phases of the bloom (e.g. Eilertsen et al., 1989; Degerlund and Eilertsen, 2010; Iversen and Seuthe, 2011). Chain-forming diatoms, sharing physiological needs and responses to nutrient limitations (e.g. Eilertsen et al., 1989; von Quillfeldt, 2005), typically dominate these blooms. In some Arctic and sub-Arctic areas the Arctic phytoplankton chosen for this model, Chaetoceros socialis, is a dominant species during spring blooms (Rey and Skjoldal, 1987; Eilertsen et al., 1989; Booth et al., 2002; Ratkova and Wassmann, 2002; von Quillfeldt, 2005; Degerlund and Eilertsen, 2010).

Line 297-299: While C. socialis may not be the dominant species in all coastal Arctic phytoplankton blooms, we argue that it is representative for chain-forming diatoms typically dominating these systems due to their shared needs and responses to nutrient limitations (e.g. Eilertsen et al., 1989; von Quillfeldt, 2005).

References

Hasle, G.R.; Syvertsen, E.E. (1997). Marine diatoms, in: Tomas, C.R. (Ed.) Identifying marine phytoplankton. pp. 5-385In: Tomas, C.R. (Ed.) (1997). Identifying marine phytoplankton. Academic Press: San Diego. ISBN 0-12-693018-X. XV, 858 pp

Degerlund M, Huseby S, Zingone A, Sarno D, Landfald B (2012) Functional diversity in cryptic species of Chaetoceros socialis Lauder (Bacillariophyceae). Journal of Plankton Research 34:416-431

Li X, Roevros N, Dehairs F, Chou L (2017) Biological responses of the marine diatom

Chaetoceros socialis to changing environmental conditions: A laboratory experiment. PloS one 12:e0188615-e0188615

De Luca D, Kooistra WHCF, Sarno D, Gaonkar CC, Piredda R (2019) Global distribution and diversity of Chaetoceros (Bacillariophyta, Mediophyceae): integration of classical and novel strategies. PeerJ 7:e7410-e7410

Gaonkar C, Kooistra W, Lange C, Montresor M, Sarno D (2017) Two new species in the Chaetoceros socialis complex (Bacillariophyta): C. sporotruncatus and C. dichatoensis, and characterization of its relatives, C. radicans and C. cinctus. Journal of phycology 53

Huseby S, Degerlund M, Zingone A, Hansen E (2012) Metabolic fingerprinting reveals differences between northern and southern strains of the cryptic diatom Chaetoceros socialis. European Journal of Phycology 47:480-489

Baumann MEM, Brandini FP, Staubes R (1994) The influence of light and temperature on carbon-specific DMS release by cultures of Phaeocystis antarctica and three antarctic diatoms. Marine Chemistry 45:129-136

Booth BC, Larouche P, Bélanger S, Klein B, Amiel D, Mei ZP (2002) Dynamics of Chaetoceros socialis blooms in the North Water. Deep Sea Research Part II: Topical Studies in Oceanography 49:5003-5025

von Quillfeldt 2005. Common Diatom Species in Arctic Spring Blooms: Their Distribution and Abundance.

Rey F, Skjoldal HR (1987) Consumption of silicic acid below the euphotic zone by sedimenting diatom blooms in the Barents Sea. MEPS36: 307-312

Rat'kova TN, Wassmann P (2002) Seasonal variation and spatial distribution of phyto- and protozooplankton in the central Barents Sea. Journal of Marine Systems 38:47-75

---

## Referee Comment (RC2) · Anonymous Referee #2 · 27 Oct 2020

Review of: "Modelling Silicate – Nitrate - Ammonium co-limitation of algal growth and the importance of bacterial remineralisation based on an experimental Arctic coastal spring bloom culture study" by Vonnahme et al.

Summary

This manuscript presents an interesting combined laboratory and modelling study of the nutrient dynamics of a diatom species common in the Arctic. The laboratory component uses two experimental set-ups: 1. axenic cultures of the diatom species; 2.

[Figure]

cultures of the diatom species that include associated bacteria. Short incubations (∼2 weeks) of these cultures take them from exponential phase through to stationary phase, with the cultures sampled throughout to measure cell counts, nutrient concentrations, etc. After an initial period of diatom cell number growth (week 1) in both cultures, this stops as NOx and dSi concentrations approach limiting concentrations. However, NH4 is consistently higher in the non-axenic cultures, and the bacterial cell counts in these cultures increase exponentially during the latter period of the incubations (week 2). The authors interpret the presence of bacteria as being conducive to supplying the diatoms with regenerated nutrients. The modelling component uses a base model, G98, and an extended model based on this that includes a number of additional processes with relevance to the laboratory setting and the hypothesised role of bacterial remineralisation in supporting phytoplankton growth. The models are tuned to fit the laboratory data, with a manual phase to retain consistent parameter values between the models. The authors conclude with a discussion on the application of their results to the real Arctic and its expected future state.

I have listed a number of significant general comments below, followed by more specific and often minor comments. Overall, my assessment is that the manuscript requires major revision to clarify and amend the work described.

General comments

Upfront, my modelling background means that I cannot comment directly on the details of the laboratory work in the study. However, I note that the experiments conducted exhibit anomalies that are not addressed in the manuscript. In Figure 1c, phosphate in bacterial cultures exhibits a strong spike upwards at day 8 that persists and shows high variability. In Figure 3d, chlorophyll in bacterial cultures shows a marked but temporary spike downward at day 8. While the latter is likely a replication or measurement issue, the former is harder to understand, and the manuscript does not discuss its scale or variability. It would be useful to know what the authors believed happened here, particularly in the case of phosphate where bottle concentrations approximately double

against a backdrop of slowly declining phytoplankton and rising bacteria concentrations. The model may even be able to help on this point.

The manuscript's model description appears incomplete, with equations for terms such as those for dSi omitted. More generally, the manuscript would be improved by simply making clear which models are being run – while the text refers to model G98 and "the extended model", the plots shown refer instead to "model + excr" and "model - excr". What might be helpful is to have some sort of diagram of the two main models being used (G98 and Extended) to help illustrate the main connections between state variables, and make clear the differences between the two models.

The description of the model tuning needs to be clearer. It's unclear why some parameters were picked for tuning while others weren't (e.g. remineralisation parameters were not tuned), or what the rationale for picking the training data streams was (e.g. model ammonium was "loosely constrained" to observations). The text mentions several R packages used, but these are presented without any information about what they do, how they work, or what assumptions they make. For instance, is parameter space sampled by latin hypercube, genetic algorithm, or via local misfit gradient? There's also an unclear distinction made around a "manual" component of this tuning exercise.

On a point related to tuning, I noted that the model has a key parameter for restricting phytoplankton growth (by 80%) in the absence of silicate, but that this parameter is not included in the tuning, which seems something of an omission (and, on a more presentation level, is hard-wired into the equations as a number rather than a parameter).

Finally, the authors identify three central hypotheses in their study:

1. Bacterial regeneration of ammonium will extend a phytoplankton growth;

2. Silicate or nitrogen limitations have different physiological responses;

3. A simple experiment can adequately represent Arctic spring bloom dynamics.

On the first, the model has a very poor performance replicating the time history of

ammonium concentrations. On the second, this study would be more convincing if the concentrations of Si and N had been experimentally manipulated to enhance / diminish limitation of each. On the third, the model's inconsistent fit with observations, and its omission of significant real world factors (e.g. zooplankton) make it difficult to evaluate whether this is true. And because the model is only being run for the short incubation period (i.e. rather than beyond the incubation period, or in some mode investigating more realistic or extrapolated settings), it's not clear how it behaves when "unleashed".

Overall, I very much like the combined laboratory and modelling approach, but judge that the modelling component in particular needs to be made much clearer, and evaluated more critically.

Specific comments

Pg. 1, ln. 20: "neglect" or "simplify"?; the distinction is important

Pg. 1, ln. 23: surplus "and"

Pg. 1, ln. 25: regarding the importance of organic matter excretion, was this based on observational evidence?

Pg. 1, ln. 26: "model complexity is comparable to other ecosystem models" – this is misleading as the model here is really an incubator model and not an ecosystem model; it's missing most of the components that such models include (e.g. detritus, zooplankton)

Pg. 2: maybe be a little clearer on the distinction between autotrophic and heterotrophic bacteria throughout; cyanobacteria, for instance, are unlikely to play the role that's described as "bacterial" here

Pg. 2: also, you should probably say something about the role of zooplankton; they graze phytoplankton and excrete some of the nitrogen they acquire; how quantitatively important is this here?; (I've added a cite to a paper that hints that they might not be all that important)

Pg. 2, ln. 36: "marine phytoplankton *are*"?

Pg. 2, ln. 41: predictions of what?

Pg. 2, ln. 47: you might want to cite something like doi:10.1016/j.dsr.2012.10.003 as evidence of the reduced role of mesozooplankton in controlling / terminating blooms

Pg. 2, ln. 48: remineralisation of what?; a bit of clarity would be helpful; dead diatoms, TEP, faecal material, etc.?

Pg. 2, ln. 57: heterotrophic bacterioplankton?

Pg. 2, ln. 60: regarding "neglected", do you mean omitted or simplified?; most models include remineralisation of detrital material, and this implicitly bacterial

Pg. 2, ln. 63: cultivation experiments normally provide parameter values for things like maximum rates of processes, half-saturations, etc., so it's not clear this is problematic; if model tuning is using cultivation experiments at equilibrium then this might be more of an issue

Pg. 3, ln. 72: this process was well-known long before this citation (Flynn, 1997); dig a bit deeper

Pg. 3, ln. 72: "iron has a strong control on silicate uptake" - I'm not sure that this is quite right; Si:C ratios are affected by Fe availability, but this is through continuing Si uptake but reduced C/N uptake and no cell division; my understanding is that Si *uptake* (within a certain range of Si:C) is not immediately affected by Fe; also the recent source given for this statement, Hohn et al., 2009, is a modelling PhD thesis

Pg. 3, ln. 75: "ultimately too complex" - they add computational expense to large-scale ecosystem models; it's not clear that they are "too complex" (or even what is meant by this)

Pg. 3, ln. 79: is phosphate limiting in the Southern Ocean?; in parts of its northern extent, yes, but in the south its concentrations are high, no?

Pg. 3, ln. 79: "coastal"?; is there a distinction to be drawn with deep Arctic locations?

Pg. 3, ln. 81: yes and no; if simple lab experiments exclude factors such as zooplankton excretion which might help fuel phytoplankton growth in parallel with bacterial remineralisation, then it is questionable that they are demonstrating something that's important in the real ocean

Pg. 3, ln. 86: how "associated" is this?; is it something that lives in direct physical contact or shares the same waters?

Pg. 3, ln. 87: again, what specifically is the issue with complexity?; is it model cost, or is there some other aspect of complexity that disfavours inclusion in large-scale models?

Pg. 3, ln. 93: good hypotheses!; however, you do not clearly return to them (e.g. "Regarding the hypotheses framed for this study . . .")

Pg. 4: this all sounds good, but my expertise in laboratory work is very limited

Pg. 4, ln. 111: just for simplicity in the labelling, you might want to come up with nice short names for these experiments; e.g. BACT- (for the axenic) and BACT+ (for the non-axenic), or similar; this will make it easier to refer to the experiments in clear, non-wordy ways later on

Pg. 5, ln. 143: the origin of the f-ratio should be cited so that less familiar readers can understand what it is

Pg. 5, ln. 144-146: this is a little confusing; perhaps spell it out with equations?

Pg. 5, section 2.2: I don't think it ever hurts to have a schematic of a model's dynamics to supplement equations and (especially) verbal description

Pg. 5, section 2.2: similarly, this section would be a lot clearer if you spelled out which models you were using, and ensured that the later plots use the same nomenclature; I initially misread the work ; I reckon it's: 1. G98; 2. Extended; 3. G98 – excretion; 4. Extended – excretion

Pg. 5, section 2.2: stating up front an outline about the modelling strategy might help (i.e. two models, tuned to the lab work, DOM addition, etc.)

Pg. 5, ln. 149: some model equations by the looks of things; the model description appears incomplete

Pg. 6, ln. 164: equation for dSi seems missing in appendix

Pg. 6, ln. 166: "80% reduction" - is this where the 0.2 in the equations comes from (i.e. 1 - 0.8 = 0.2)?

Pg. 6, ln. 167: some syntheses would suggest that N dynamics *are* coupled to Si dynamics: e.g. Martin-Jezequel, V., M. Hildebrand, and M. A. Brzezinski, Silicon metabolism in diatoms: Implications for growth, J. Phycol., 36, 821 – 840, 2000.

Pg. 6, ln. 171: make it clear here that your model has labile and refractory DON

Pg. 6, ln. 173: it seems unlikely that the bacteria would simply "give up" on remineralisation if the C:N ratio is too high; perhaps expand on why Tezuka suggests this is happening

Pg. 6, ln. 175: this is unclear; when you say "substrate" what do you mean?; typically substrate is used to indicate a resource consumed by an organism; here you're talking about phytoplankton, so DIN and DIC would appear to be meant - but DIC will likely be much higher than 10x DIN

Pg. 6, ln. 177: does this mean that bacteria won't remineralise material with a C:N > 10?; that seems a little unlikely

Pg. 6, ln. 178: as the paper makes a fuss earlier about other models glossing over bacterial remineralisation, this simplified form is surprising

Pg. 6, ln. 185: Table A6 - it looks to me like some equations are missing

Pg. 6, ln. 186: which order of RK?; e.g. 3 or 4 (or higher)

Pg. 6, ln. 185-191: this description of model tuning is far too brief; I'm not sure what's going on here; readers unfamiliar with R (I am one) will not understand what these different packages are doing or what their underlying assumptions are; this aspect of the modelling is too important to be glossed over so quickly; in general, to avoid the appearance of having just used the first package that occurred to you, expanding on the detail of the tuning (tools, approach, goal) would greatly benefit this description (hence and on Pg. 7)

Pg. 6, ln. 192: the text should be clear on which observed variables were used to fit the model, why these were favoured, and whether any weighting was made to account for those judged better observed or more important; I would naively expect nutrient concentrations to be of prime importance but it's unclear what criteria the authors are using here (see my later remark about ammonium)

Pg. 7, ln. 197: this seems rather unsatisfactory; I would expect parallel runs with the same parameter values to be performed for axenic and not-axenic simulations, with an automated process (e.g. a genetic algorithm) to evaluate cost (i.e. misfit) before somehow generating new parameter values and iterating; having a manual component seems odd

Pg. 7, ln. 198: what are these "known limitations"?; also, it's noticeable in the plots that the model solutions inflect strongly around the lag/stationary phase time point - is the model somehow different either side of this division?

Pg. 7, ln. 202: "Colinearlity" - do you mean that you're looking for linkages between parameters here?

Pg. 7, ln. 205: ammonium was "constrained loosely" - perhaps given later results this was a mistake?

Pg. 7: ecosystem models have notoriously non-linear misfits in their parameter space; when this is highly multidimensional (as here) it can be difficult for optimisation to find

the global minimum misfit; how has this been achieved here?

Pg. 7, ln. 220: "stationary phase" - how exactly defined here?; particularly in the context of Figure B3c, which shows chlorophyll concentrations peaking ∼2 days later in the bacterial incubations

Pg. 8, ln. 234: can you explain if these values are meaningful, or is it just the relative values between phases that's important?

Pg. 8, ln. 279: This seems a pretty serious deficiency given the focus of this paper; I would interpret this as potentially a problem at the tuning stage; did you consider weighting fitting ammonium more heavily?

Pg. 9, ln. 283: "complexity" is an unusual way to describe a lack of sensitivity (which is what you seem to be suggesting); also, given the extended model performs no better (worse?) than the G98 model is this not to be expected?; i.e. you've added a means for the model to be different, but this means is far less powerful than what the model already has

Pg. 9, section 4: this discussion seems far too long for what's quite a simple set of experiments

Pg. 10, section 4.1: there's nothing in here about the (hard-wired!) 80% adjustment to growth rates caused by low silicate; this appear to be an unchangeable assumption

Pg. 10, ln. 310: do values of the f-ratio from bottle experiments relate well to those measured from the open ocean?; I can't think of any reason to suspect that they will, not least because there are no nitrifying bacteria including in the cultures here

Pg. 11, ln. 357: ah-ha, computational cost is finally mentioned

Pg. 11, ln. 351: I don't think it's ever made clear why there may be a preference for NH4 over NO3; it would be good to include mention of this so that readers understand why this aspect may be important in the work here

Pg. 12, ln. 360: the authors note different conceptual models for the Si:N relationship in this section, but stick instead with a highly simplified approach from a review almost 40 years old; and also remove this relationship from the tuning exercise undertaken; I would expect to see more justification for this - or potentially some form of model sensitivity analysis to evaluate how important it is

Pg. 12, ln. 375: is a biofilm something one might expect in the natural system?; it doesn't seem to be the sort of thing that would form in free water; also, it's unclear from the methods whether there's any agitation of the cultures to mimic ocean mixing

Pg. 13, ln. 392: the value of the f-ratio has been questioned as the wider role of nitrifying bacteria has been recognised; perhaps rephrase talking instead about the balancing roles of export and remineralisation?

Pg. 13, ln. 406: consider: Kamatani, A., Dissolution rates of silica from diatoms decomposing at various temperatures, Mar. Biol., 68, 91– 96, 1982

Pg. 14, ln. 426: model availability?; might be good to include the code too - it's simple enough

Pg. 21, Figure 1: presumably the gap between (NOx + NH4) in the two experiments is due to N getting stuck in (dead) organic matter?; bar PON / POC, was anything about this recorded in the experiments?

Pg. 21, Figure 1: the span of PO4 at day 14 (5-55) seems implausible given its narrow span at day 11 (30-35); especially as it narrows again at day 15 (5-18)

Pg. 22, Figure 2: not so axenic, eh?; is this contamination in the axenic incubations from repeatedly opening the vessels?

Pg. 23, Figure 3: so as well as having less NOx and NH4, the axenic experiments have less PON; where is the N going?

Pg. 24, Figure 4: it's idle curiosity, but what happens if you extend your model runs

past the time point that the laboratory cultures ran?; the model should permit this

Pg. 24, Figure 4: the inflections on some of the model plots here look rather artificial; can you explain why there are such sharp transitions around the 8-day mark?

Pg. 24, Figure 4: the spikes in chlorophyll in the cultures seem difficult to believe; do you think they are perhaps artifacts / measurement error?

Pg. 25, Figure 5: given that the key is the same in all of the plots, it would be better to not use it in plots where it interferes with the data (e.g. 5c)

Pg. 25, Figure 5: why are the fits without the excretion term all flat?; that's not what I'd expect at all; actually, I now realise that you're using two sets of dotted lines on this plot; one for the model output, one for the limiting concentration of the nutrients; this should be changed as it's a very confusing presentational choice

Pg. 26, Table 1: the text reads as if these crosses denote both (a) remineralisation, and (b) variable stoichiometry?; that seems a lot for one cross to bear!; however, in the table, it looks like you separate out the stoichiometry - I think this sentence needs rewording

Pg. 26, Table 1: as a stylistic aside, a cross is not necessarily the best way to denote that a model includes something; conventionally, ticks are used, with ticks and crosses meaning opposite things

Pg. 26, Table 1: where other models are presented, these are often older versions of these models; might it be better to report their current versions?

Pg. 27, Table A1: this is confusing; why not have separate columns for G98 and the extended model?; also, this table implies that some properties are not in the model, but you seem to have equations for them; meanwhile, there are other properties, e.g. dSi, for which no equation is presented

Pg. 27, Table A1: you appear to be using underscores rather than minus signs in units

at the base of this table

Pg. 30, Table A4: what do all of the columns mean here?; some explanation would be useful

Pg. 31, Table A5: please choose a table size that doesn't line-break your units

Pg. 32, Table A6: maybe pull the ODEs together in one place then follow-up with the separate terms afterwards?; it's a little difficult to parse the equations otherwise

Pg. 33, Equation 1: if there's a conditionality on a single term in an equation (as here) better to have a single ODE and put the conditionality inside this term (i.e. it's this value if X, zero if Y); this is easier to follow and makes it much easier to see where the important parts of the model's behaviour lie; duplicating the equations for the sake of a single term in them does not make things clear

Pg. 33, Equation 1: you should note somewhere that organic C is removed from an un-modelled reservoir of DIC; unmodelled because it's always in excess of the ecosystem model's requirements

Pg. 35, Equation 15: the presentation of equations 14 and 15 around the 14e3 divisor is different; this is an unnecessary confounding factor that makes the equations less readable

Pg. 35, Equation 16: why is this a hard-wired number (0.2) and not a parameter?; even if it's not something you change in your study (which seems a little strange given what you do change), having this as a clearly parameter rather than an undescribed constant is important

Pg. 36, Figure B1: I don't understand what this plot is showing; please explain what it means for a line to deviate from zero here; also, why is sensitivity time-variable in any case?; and why is it not monotonically variable in time?; I also note that it looks like DIN is super-sensitive compared to the other properties - is that a correct interpretation of this plot?

Pg. 37, Figure B2: a full explanation for what this plot is showing is critical; it is very difficult to understand what's being shown; also does the frequent occurrence of "NA" imply that some parameters should be excluded from this analysis?

Pg. 38, Figure B3: the key seems to omit reference to the bacterial model

Pg. 38, Figure B3: the failure of the model to capture the observed behaviour of the PON seems quite significant, but is not well-described in the text; it is also noticeably different from that of POC, which suggests interesting POM dynamics in the model that I would not expect; do the authors know what is going on here?

Pg. 38, Figure B3: would quartile or decile range be better here?; this may make your experiments look more messy than they actually are (i.e. it looks like you may have an outlier experiment); this may not be possible given the number of replicates

---

## Referee Comment (RC3) · Anonymous Referee #3 · 11 Nov 2020

In the manuscript "Modelling Silicate – Nitrate - Ammonium co-limitation of algal growth and the importance of bacterial remineralisation based on an experimental Arctic coastal spring bloom culture study" by Vonnahme et al. the authors present a new model development for diatom co-limitation of nutrients. Based on experimental data they expand the classical model by Geider et al. (1998), which remains its feasibility for larger (ecosystem) models, while improving the representation of algae growth. Improving biological parameterizations in ecosystem models is important and contributes to to improving their predicative capability. However, the authors should address a

some points listed below.

The authors report that "With the onset of the stationary phase in the bacteria-enriched cultures about 30% of the cells formed biofilms on the walls of the cultivation bottles (estimated after sonication treatment)." (line 230). The formation of such biofilms has occurred in other experiments before and cannot always be avoided. However, it does potentially have a huge impact of microbial dynamics and interactions. Therefore only reporting (and discussing) it is in sufficient, if one is to compare experimental results with a new modelling approach. I would suggest to run a model sensitivity analysis specifically targeting this.

The authors appropriately discuss quota models and their use. A different approach to model celluar nutrient kinetics, that has been argued to be more mechanistic, considers uptake sites for nutrients (Aksnes & Egge, 1991, Mar Ecol Prog Ser. 70:65-72). A good, though slightly technical, paper applying this approach and combining it with variable cellular stoichiometry is Flynn et al., 2018, PLoS Comput Biol 14(4): e1006118. Setting up a model like this for your data could be highly interesting, but beyond the scope of this study. However discussion the approach would provide a very useful context.

In the introduction (line 46) and in the discussion the authors mention the role of the impact of climate change on coastal phytoplankton succession, including projected increased DOM inputs via river run off. Several studies have found and/or suggested a delayed bloom due to increase turbidity (e.g. Opdal et al. 2019, Glob Change Biol. 2019;00:1–8), which should be mentioned here.

The authors mention both nitrate and ammonium as nitrogen sources. Additionally, urea can be a relevant nitrogen source in some systems. I am not sure how much of a role this plays in arctic ecosystems, but it should either be discussed or mentioned why it does not play a significant role.

Line 168: "…but the growth rate can be reduced (Hildebrand, 2002; Gilpin, 2004)". How can the growth rate be reduced? What can lead to this reduction?

Figure 6 and figure 7 do not exist.

Line 660: Table 1 is not the most up-to-date. Especially on the ecosystem model side it would be nice to see more recent developments reflected as well.

Especially in the abstract and the introduction there are several long (sometimes convoluted) sentences. To increase readability it would be could to rephrase these (Schachtelsaetze sind im Englischen nicht so hoch angesehen wie im Deutschen ;) ).
* * *

---

## Author Comment (AC2) · 26 Nov 2020

Summary
This manuscript presents an interesting combined laboratory and modelling study of the nutrient dynamics of a diatom species common in the Arctic. The laboratory component uses two experimental set-ups: 1. axenic cultures of the diatom species; 2. cultures of the diatom species that include associated bacteria. Short incubations (_2 weeks) of these cultures take them from exponential phase through to stationary phase, with the cultures sampled throughout to measure cell counts, nutrient concentrations, etc. After an initial period of diatom cell number growth (week 1) in both cultures, this stops as NOx and dSi concentrations approach limiting concentrations. However, NH4 is consistently higher in the non-axenic cultures, and the bacterial cell counts in these cultures increase exponentially during the latter period of the incubations (week 2). The authors interpret the presence of bacteria as being conducive to supplying the diatoms with regenerated nutrients. The modelling component uses a base model, G98, and an extended model based on this that includes a number of additional processes with relevance to the laboratory setting and the hypothesised role of bacterial remineralisation in supporting phytoplankton growth. The models are tuned to fit the laboratory data, with a manual phase to retain consistent parameter values between the models. The authors conclude with a discussion on the application of their results to the real Arctic and its expected future state.

I have listed a number of significant general comments below, followed by more specific and often minor comments. Overall, my assessment is that the manuscript requires major revision to clarify and amend the work described.

We want to sincerely thank the reviewer for the very thorough review and believe the suggestions helped to improve the manuscript considerably. We included all suggestions into a revised version as described below. We also changed the fixed 80% reduction term in our model to a parameter that was subject to the fitting approach and sensitivity analyses. We fitted the model again with a more automated fitting approach and reached better fits for both the G98 and extended EXT model.

General comments
Upfront, my modelling background means that I cannot comment directly on the details of the laboratory work in the study. However, I note that the experiments conducted exhibit anomalies that are not addressed in the manuscript. In Figure 1c, phosphate in bacterial cultures exhibits a strong spike upwards at day 8 that persists and shows high variability. In Figure 3d, chlorophyll in bacterial cultures shows a marked but temporary spike downward at day 8. While the latter is likely a replication or measurement issue, the former is harder to understand, and the manuscript does not discuss its scale or variability. It would be useful to know what the authors believed happened here, particularly in the case of phosphate where bottle concentrations approximately double against a backdrop of slowly declining phytoplankton and rising bacteria concentrations. The model may even be able to help on this point.

Since our study, does not focus on phosphate, we did not describe its dynamics in detail. However, we acknowledge that a short description and explanation of the anomalies is helpful for the reader to understand the overall experiment and nutrient dynamics and added some details.

The strong spike of phosphate after day 8 corresponds with the end of the exponential phase for algal growth and a spike of ammonium. At the same time bacteria abundances start increasing considerably. Thus, we explain the phosphate peak by increased bacterial regeneration (source of phosphate) and decreased algal uptake (sink of phosphate) at the same day. Due to the small bacteria biomass compared to algae, we assume limited phosphate incorporation in the bacteria biomass pool. Besides, the diatom culture may excrete additional DOM under stress, such as silicate limitation, contributing to labile DOM available for regeneration and thereby increasing the phosphate peak, which is however not part of the current extended model. We calculated the N:P ratio of the NH4 and PO4 peak at day 8, and realized that the ratio is approximately 1:1, which is different from the Redfield ratio. We see this as evidence that increased regeneration of NH4 and PO4 is not the only explanation for the PO4 peak and suggest the storage of (organic) polyphosphate in diatoms and release under stress as another potential source.

Changes in the text:

3.1) "With the onset of the stationary phase in the BAC+ experiment, PO4 and NH4 concentrations doubled within 2 to 4 days and stayed high with variations in phosphate concentrations, while they stayed low in BAC-. With depletion of NO3 in BAC+, NH4 concentrations remained high, while PO4 concentrations dropped."

4.1) "With the start of the stationary phase, NH4 and PO4 concentrations doubled, presumably due to decreased assimilation by the silicate starved diatoms and increased regeneration by bacteria, supplied with increasing labile DOM (doubled remineralisation rate in EXT) excreted by the stressed algae. After NO3 depletion at day 15, also PO4 concentrations drop, indicating a coupling of N:P metabolism "…" Excretion of organic phosphate by diatoms is also common in cultures with surplus orthophosphate (Admiraal and Werner, 1983), which can be another explanation of the phosphate peak after silicate becomes limiting."

The spike in Chl is based on one single measurement, since the upper and lower range represent max and min values. Since chlorophyll measurements are sensible towards light, and pH, we argue that this negative spike is a measurement artifact of a single sample of the experiment.

Changes:

Figure 4, 5, B1) "…Chlorophyll a concentration in experimental cultures with a potential outlier at day 8, presumably due to photodegradation, causing a negative spike."

The manuscript's model description appears incomplete, with equations for terms such as those for dSi omitted. More generally, the manuscript would be improved by simply making clear which models are being run – while the text refers to model G98 and "the extended model", the plots shown refer instead to "model + excr" and "model - excr". What might be helpful is to have some sort of diagram of the two main models being used (G98 and Extended) to help illustrate the main connections between state variables, and make clear the differences between the two models.

We added the missing equation and double-cheked for any other incomplete model descriptions.

Changes in Table:

| 7a) | Silicate uptake | $\frac{dSi_d}{dt} = V_S^C = \left(V_{max}Si_d \frac{Si_d - S_{min}}{K_{si}S_{min}}\right)C$ |
|---|---|---|
|  | *(Monod kinetics after Spilling et al., 2010)* |  |
| 7b) |  | $\frac{dSi_p}{dt} = \frac{-dSi_d}{dt}14$ |

We also clarified, which models are run and defined abbreviations (G98 and EXT models/ BAC- and BAC+ treatments) for the different models that we kept throughout the manuscript and figures. We also added a schematic diagram of the two main models, which we agree helps clarifying the differences considerably.

Changes:

2.2) "Details regarding model equations are provided in the Appendix (Table A1) and a schematic representation of the models is given in Figure 1. We used a dynamic cell quota model by Geider et al. (1998) to describe the BAC- experiment (G98). We then extended the G98 model to represent the role of silicate limitation, bacterial regeneration of ammonium, and different kinetics for ammonium and nitrate uptake (EXT) and fitted it to the BAC+ experiment while retaining the parameter values estimated for G98."…" For testing the importance of DON excretion we also ran the EXT model without DON excretion (EXT–excr)."

Fig. 1)
"

[Figure]

Figure 1. Schematic representation of the state variables and connections and controls in the G98 model (blue) and EXT model (purple). The EXT model has the same formulations as G98 with the additions shown in purple."

The description of the model tuning needs to be clearer. It's unclear why some parameters were picked for tuning while others weren't (e.g. remineralisation parameters were not tuned), or what the rationale for picking the training data streams was (e.g. model ammonium was "loosely constrained" to observations). The text mentions several R packages used, but these are presented without any information about what they do, how they work, or what assumptions they make. For instance, is parameter space sampled by latin hypercube, genetic algorithm, or via local misfit gradient? There's also an unclear distinction made around a "manual" component of this tuning exercise.

We added a detailed description of the model tuning, R packages, and parameter selection.

Changes:

[revised manuscript text omitted]

On a point related to tuning, I noted that the model has a key parameter for restricting phytoplankton growth (by 80%) in the absence of silicate, but that this parameter is not included in the tuning, which seems something of an omission (and, on a more presentation level, is hard-wired into the equations as a number rather than a parameter).

We agree that this parameter should be included in the tuning process since there may be variations from the study where this parameter is based on depending on the species and environment. It would also be interesting to include it in the tuning exercise to test if the 80% reduction can be confirmed after rigorous model tuning. We changed the model formulation and number of parameters in the tables accordingly and did the model fitting and sensitivity analyses again. The best fit is still an approximately 80% reduction.

Changes:

2.2) "Werner (1978) found that silicate limitation can lead to a 80% reduction in photosynthesis and a stop of chlorophyll synthesis in diatoms within a few hours. Hence, we added a parameter for the reduction of photosynthesis under silicate limitation (SiPS) and formulated a stop of chlorophyll synthesis under silicate limitations."

3.2) "The most sensitive added parameters in EXT were the remineralisation rate of refractory DON (remd, L1=0.24), the half saturation constant for ammonium (Knh4, L1=0.1) and the inhibition of photosynthesis under Si limitation (SiPS, L1=0.07), which was comparable to other sensitive parameters of the G98 model (Qmax, RC, αChl, ζ, n, I, ΘNmax, Table A1)."

4.2) "we modelled the response of diatom growth to silicate limitation by reducing photosynthesis through a parameterized fraction (SiPS) and a stop of chlorophyll synthesis below a certain threshold, based on experimental studies (Werner, 1978; Gilpin et al., 2004) and in accordance to other ecosystem scale approaches. Automated fitting showed the same 80 % reduction of photosynthesis as described by Werner (1978)."

Table A7, 1b)
$$\frac{dC}{dt} = Si_{PS}(P^C - \zeta V_N^C - R^C - xf)C = \mu C$$

Finally, the authors identify three central hypotheses in their study:

1. Bacterial regeneration of ammonium will extend a phytoplankton growth;
2. Silicate or nitrogen limitations have different physiological responses;
3. A simple experiment can adequately represent Arctic spring bloom dynamics.

On the first, the model has a very poor performance replicating the time history of ammonium concentrations. On the second, this study would be more convincing if the concentrations of Si and N had been experimentally manipulated to enhance / diminish limitation of each. On the third, the model's inconsistent fit with observations, and its omission of significant real world factors (e.g. zooplankton) make it difficult to evaluate whether this is true. And because the model is only being run for the short incubation period (i.e. rather than beyond the incubation period, or in some mode investigating more realistic or extrapolated settings), it's not clear how it behaves when "unleashed".

Overall, I very much like the combined laboratory and modelling approach, but judge that the modelling component in particular needs to be made much clearer, and evaluated more critically.

We agree that the hypotheses are not perfectly addressed with the data and model, due to the resons mentioned and reformulated the hypotheses in the following way:

We hypothesize that: I) Bacterial regeneration extends a phytoplankton growth period and gross carbon fixation; II) Diatoms continue photosynthesis under silicate limitation at a reduced rate if DIN is available; III) Cultivation experiments are powerful for understanding the major spring bloom dynamics.

1. Bacterial regeneration extends a phytoplankton growth period and gross carbon fixation
2. Diatoms continue photosynthesis under silicate limitation at a reduced rate if DIN is available
3. Cultivation experiments are powerful for understanding the major spring bloom dynamics

Each hypothesis can be tested by the cultivation experiment and can be discussed and evaluated in more detail with the modelling approach.

Concerning hypothesis 1 we suggest that the poor fit to ammonium is mainly related to the measurements rather than the model. Ammonium is most likely immobilized in the biofilm via adsorption to the EPS and accumulation in pockets unavailable to diatoms (See response to Referee #3). These immobile NH4 pools are still part of the measured data. With the model assuming all NH4 being available for algae growth, this is a problem. Hence, we did not put a strong weighting on ammonium for the fitting

routines but fitted the parameter to DIN instead. We did try to fit the model with heavy weighting on ammonium, but could still not reproduce the high ammonium concentrations in the stationary phase, while having a substantially worse fit for the other measured variables (RMSE=8.8).

Response to Referee #3:

"This could, in particular, explain the high values of measured NH4 compared to the model results as shown in Figure 5c. In addition we could add a small discussion of a potential pH dependence of NH4+ adsorption to the EPS in terms of the pKa values of NH4+ and carboxylic groups, which belongs to the acidic polysaccharides as a fraction of EPS:

- Carboxylic groups have a pKa < 5, i.e. far away from seawater pH ~ 8, which means that they are always in the deprotonized negatively charged form R-COO- in seawater.
- NH4+ has a pKa ~9 closer to seawater pH.
- Thus, the NH4+/NH3 ratio will be higher in more acidic microenvironments (pH ~7.5-8).
- Thus, a lower pH due to bacterial respiration would increase the concentration of NH4+ in comparison to the bulk medium, which results in a higher immobile NH4 pool due to adsorption to the EPS.
- This could explain the higher discrepancy between modelled and measured NH4+ values in the experiments with bacteria (as seen in Figure 5c)."

We also included a model run going beyond the experimental time frame in the supplementary material. Overall, the model reaches stable state after approx. 20 days when all nutrients are used up. Bacterial regeneration can keep some levels of N and C assimilation going beyond the loss for excretion and maintenance respiration, but they do not build substantially more biomass, which would be expected in the environment, where, however, sinking and grazing would lead to an additional export leading to a net loss. In order to keep the keep the manuscript streamlined, would prefer not adding these plots to the main manuscript, but to the Supplement instead. In the main manuscript, we suggest adding a short statement of the models stability if run longer (stable state after all nutrients are used up).

[Figure]

Figure S1: Model fit of the EXT model to the BAC- (blue) and BAC+ (red) experiment. Circles show median values and the colored polygons show the minimum and maximum of the measured data (n=3). Solid lines show the model outputs extended to 30 days of a) POC, b) PON, c) Chl (including outlier at day 8 in BAC+), d) NOX, e) NH₄, and f) Silicate. The dotted line show the output of EXT without excretion.

Specific comments

Pg. 1, ln. 20: "neglect" or "simplify"?; the distinction is important
We changed it to the term "simplify" since a general regeneration component is common in most models. When using the term "neglect" we were mostly focusing on a regeneration component dependent on bacteria biomass which is typically neglected in favor of a general purely substrate dependent remineralization formulation.

Pg. 1, ln. 23: surplus "and"
We removed the surplus "and"

Pg. 1, ln. 25: regarding the importance of organic matter excretion, was this based on observational evidence?
Yes, the excretion is based on the observation of algae aggregation in the stationary phase. However, since this is not the strong part of the model, we removed this statement from the abstract. We also added a more detailed discussion of the biofilm formation and implications for the model as described below and in more detail in the response to reviewer 3.

Pg. 1, ln. 26: "model complexity is comparable to other ecosystem models" – this is misleading as the model here is really an incubator model and not an ecosystem

model; it's missing most of the components that such models include (e.g. detritus, zooplankton)

When comparing model complexity (or number of parameters), we only compare the phytoplankton growth compartment within the ecosystem scale models, which are comparable to our extended model. We omit the complexity in ecosystem scale models not part of our model (e.g. Zooplankton, detritus). We clarified this by following change:

"Overall, model complexity (number of parameters) is comparable to the phytoplankton growth    and nutrient biogeochemistry formulations in common ecosystem models used ..."

Pg. 2: maybe be a little clearer on the distinction between autotrophic and heterotrophic bacteria throughout; cyanobacteria, for instance, are unlikely to play the role that's described as "bacterial" here

We distinguished heterotrophic and autotrophic bacteria clearer by adding the term heterotrophic to the bacteria, we are discussing for remineralization. We also agree that the term phototroph for cyanobacteria and heterotroph for the cyanobacteria associated bacteria is especially important to avoid confusion. However, we suggest adding the term heterotrophic only to the first occurrence of bacterial regeneration to keep the text readable.

Pg. 2: also, you should probably say something about the role of zooplankton; they graze phytoplankton and excrete some of the nitrogen they acquire; how quantitatively important is this here?; (I've added a cite to a paper that hints that they might not be all that important)

We added a section about zooplankton with a statement that their importance is low for regenerated production, compared to bacteria regeneration citing additional literature including the suggested paper doi:10.1016/j.dsr.2012.10.003

Change:

"Zooplankton grazing is typically of low importance for terminating blooms (e.g. Saiz et al., 2013), while inorganic nutrients are considered driving bloom termination (Krause et al. 2019, Mills et al. 2018)."…"Zooplankton may also release some ammonium after feeding on phytoplankton, but we suggest that this process is likely far less important than bacterial regeneration (e.g. Saiz et al., 2013). Previously measured ammonium excretion of Arctic mesozooplankton is typically low compared to bacterial remineralization (Conover and Gustavson, 1999), with the exception for one study in summer in a more open ocean setting (Alcaraz et al., 2010). "

Pg. 2, ln. 36: "marine phytoplankton *are*"?
We corrected the term accordingly

Pg. 2, ln. 41: predictions of what?
Corrected to: "predicitons of primary production with climate change"

Pg. 2, ln. 47: you might want to cite something like doi:10.1016/j.dsr.2012.10.003 as evidence of the reduced role of mesozooplankton in controlling / terminating blooms
As described above, we added following statement:

"Zooplankton grazing is typically of low importance for terminating blooms (e.g. Saiz et al., 2013), while inorganic nutrients are considered driving bloom termination (Krause et al. 2019, Mills et al. 2018)."

Pg. 2, ln. 48: remineralisation of what?; a bit of clarity would be helpful; dead diatoms, TEP, faecal material, etc.?
Clarified in the following way: "remineralisation of organic matter"

Pg. 2, ln. 57: heterotrophic bacterioplankton?
Yes, we clarified it by using the term "heterotrophic bacterioplankton" (See also response above).

Pg. 2, ln. 60: regarding "neglected", do you mean omitted or simplified?; most models include remineralisation of detrital material, and this implicitly bacterial
We are mainly referring to culture based experiments such as Ross and Geider 2009, and Flynn 2001, where remineralisation is omitted. We changed the term to "simplified or omitted".

Pg. 2, ln. 63: cultivation experiments normally provide parameter values for things like maximum rates of processes, half-saturations, etc., so it's not clear this is problematic; if model tuning is using cultivation experiments at equilibrium then this might be more of an issue
Problematic is that obtaining axenic algae cultures is challenging, not possible for most species, and does usually not last long (See our response to bacteria growing in the axenic treatment below). In previous cultivation experiments, no efforts for obtaining axenic cultures were mentioned, which hints to bacteria contaminated cultures. In these cultures, regeneration of e.g. ammonium and phosphate takes part. If half saturation constants, maximum uptake rates e.g. are based on non-axenic cultures with the assumption of the absence of regeneration, the values are likely too high, since the experiment will have a nutrient source not accounted for, which leads to underestimations of nutrient uptake, or in the worst case overestimation of growth efficiency if ammonium is not measured at all.

Changes:

"These latter models have been often developed and tuned based on cultivation experiments in which microbial remineralization reactions were assumed to be absent (e.g. Geider et al., 1998; Flynn, 2001) despite the fact that most algae cultures, likely including Geider et al., (1998) and Flynn (2001) are not axenic. Parameters estimated by fitting axenic models on non-axenic experiment may be misleading, mostly by an inflated efficiency of DIN uptake."

Pg. 3, ln. 72: this process was well-known long before this citation (Flynn, 1997); dig a bit deeper
We chose the citation by Flynn (1997) due to the modelling component in the paper, but agree that we should cite earlier literature. Thus we included the review by Morris (1974) and the review by Dortch (1990).

Dortch, Q.: The interaction between ammonium and nitrate uptake in phytoplankton, Marine ecology progress series, 61(1), 183-201, 1990.

Morris, I.: Nitrogen assimilation and protein synthesis, Algal physiology and biochemistry, 10, 1974.

Pg. 3, ln. 72: "iron has a strong control on silicate uptake" - I'm not sure that this is quite right; Si:C ratios are affected by Fe availability, but this is through continuing Si uptake but reduced C/N uptake and no cell division; my understanding is that Si *uptake* (within a certain range of Si:C) is not immediately affected by Fe; also the recent source given for this statement, Hohn et al., 2009, is a modelling PhD thesis

We agree that the control of Fe on Si is controversial and not well documente din earlier literature and that a modelling PhD thesis is not the best support for this hypothesis. Thus, we changed the statement in the following way:

Change:

"…C and N uptake is reduced under Fe limitation, while Si uptake continues, leading to increasing Si:C/N ratios (Werner, 1977; Firme et al., 2003),…"

Pg. 3, ln. 75: "ultimately too complex" - they add computational expense to large-scale ecosystem models; it's not clear that they are "too complex" (or even what is meant by this)
We changed it to "ultimately too computationally expensive when implemented in a global biogeochemical model"

Pg. 3, ln. 79: is phosphate limiting in the Southern Ocean?; in parts of its northern
extent, yes, but in the south its concentrations are high, no?
Pg. 3, ln. 79: "coastal"?; is there a distinction to be drawn with deep Arctic locations?
We clarified the sentence by: 1) removing the statement about phosphate, which is indeed only limiting
in the northern parts of the Southern Ocean, 2) by removing "coastal", since Fe is not limiting and DIN
is the primary limiting nutrient in most Arctic marine systems and by 3) adding supportive citations.

Change:

"In contrast to Antarctica, DIN is the primary limiting nutrient and iron is not limiting in most Arctic
systems (Tremblay and Gagnon, 2009; Moore et al., 2013)"

Pg. 3, ln. 81: yes and no; if simple lab experiments exclude factors such as zooplankton
excretion which might help fuel phytoplankton growth in parallel with bacterial
remineralisation, then it is questionable that they are demonstrating something that's
important in the real ocean
We agree that zooplankton N excretion may have an additional role, but as mentioned above, argue
that bacterial N regeneration is quantitatively more important. However, we relativized the sentence in
the following way:

Change

 "While simple lab experiments cannot represent all nutrient dynamics found in the environment (e.g. N
excretion by zooplankton), they can focus on the quantitatively most important dynamics, to facilitate
the development of simple, but accurate multinutrient models scalable to larger ecosystem models.

"

Pg. 3, ln. 86: how "associated" is this?; is it something that lives in direct physical
contact or shares the same waters?

The bacteria cultures were obtained from the diatom culture directly plated onto LB agar plates. This
means they grew together with the diatom outcompeting other bacteria in an environment heavily
influence by the algae, which was the only carbon source to the system. Microscopy showed bacteria
attached to the diatoms (mostly in the stationary phase), but mostly free-living. However, since the
bacteria are heterotroph and there was no other carbon soruce than the DOM coming from the diatom
culture, we see them as associated.

Change:

 "…inoculation with bacteria cultures, isolated beforehand from the non-axenic culture."

Pg. 3, ln. 87: again, what specifically is the issue with complexity?; is it model cost, or is
there some other aspect of complexity that disfavours inclusion in large-scale models?
Yes, it is model cost. We changed it to: "... aiming to keep the number of parameters, and
computational costs low to allow its use in larger ecosystem models."

Besides the computational costs a large number of parameters, as used in detailed physiological
models, is also more difficult to tune or verify with experimental/environmental data, which leads to the
issue of overfitting.

Pg. 3, ln. 93: good hypotheses!; however, you do not clearly return to them (e.g. "Regarding the hypotheses framed for this study . . .")
As mentioned above, we changed the hypotheses in the following way:

We hypothesize that: I) Bacterial regeneration extends a phytoplankton growth period and gross carbon fixation; II) Diatoms continue photosynthesis under silicate limitation at a reduced rate if DIN is available; III) Cultivation experiments are powerful for understanding the major spring bloom dynamics.

Pg. 4: this all sounds good, but my expertise in laboratory work is very limited
Thank you

Pg. 4, ln. 111: just for simplicity in the labelling, you might want to come up with nice short names for these experiments; e.g. BACT- (for the axenic) and BACT+ (for the non-axenic), or similar; this will make it easier to refer to the experiments in clear, non-wordy ways later on
We appreciate the suggestion and used the abbreviations BAC- and BAC+ throughout the corrected manuscript and figures.

Pg. 5, ln. 143: the origin of the f-ratio should be cited so that less familiar readers can understand what it is
We added following citation:
Eppley, R. W.: Autotrophic production of particulate matter, Analysis of marine ecosystems/AR Longhurst, 1981.

Pg. 5, ln. 144-146: this is a little confusing; perhaps spell it out with equations?
We added following equation:

"Equation C1. F-ratio estimation in the cultivation experiments with the average PON concentrations at day 13 to 15 ($PON^{d13-15}$) for the BAC- and BAC+ treatments."

$$f-ratio = \frac{PON_{BAC-\ d13-15}}{PON_{BAC-\ d13-15} + PON_{BAC+\ d13-15}}$$

Pg. 5, section 2.2: I don't think it ever hurts to have a schematic of a model's dynamics to supplement equations and (especially) verbal description
We added a schematic of the model dynamics of G98 and the extended model and briefly described the main dynamics focusing on the controls of photosynthesis, nitrogen assimilation and chlorophyll synthesis by C:N and Ch:N ratios, DIN concentrations, and light.

The schematic figure is shown above (Fig. 1). The following details, were added to the text:

Change:
"The Geider et al. (1998) model (G98) describes the response of phytoplankton to different nitrogen and light conditions and is based on both intracellular quotas and extracellular dissolved inorganic nitrogen (DIN) concentrations, allowing decoupled C and N growth (Fig. 1). Within this model, light is a control of photosynthesis and chlorophyll synthesis. C:N ratios and DIN concentrations control nitrogen assimilation, which is coupled to chlorophyll synthesis and photosynthesis. Chl:N ratios are controlling photosynthesis and chlorophyll synthesis."…"The EXT model keeps all formulations of the G98 and adds dynamics and interactions of silicate, nitrate and ammonium uptake, carbon and nitrogen excretion and bacterial remineralisation (Fig. 1)."…

Pg. 5, section 2.2: similarly, this section would be a lot clearer if you spelled out which models you were using, and ensured that the later plots use the same nomenclature; I initially misread the work ; I reckon it's: 1. G98; 2. Extended; 3. G98 – excretion; 4. Extended – excretion
We clarified the model runs used in the manuscript and used consistent nomenclature: 1. G98, 2. EXT (by default with excretion) 3. EXT$_{-excr}$. G98 does not have an excretion compartment.

Change:

"We used a dynamic cell quota model by Geider et al. (1998) to describe the BAC- experiment (G98). We then extended the G98 model to represent the role of silicate limitation, bacterial regeneration of ammonium, and different kinetics for ammonium and nitrate uptake (EXT) and fitted it to the BAC+ experiment while retaining the parameter values estimated for G98."…" For testing the importance of DON excretion we also ran the EXT model without DON excretion (EXT–excr)."

Pg. 5, section 2.2: stating up front an outline about the modelling strategy might help
(i.e. two models, tuned to the lab work, DOM addition, etc.)
We added a summary of which models were used for which experiment in the beginning in the now estensive chapter describing the fitting routines.

Change:

"Parameter of the G98 model were fitted to the BAC- experiment data and the EXT model was fitted to the BAC+ experiment data. The G98 parameter values were fitted first and retained without changes for the EXT model fitting."

Pg. 5, ln. 149: some model equations by the looks of things; the model description
appears incomplete
We added the missing equation and double-cheked for any other incomplete model descriptions.

Changes in Table:

| 7a) | Silicate uptake | $\frac{dSi_d}{dt} = V_S^C = \left(V_{max}Si_d\frac{Si_d - S_{min}}{K_{si}S_{min}}\right)C$ |
|---|---|---|
| | *(Monod kinetics after Spilling et al., 2010)* | |
| 7b) | | $\frac{dSi_p}{dt} = \frac{-dSi_d}{dt}14$ |

Pg. 6, ln. 164: equation for dSi seems missing in appendix
We added the missing equation in Table 7

$$\frac{dSi_d}{dt} = V_S^C = \left(V_{max}Si_d\frac{Si_d - S_{min}}{K_{si}S_{min}}\right)C$$

Pg. 6, ln. 166: "80% reduction" - is this where the 0.2 in the equations comes from (i.e.
1 - 0.8 = 0.2)?
Yes, we clarified it in the following way: "Werner (1978) found that silicate limitation can lead to a 80% reduction in photosynthesis and a stop of chlorophyll synthesis in diatoms within a few hours. Hence, we added a parameter for the reduction of photosynthesis under silicate limitation and formulated a stop of chlorophyll synthesis under silicate limitations."

However, we changed the fixed 80% value to a tunable parameter and rerun the fitting routine and sensitivity analysis as described above.

We argue that N dynamics are not directly coupled to Silicate limitation, but indirectly via reduced photosynthesis and inhibited chlorophyll production. The reference by Martin-Jezequel shows no direct coupling of N and Si, but overall different controls for Si and N/P, where Si is tighly linked to the cell cycle, fueled by heterotrophic respiration, while N/P are controlled by photosynthesis. Overall, Martin-Jezequel et al. supports our assumption of decoupled Si and N metabolism and is included in the manuscript as additional support:

Change:

"N and Si metabolism have different controls and intracellular dynamics, with N uptake fuelled by photosynthesis (as PCref in G98) and Si mainly fuelled by heterotrophic respiration (Martin-Jezequel et al., 2000). In general, we assume that nitrogen metabolism is not directly affected by silicate limitation (Hildebrand 2002, Claquin et al., 2002), but we expect cellular ratios to be affected by reduced photosynthesis and chlorophyll synthesis under Si limitation (Hildebrand, 2002; Gilpin, 2004)."

However, we acknowledge that there is 1 study by Gilpin et al., 2004, discussing a coupling of N:Si. Hence, we added it in the discussion:

Change:

"Studies on the coupling of silicate limitation on C, N, and Chl show inconclusive patterns, including a complete decoupling (Claquin et al., 2002), a relation of N to Si (Gilpin et al., 2004) and reduction of photosynthesis (Werner, 1978; Gilpin et al., 2004) while no new chlorophyll is produced (Werner, 1978; Gilpin et al., 2004)."…" Our cultivation study shows"…" ii) that coupling of Si:N:C:Chl is present. We do not expect a direct Si:N coupling, due to different controls of Si and N metabolism (Martin-Jézéquel et al., 2000.), but suggest indirect coupling via reduced photosynthesis."

We agree that this needs to be clarified.

Change:

 "It was assumed that this process is faster for freshly excreted DON compared to DON already present in the medium. Thus, we implemented a labile (DONl) and refractory (DONr) DON pool with different remineralization rates (rem, remd)."

We do not suggest a complete stop of remineralisation, but a net release of nitrogen of 0, since bacteria need more DIN on their own, rather than having the luxury of releasing it to the environment. We mention now two papers as support. Both papers base their fining on net changes in DIN. We tried to clarify it by following change:

Change:

"After Tezuka (1989), net bacterial regeneration of ammonium occurs at DOM C/N mass ratio below 10 and is proportional to bacterial abundances. Higher thresholds up to 29 have been found (e.g. Kirchmann, 2000), but we selected a lower number to stay conservative."

We refer to DOM as substrate for bacteria and clarified it: "DOM C/N ratios…. "

Pg. 6, ln. 177: does this mean that bacteria won't remineralise material with a C:N > 10?; that seems a little unlikely
As for line 173 we changed "bacterial remineralization" to "net bacterial ammonium regeneration".

Pg. 6, ln. 178: as the paper makes a fuss earlier about other models glossing over bacterial remineralisation, this simplified form is surprising
The main improvement of the model is to include a remineralisation rate controlled by: i) bacteria biomass, ii) substrate (DOM) C:N ratios, and iii) substrate origin (autochthonous, allochthonous). Other models typically have a fixed remineralisation rate either only dependent on the DOM/POM, or not controlled by any environmental variable. Thus, we still see our extension as a considerable improvement and consider a simple logistic growth estimate sufficient.
We could of course model bacteria growth via Michaelis-Menten kinetics based on 2 DOM pools, but this would not have any effect on the parameterization or modelling of algae physiology, which is the main goal of the paper, while increasing the number of parameters and computational costs, which we tried to keep low. Since, the aim of the model is not to model bacteria growth, but algae growth and intracellular C:N:Chl ratios we do not see that a more accurate and more complex model of bacteria growth would improve the manuscript.

Pg. 6, ln. 185: Table A6 - it looks to me like some equations are missing
We added the missing equations as mentioned above.

Pg. 6, ln. 186: which order of RK?; e.g. 3 or 4 (or higher)
We used the 2nd-3rd order Runge-Kutta method with automated stepsize control and added this information in the manuscript.

"The differential equations were solved using the ode function of the deSolve package (Soetaert et al., 2010) with the 2nd-3rd order Runge-Kutta method with automated stepsize control."

Pg. 6, ln. 185-191: this description of model tuning is far too brief; I'm not sure what's going on here; readers unfamiliar with R (I am one) will not understand what these different packages are doing or what their underlying assumptions are; this aspect of the modelling is too important to be glossed over so quickly; in general, to avoid the appearance of having just used the first package that occurred to you, expanding on the detail of the tuning (tools, approach, goal) would greatly benefit this description (hence and on Pg. 7)
We added a more detailed and extensive description of what the R packages are doing. deSolve is the most widely used solver for differential equations in R, and FME is a package for model fitting and sensitivity analysis developed as add on to deSolve. The tuning approaches via
$1^{st}$ manual fitting (based on RMSE error and graphical comparisons),
$2^{nd}$ automated fitting of selected parameters (avoiding collinearity/ linear dependence of sensitivity of 2 parameters = unidentifiable parameters), and choosing the more sensitive parameter in case of conflicts) via the Pseudorandom algorithm (searching for a global optimum),
$3^{rd}$ fine tuning for a local optimum using the Nelder Mead algorithm. $4^{th}$ check if the new parameters give a better fit regarding graphical comparisons and RMSE.

See the added chapter above in this response.

Pg. 6, ln. 192: the text should be clear on which observed variables were used to fit the model, why these were favoured, and whether any weighting was made to account for those judged better observed or more important; I would naively expect nutrient concentrations to be of prime importance but it's unclear what criteria the authors are using here (see my later remark about ammonium)
We also added more details about the observed variables used for tuning. We used POC, PON, Chl, and DIN (NOx + NH4) with standardized values (POC, 10xPON, 10xChl, DIN/10) and no further weighting. Due to rather poor quality of the NH4 data, we did not fit the model to NOx and NH4 separately.
Details about the parameters tuned and the constraints are also given in Table A3. Parameters were partly given by measured data, or tuned after constraining with measured or published constraints. In case of strong collinearity, only the most sensitive of the collinear parameters was tuned.

See the added chapter above in this response.

Pg. 7, ln. 197: this seems rather unsatisfactory; I would expect parallel runs with the same parameter values to be performed for axenic and not-axenic simulations, with an automated process (e.g. a genetic algorithm) to evaluate cost (i.e. misfit) before somehow generating new parameter values and iterating; having a manual component seems odd

We did do parallel runs with the same parameter values. The G98 model was fitted to BACT- data, but the resulting parameters were used for a G98 model run of both BACT- and BACT+ and kept without changes or further fitting as part of the EXT model. For the EXT model only the extended parameters relevant for describing our key observed variables (POC, PON, Chl, DIN) were fitted with the same rigorous fitting approach used for G98. The resulting parameters were used for the model of both BACT+ and BACT-.
We corrected the text in the manuscript to clarify what we did

We argue for an initial manual tuning approach in order to account have control of the model dynamics and to obtain good start parameters for the automated tuning approach and sensitivity/ collinearity analyses.

See the added chapter above in this response.

Pg. 7, ln. 198: what are these "known limitations"?; also, it's noticeable in the plots that the model solutions inflect strongly around the lag/stationary phase time point - is the model somehow different either side of this division?

The known limitations is that parameter tuning of the G98 in earlier attempts did not allow modelling the lag phase (Pahlow, 2005); later, howerver, Smith and Yamanaka (2007) showed that the Geider model can be brought to reproduce an initial lag phase.

The strong change around the beginning of the stationary phase is based on the threshold based approach to responses of Photosynthesis and Chl synthesis after Silicate limitation. Once silicate falls below a threshold, the physiology changes, which can be seen as a sudden change. Threshold based approaches are common in other dynamic models (e.g. threshold for cell division in Ross & Geider, 2009, threshold deciding which limiting nutrient decides the growth in Vichi et al., 2007).

Pahlow, M. (2005). Linking chlorophyll-nutrient dynamics to the Redfield N: C ratio with a model of optimal phytoplankton growth. Marine Ecology Progress Series, 287, 33-43.

Smith, S. L., & Yamanaka, Y. (2007). Quantitative comparison of photoacclimation models for marine phytoplankton. Ecological Modelling, 201(3–4), 547–552. https://doi.org/10.1016/j.ecolmodel.2006.09.016

See also the added chapter above in this response.

Pg. 7, ln. 202: "Colinearlity" - do you mean that you're looking for linkages between parameters here?

Collinearity is a measure for the parameter identifiability in complex simulation models (Brun et al., 2001) and allow identifying which parameter(s) (sets) can be uniquely constrained from the data.
If the perturbation of two different parameters can lead to the same change in the output variables, they are collinear, which makes them unidentifiable. Parameters were considered collinear and not identifiable in combination with a collinearity index higher than 20 as described in (Brun et al., 2001). In this case, only the more sensitive parameter was fitted.

Brun, R., Reichert, P. and Kunsch, H. R., 2001. Practical Identifiability Analysis of Large Environmental Simulation Models. Water Resour. Res. 37(4): 1015–1030.

See the added chapter above in this response.

Pg. 7, ln. 205: ammonium was "constrained loosely" - perhaps given later results this was a mistake?

Due to potential uncertainties associated with the ammonium data (e.g. immobilization in the biofilm by adsorption and micro-pockets, leakage of intracellular NH4 during filtration, freeze-thaw cycle), and high variability in published parameters (e.g. Eppley et al., 1969), we used wider constraints for ammonium related parameters. We do not agree that narrower constraints would lead to a better model fit to ammonium, since the new values would be within the same parameter space/ constraints. However, for consistency and usability of the model in other settings we narrowed down the constraints of published half saturation constants by Eppley et al., 1969. The reason for the poor fit is partly the lower weighting of the ammonium output during model fitting (We only fitted to DIN (NH4+NO3) and not separately to NH4 and NO3, but also the uncertainty of the ammonium values which likely include immobilized ammonium from algae cells, and the biofilm. We did a test run where we fitted the EXT model to POC, PON, Chl, NH4 and NO3, but the overall fit was substantially worse (RMSE = 9, instead of 2 with the DIN fit) with parameter values reaching the limits of the constraints.

Eppley, R. W., Rogers, J. N., & McCarthy, J. J. (1969). HALF-SATURATION CONSTANTS FOR UPTAKE OF NITRATE AND AMMONIUM BY MARINE PHYTOPLANKTON 1. Limnology and oceanography, 14(6), 912-920.

See the added chapter above in this response.

Pg. 7: ecosystem models have notoriously non-linear misfits in their parameter space; when this is highly multidimensional (as here) it can be difficult for optimisation to find the global minimum misfit; how has this been achieved here?

We agree that his is a potential problem and therefore we approached the problem from different angles.
First, we started with extensive manual tuning, as this gives a lot of insight for the modeler on how an optimal fit can be achieved and which parameters influence the results the most.
Secondly, we applied an automated parameter fitting procedure, which started with a collinearity analysis to make sure we are working with a parameter set that can actually be identified from the data. This reduces the risk of getting stuck in a local minimum. Subsequently, we ran a pseudorandom optimization routine to ensure a better coverage of the (identifiable) parameter space to increase the chance of approaching the global minimum randomly. The automated optimization routine ended with a directed descent algorithm, i.e. the Nelder Mead algorithm, that ensures quick convergence to the minimum.

See the added chapter above in this response.

Pg. 7, ln. 220: "stationary phase" - how exactly defined here?; particularly in the context of Figure B3c, which shows chlorophyll concentrations peaking _2 days later in the bacterial incubations

We defined the stationary phase by the sudden increase in phosphate and ammonium, silicate and DIN (for axenic cultures) values falling below minimum values in the model, and the Quantum yield dropping below 0.63. Since  the explanation of all of these evidence is spread over the page, we changed the term "stationary phase" to "day 8", which is less objective.

Pg. 8, ln. 234: can you explain if these values are meaningful, or is it just the relative values between phases that's important?

The Quantum yield is a ratio based on variable fluorescence of chlorophyll. The number ranges between 0 and 1 and show how efficiently energy is transported after adsorption. Generally, high numbers  indicate fit and active cells, while low numbers indicate stressed algae cells. Low N:C ratios are one stressor described to lead to inefficient energy transfer (low QY, Cleveland and Perry, 1987).

Cleveland, J. S., & Perry, M. J.: Quantum yield, relative specific absorption and fluorescence in nitrogen-limited Chaetoceros gracilis. Marine Biology, 94(4), 489-497, 1987.

Change:

 "The maximum photosynthetic quantum yield (Fv/Fm) is commonly used as a proxy of photosynthetic fitness (high QY), indicating the efficiency of energy transfer after adsorption in photosystem II.  Low values are typically related to stress, including for example nitrogen limitation (Cleveland and Perry, 1987). We found an increase in QY from approx. 0.62 to 0.67 d-1 in the exponential phase and a decrease to approx. 0.62 in the BAC+ treatment after 8 days and to approx. 0.58 in the BAC- treatment (Table A8)."

We suggest that the poor fit to ammonium is mainly related to the measurements rather than the model (immobilized ammonium in the biofilm). Hence, we did not put a strong weighing on ammonium for the fitting routines but fitted the parameter to DIN (NO3 + NH4) instead. When we did try to fit the model with heavy weighting on ammonium, we could still not reproduce the high ammonium concentrations in the stationary phase, while having a substantially worse fit for the other measured variables (RMSE=8.8). We discuss this limitation in the manuscript as follows:

Changes:

"The model was only fitted to total DIN, due to the potential uncertainties related to immobilized ammonium in the biofilm. In fact, a test run, fitting the EXT model to NO3 and NH4 separately lead to a substantially worse overall fit (RMSE=8.79)."

"While not all ammonium measured is also available for algae growth, discussion of the dynamics (decrease in the start, increase with the onset of the stationary phase), especially if also shown in the EXT model, are still useful to understand multinutrient dynamics (e.g. regeneration). Considering the overall higher concentrations of NO3, compared to NH4, discussions of total DIN dynamics, DIN.DIP ratios, and limitations are also meaningful."

"Fine scale DIN dynamics caused by ammonium – nitrate interactions were represented well (Fig. 6a). However, at the onset of the stationary phase, ammonium concentrations of the model were one order of magnitude lower than in the experiment, showing a major weakness (Fig. 6c). Increased weighting of ammonium during the model fitting led to a slightly better fit to ammonium, but a substantially worse fit of the model to POC, PON, and Chl (RMSE$_{EXT}$=8.79), indicating that the problem lies with the ammonium data (immobilized ammonium)."

We agree that complexity is not the best fitting term and changed it to "sensitivity" or "added parameters".
Changed to: "…was more sensitive than any of the original model parameters. Hence, the added parameters of the extended…"

Change:
"The sensitivity analysis (Fig. B1, Table A1) revealed that the sensitivity of the added parameters in EXT is overall comparable to the sensitivity of the original parameters in G98. The model outputs were most sensitive to $P_C^{Ref}$ (L1=0.8, L2=1.5), which is a parameter in both G98 and EXT. The most sensitive added parameters in EXT were the remineralisation rate of refractory DON (rem$_d$, L1=0.24), the half saturation constant for ammonium (K$_{nh4}$, L1=0.08) and the inhibition of photosynthesis under Si limitation (Si$_{PS}$, L1=0.08), which was comparable to other sensitive parameters of the G98 model (Q$_{max}$, R$_C$, $\alpha_{Chl}$, $\zeta$, n, I, $\Theta_N^{max}$, Table A1). Small perturbations of the parameters only indirectly related to the fitted output variables did not lead to changes in POC, PON, Chl, or DIN."

We agree that we originally thought this was simple set of experiments. However, the additional model interpretation, though very valuable we believe, does warrant a lengthier discussion. Also the detailed and thorough reviews for this manuscript made it impossible for us to substantially shorten the discussion and we believe that shortening the discussion would not be possible while addressing all comments of the three reviewers. We still tried to keep it as short as possible with the suggested changes.

Pg. 10, section 4.1: there's nothing in here about the (hard-wired!) 80% adjustment to growth rates caused by low silicate; this appear to be an unchangeable assumption
Firstly, we changed the 80% formulation into a tuneable parameter. Secondly, we added a sentence in the discussion. "Photosynthesis was reduced by 80% after silicate became limiting, which is in accordance with earlier experimental studies (Tezuka..)."

Pg. 10, ln. 310: do values of the f-ratio from bottle experiments relate well to those measured from the open ocean?; I can't think of any reason to suspect that they will, not least because there are no nitrifying bacteria including in the cultures here
We do not expect that the f-ratio in our bottle experiment is representative for the open ocean, but compare the values as starting point for discussing why. We argue that a discussion of the differences between the bottle experiment and open ocean values (e.g. grazing, nitrification) can show the limitations of the experiment and thereby the limitations of our model. We also add a reference to nitrification to the lacking processes.

Change:

"While we do not expect the f-ratio in our bottle experiment to be directly comparable to open ocean system, which does include a variety of algal taxa beyond C. socialis, a comparison can aid to identify limitations in our experiment and model. Regenerated production is significant in polar systems and our estimated experimental f-value of 0.31 is slightly below the average for polar systems (Harrison and Cota, 1990, mean f-ratio=0.54). Nitrification is a process supplying about 50% of the $NO_3$ used for primary production in the oceans, which may lead to a substantial underestimation of regenerated production (Yool et al., 2007), inflating the f-ratio interpreted as estimate for new production, potentially also in the study by Harrison and Cota (1990). The absence of vertical PON export in our experiment may be another explanation..."

Pg. 11, ln. 357: ah-ha, computational cost is finally mentioned
We also added this information in previous formulations of "cost" and "complexity" in the corrected version of the MS.

Pg. 11, ln. 351: I don't think it's ever made clear why there may be a preference for NH4 over NO3; it would be good to include mention of this so that readers understand why this aspect may be important in the work here
The conversion of $NH_4$ to biomass $NH_3$ is energetically much cheaper, making it the preferred source. We added following information:

"Due to the metabolic costs related to nitrate reduction to ammonium, ammonium uptake is preferred over nitrate, potentially leaving more energy for other processes (Lachmann et al., 2019). Ammonium can even inhibit or reduce nitrate uptake over certain concentrations (Morris, 1974). The dynamics are mostly controlled by intracellular processes, such as glutamate feedbacks on nitrogen assimilation, cost for nitrate conversion to ammonium, or lower half saturation constants of ammonium transporters (Flynn et al., 1997)."

Lachmann, S. C., Mettler-Altmann, T., Wacker, A., & Spijkerman, E.: Nitrate or ammonium: Influences of nitrogen source on the physiology of a green alga, Ecology and evolution, 9(3), 1070-1082, 2019.

Pg. 12, ln. 360: the authors note different conceptual models for the Si:N relationship in this section, but stick instead with a highly simplified approach from a review almost 40 years old; and also remove this relationship from the tuning exercise undertaken; I would expect to see more justification for this - or potentially some form of model sensitivity analysis to evaluate how important it is

We argue that N dynamics are not directly coupled to Silicate limitation, but indirectly via reduced photosynthesis and inhibited chlorophyll production. The reference by Martin-Jezequel shows no direct coupling of N and Si, but overall different controls for Si and N/P, where Si is tighly linked to the cell cycle, fueled by heterotrophic respiration, while N/P are controlled by photosynthesis. Overall, Martin-Jezequel et al. supports our assumption of decoupled Si and N metabolism and is included in the manuscript as additional support.

However, we acknowledge that there is 1 study by Gilpin et al., 2004, discussing a coupling of N:Si. Hence, we added it in the discussion:

Change:

"Studies on the coupling of silicate limitation on C, N, and Chl show inconclusive patterns, including a complete decoupling (Claquin et al., 2002), a relation of N to Si (Gilpin et al., 2004) and reduction of photosynthesis (Werner, 1978; Gilpin et al., 2004) while no new chlorophyll is produced (Werner, 1978; Gilpin et al., 2004)."…" Our cultivation study shows"…" ii) that coupling of Si:N:C:Chl is present. We do not expect a direct Si:N coupling, due to different controls of Si and N metabolism (Martin-Jézéquel et al., 2000.), but suggest indirect coupling via reduced photosynthesis."

We agree that the 80% reduction should not be a fixed parameter, but tuneable. We adjusted the model accordingly. We also included the parameter to the sensitivity analysis and repeated the fitting routine.

Pg. 12, ln. 375: is a biofilm something one might expect in the natural system?; it doesn't seem to be the sort of thing that would form in free water; also, it's unclear from the methods whether there's any agitation of the cultures to mimic ocean mixing

We would not expect biofilm formation in open oceans, but aggregation, which is commonly found in the end of spring blooms increasing the vertical export (e.g. Thornton, 2002). Both processes are similar in the way that algae aggregate via EPS facilitating a specific and active microbiome. We added a sentence about biofilm as proxy for marine snow in the discussion. Ocean mixing was mimicked by inverting all bottles 2-3 times a day (added to the methods).

Change in the discussion:

"While we would not expect biofilms in the open ocean, aggregation of algae cells, facilitated by EPS is common towards the end of spring blooms, increasing vertical export fluxes (e.g. Thornton, 2002). Chaetoceros socialis is in fact a colony forming diatom building EPS-rich aggregates in nature (Booth et al., 2002)."

Change in the methods:

"The cultures were incubated at 4°C and 100 µE m-2 s-1 continuous light and mixed 2-3 times a day to keep the algae and bacteria in suspension."

Pg. 13, ln. 392: the value of the f-ratio has been questioned as the wider role of nitrifying bacteria has been recognised; perhaps rephrase talking instead about the balancing roles of export and remineralisation?

We replaced the term f-ratio by "regenerated production" and added that the higher regenerated production is due to increased remineralization compared to export.

Pg. 13, ln. 406: consider: Kamatani, A., Dissolution rates of silica from diatoms decomposing at various temperatures, Mar. Biol., 68, 91– 96, 1982

We included the reference

Pg. 14, ln. 426: model availability?; might be good to include the code too - it's simple
Enough
The R code is now available at github.

Pg. 21, Figure 1: presumably the gap between (NOx + NH4) in the two experiments is
due to N getting stuck in (dead) organic matter?; bar PON / POC, was anything about
this recorded in the experiments?
We did not differentiate between life and dead organic matter, but assume mostly life organic matter until the stationary phase where biofilm formation played a role indicating EPS production, which can contribute to the measured PON and POC.
We agree that NH4 adsorption to organic matter (EPS) can play an important role and is likely one of the main explanations for the poor (lower) model fit of ammonium to the measured data. In addition, NH4 may be immobilized in micro-pockets of the biofilm unavailable for algae uptake.
However, we attribute the gap of DIN between the experiments mainly to a) increased NH4 regeneration in BACT+, with some ammonium likely immobilized in the biofilm (= higher NH4 concentrations), and b) preferred NH4 uptake over NO3 and NO3 uptake inhibition by NH4 leading to higher NO3 concentrations in the BACT+ treatment due to slower uptake. The PON/POC ratios change due to carbon overconsumption (Schartau et al. 2007), which is most relevant under N limitation, whule Si limitation has a more direct effect on photosynthesis (Lippemeier et al., 1999, Thangaraj et al., 2019; Liu et al., 2020). All these 3 dynamics are part of the extended model taking bacterial processes and NH4-NO3 interactions into account.

Pg. 21, Figure 1: the span of PO4 at day 14 (5-55) seems implausible given its narrow
span at day 11 (30-35); especially as it narrows again at day 15 (5-18)
We argue that the large range is plausible since it is i) based on 1 data point, which may be an outlier and ii) it corresponds with high variation in bacteria abundances, which are ultimately responsible for the high PO4 value presumably originating from remineralization. Especially towards the end of the experiment it is not implausible that the different bottles behave differently.

Change in figure legend:

"c) PO42- with a potential outlier at day 14 leading to a negative peak"

Pg. 22, Figure 2: not so axenic, eh?; is this contamination in the axenic incubations
from repeatedly opening the vessels?
As mentioned in the results, the bacteria growing towards the end are still in so low abundances compared to the bacteria enriched experiment, that it is effectively axenic. Obtaining and especially maintaining axenic diatom cultures is challenging and does typically not last very long. Since we used independent bottles during the experiment, contaminations during the course of the experiment are not possible (bottles were not opened before the sampling day). However, antibiotic treatments attack mostly active bacteria cells susceptible to the antiobiotics, while endospores and antibiotic resistant bacteria can survive. We believe that the bacteria starting to grow at day 14 originate form endospores activated by the high concentrations of DOM excreted by the stressed algae.

Change in the methods:

"We ensured sterile conditions during the experiment by keeping the cultivation bottles closed until sampling. However, endospores may survive the antibiotic treatment in low numbers and start growing especially towards the end of the experiment."

Pg. 23, Figure 3: so as well as having less NOx and NH4, the axenic experiments have
less PON; where is the N going?

We suggest that the N is contributing to a higher DON pool (not measured) in the axenic experiments, which is not shown in Figure 1 and 3. The DON could be remineralized in the experiments with bacteria yielding higher NOx, NH4+ and PON.

We hope that our schematic representation of the model added to the methods helps to clarify it (See Fig. 1 above).

we added the output of a prolonged model run in the supplement (See above).

The sharp transition is due to the threshhold based formulation of reduced photosynthesis and
inhibited Chl synthesis under Si limitation. At day 8, the silicate limitation threshold is reached and
Photosynthesis is reduced and Chl synthesis inhibited. As describe dabove, threshold based
modelling approaches are not uncommon.

Yes, as mentioned above these spikes represent a single data point that can be measurement
artifacts and added this information to the figure legend.

We now only put the legend in plot a and mention that the legend is valid for all subplots.

We changed the style of the lines. The fit without excretion is not the flat line, but the dotted line close
to the +excr model. The –excr model simply modelled the excretion fraction of the +excr model into the
maintenance respiration term (general loss without being available for remineralization). Since our
system was highly affected by ambient DOM (likely terrestrial), the difference is little, showing that the
regenerated production in our experiment is mostly caused by terrestrial DOM regeneration rather
than freshly produced DOM regeneration.

We clarified the table caption and used ticks and crosses instead.
While the full ecosystem scale models may have more recent versions with updated formulations, we
give the original reference to the biogeochemical compartment of the ecosystem scale models, which
are still quite old. We will however, added references to the most recent full-scale models used in
addition to the reference only describing the algae growth formulations. We added following
references to more recent ecosystem scale model formulations:

BFM model:_ Smith, K. M., Kern, S., Hamlington, P. E., Zavatarelli, M., Pinardi, N., Klee, E. F., &
Niemeyer, K. E. (2020). BFM17 v1. 0: Reduced-Order Biogeochemical Flux Model for Upper Ocean
Biophysical Simulations. *Geoscientific Model Development Discussions*, 1-35.

ReCom-2 model: Schourup-Kristensen, V., Wekerle, C., Wolf-Gladrow, D., Völker, C. (2018): Arctic
Ocean biogeochemistry in the high resolution FESOM 1.4-REcoM2 model, Progress in
Oceanography, 168, 65-81,doi:10.1016/j.pocean.2018.09.006.

MEDUSA model: Henson, S. A., Cole, H. S., Hopkins, J., Martin, A. P., & Yool, A. (2018). Detection of
climate change-driven trends in phytoplankton phenology. *Global Change Biology*, *24*(1), e101-e111.

NEMURO model: Anju, M., Sreeush, M. G., Valsala, V., Smitha, B. R., Hamza, F., Bharathi, G., &
Naidu, C. V. (2020). Understanding the Role of Nutrient Limitation on Plankton Biomass Over Arabian

Sea Via 1-D Coupled Biogeochemical Model and Bio-Argo Observations. *Journal of Geophysical Research: Oceans*, *125*(6), e2019JC015502.

SINMOD model: Alver, M. O., Broch, O. J., Melle, W., Bagøien, E., & Slagstad, D. (2016). Validation of an Eulerian population model for the marine copepod Calanus finmarchicus in the Norwegian Sea. *Journal of Marine Systems*, *160*, 81-93.

NPZD model: Gruber, N., Frenzel, H., Doney, S. C., Marchesiello, P., McWilliams, J. C., Moisan, J. R., Oram, J. J., Plattner, G., and Stolzenbach, K. D.: Eddy-resolving simulation of plankton ecosystem dynamics in the California Current System, Deep Sea Research Part I: Oceanographic Research Papers, 53(9), 1483-1516, 2006.

And we added and discussed following culture-scale model suggested by reviewer 4:

Flynn, K. J., Skibinski, D. O., & Lindemann, C. (2018). Effects of growth rate, cell size, motion, and elemental stoichiometry on nutrient transport kinetics. PLoS computational biology, 14(4), e1006118.

Pg. 27, Table A1: this is confusing; why not have separate columns for G98 and the extended model?; also, this table implies that some properties are not in the model, but you seem to have equations for them; meanwhile, there are other properties, e.g. dSi, for which no equation is presented
We had only a G98 column because all state variables are part of the extended model (The EXT model is the G98 model with added variables). We added a column for EXT besides G98 with ticks for every state variable for clarification. We also mentioned the equation for each state variable in table A6/A7 and added the missing equations.

Pg. 27, Table A1: you appear to be using underscores rather than minus signs in units at the base of this table
We changed it.

Pg. 30, Table A4: what do all of the columns mean here?; some explanation would be Useful
We shortened the table slightly and explained all columns in detail in the corrected version.

Change:
Table A4. Output of the sensitivity analysis (senFun of the FME package in R) with the value for each parameter and different sensitivity indices obtained after quantifying the effects of small perturbations of the parameters.on the output variables (POC, PON, Chl, DIN). The L1 and L2 norms are normalized sensitivity indices defined as $L1 = \sum \frac{|S_{i,j}|}{n}$ and $L2 = \sqrt{\frac{S_{i,j}^2}{n}}$ with $S_{i,j}$ being the the sensitivity of parameter i for model output j.

| par | value | L1 | L2 | Mean | Min | Max |
|-----|-------|------|------|-------|-------|------|
| | | | G98 | | | |
| ζ | 1.00 | 0.10 | 0.19 | -0.02 | -0.15 | 0.98 |
| R$^C$ | 0.07 | 0.04 | 0.05 | -0.03 | -0.08 | 0.14 |

Pg. 31, Table A5: please choose a table size that doesn't line-break your units
We adjusted the table size

Pg. 32, Table A6: maybe pull the ODEs together in one place then follow-up with the separate terms afterwards?; it's a little difficult to parse the equations otherwise
we changed the order accordingly.

Pg. 33, Equation 1: if there's a conditionality on a single term in an equation (as here) better to have a single ODE and put the conditionality inside this term (i.e. it's this value if X, zero if Y); this is easier to follow and makes it much easier to see where the important parts of the model's behaviour lie; duplicating the equations for the sake of a single term in them does not make things clear
We changed the equations accordingly.

Pg. 33, Equation 1: you should note somewhere that organic C is removed from an unmodelled reservoir of DIC; unmodelled because it's always in excess of the ecosystem model's requirements
We added the information to the schematic figure in the methods and mentioned it next to the equation.

Pg. 35, Equation 15: the presentation of equations 14 and 15 around the 14e3 divisor is different; this is an unnecessary confounding factor that makes the equations less readable
We changed the form of eq 14 to the same format as in eq 15.

Pg. 35, Equation 16: why is this a hard-wired number (0.2) and not a parameter?; even if it's not something you change in your study (which seems a little strange given what you do change), having this as a clearly parameter rather than an undescribed constant is important
We changed this parameter to a tuneable parameter and included it into the sensitivity analyses and parameter fitting exercise.

Pg. 36, Figure B1: I don't understand what this plot is showing; please explain what it means for a line to deviate from zero here; also, why is sensitivity time-variable in any case?; and why is it not monotonically variable in time?; I also note that it looks like DIN is super-sensitive compared to the other properties - is that a correct interpretation of this plot?
The sensitivity analyses in the FME package tests the sensitivity of the model output (here DIN, POC, Chl, DIN) with changing parameter values within the predefined constraints. The plot shows the deviation from the model output towards the measured data over time. We realized that this figure is too complex while adding little information to the manuscript and removed it.

Pg. 37, Figure B2: a full explanation for what this plot is showing is critical; it is very difficult to understand what's being shown; also does the frequent occurrence of "NA" imply that some parameters should be excluded from this analysis?
The plot shows pairwise comparisons of parameter sensitivity/ sensitivity functions. On the upper right the pairwise data are shown for each tuneable parameter with the boundaries/constraints given in table A3. The sensitivity is given for POC (blue), PON (red) and Chl (green). The correlation coefficients are given in the lower left corner. NAs indicate no correlation because of low sensitivity. We realized that this figure is too complex while adding little information to the manuscript and replaced it with following table.

Change:

Table A8. Output of the collinearity or parameter identifiability analysis using the collin function of the FME R package (Soetaert et al., 2010b). A subset of any combinations of two parameter with a collinearity above 20, indicating non-identifiable parameter combinations is given (Brun et al., 2001).

| $\zeta$ | $R^C$ | $\theta^N_{max}$ | $Q_{min}$ | $Q_{max}$ | $\alpha^{Chl}$ | $I$ | $n$ | $K_{no3}$ | $P^C_{ref}$ | collinearity |
|---|---|---|---|---|---|---|---|---|---|---|
| 1 | 0 | 1 | 0 | 0 | 0 | 0 | 0 | 0 | 0 | 31 |
| 1 | 0 | 0 | 0 | 1 | 0 | 0 | 0 | 0 | 0 | 59 |
| 1 | 0 | 0 | 0 | 0 | 1 | 0 | 0 | 0 | 0 | 42 |
| 1 | 0 | 0 | 0 | 0 | 0 | 1 | 0 | 0 | 0 | 42 |
| 1 | 0 | 0 | 0 | 0 | 0 | 0 | 1 | 0 | 0 | 74 |
| 0 | 1 | 0 | 0 | 0 | 0 | 0 | 0 | 0 | 1 | 22 |
| 0 | 0 | 1 | 0 | 1 | 0 | 0 | 0 | 0 | 0 | 32 |
| 0 | 0 | 1 | 0 | 0 | 1 | 0 | 0 | 0 | 0 | 26 |

Pg. 38, Figure B3: the key seems to omit reference to the bacterial model
We added the information in the legend.

Pg. 38, Figure B3: the failure of the model to capture the observed behaviour of the PON seems quite significant, but is not well-described in the text; it is also noticeably different from that of POC, which suggests interesting POM dynamics in the model that I would not expect; do the authors know what is going on here?
After G98 carbon is continuously fixed, even under nitrogen limitation (Carbon overconsumption, Schartau et al., 2007), while nitrogen is slowly used up for maintenance (maintenance respiration term), leading to a decoupling of POC and PON.  The main reason for the failure of the G98 model is the neglection of bacterial DIN regeneration. Thus, the PON dynamics are quite well modelled for the BACT- experiment, while the BACT+ experiment shows severe limitations. In fact, this is one of the main arguments showing the need to include bacterial regeneration. The model may be tuned to an artificially better fit the BACT+ treatment by increased DIN uptake efficiencies, but this would lead to a substantially poorer fit to the BACT- experiment. As discussed on p2 l.63 this fitting of the G98 model without a bacterial regeneration component on non-axenic culture experiment can lead to misleading interpretations and kinetic parameters (e.g. half saturation constants). We added this information to the discussion.

Schartau, M., Engel, A., Schröter, J., Thoms, S., Völker, C., & Wolf-Gladrow, D.: Modelling carbon overconsumption and the formation of extracellular particulate organic carbon, 2007.

Pg. 38, Figure B3: would quartile or decile range be better here?; this may make your experiments look more messy than they actually are (i.e. it looks like you may have an outlier experiment); this may not be possible given the number of replicates

With three measured values per day and treatment we prefer to show all values separately instead of artificially calculating error estimates (e.g. quartiles, deciles, standard deviations).

---

## Author Comment (AC3) · 26 Nov 2020

In the manuscript, "Modelling Silicate – Nitrate - Ammonium co-limitation of algal growth and the importance of bacterial remineralisation based on an experimental Arctic coastal spring bloom culture study" by Vonnahme et al. the authors present a new model development for diatom co-limitation of nutrients. Based on experimental data they expand the classical model by Geider et al. (1998), which remains its feasibility for larger (ecosystem) models, while improving the representation of algae growth. Improving biological parameterizations in ecosystem models is important and contributes to improving their predicative capability. However, the authors should address some points listed below.

We want to thank the reviewer for the positive and very helpful review and addressed the points raised as described below.

The authors report that "With the onset of the stationary phase in the bacteria-enriched cultures about 30% of the cells formed biofilms on the walls of the cultivation bottles (estimated after sonication treatment)." (line 230). The formation of such biofilms has occurred in other experiments before and cannot always be avoided. However, it does potentially have a huge impact of microbial dynamics and interactions. Therefore only reporting (and discussing) it is insufficient, if one is to compare experimental results with a new modelling approach. I would suggest to run a model sensitivity analysis specifically targeting this.

We agree that the biofilm formation can have a substantial impact on some microbial dynamics and the identity of different carbon pools. Since, the biofilm only contributed to 30% of the cell counts we are still confident that the model is well suited to represent the dynamics of the experiment and coastal Arctic spring blooms. The current model has various dynamics changing after silicate limitation, especially in the presence of bacteria and $NH_4$ regeneration. As outlined in the response to reviewer 2, we changed the reduction of photosynthesis after Si limitation from a hard-wired number (80%) to a tuneable parameter. Since Si limitation corresponds with the timing of biofilm formation, we assumed that the silicate limitation parameters (in particular $Si_{PS}$ – reduction of photosynthesis after Si limitation) could describe the changed dynamics.

To test if this assumption holds true and to deepen the discussion of biofilm related dynamics, we suggest 3 potential dynamics which are likely different in the biofilm. Since the biofilm formation corresponds with Si limitation, we modelled changed dynamics after Si limitation to represent specific changes known to be differen in biofilms.

1) DOC coagulation to EPS as part of the POC pool
   a. $dPOC_{EPS}/dt = x_f * x_{eps} * C$
   b. ($x_{eps}$ – fraction of coagulation excreted DOC)
2) Increased DOM excretion in the stationary phase
   a. IF (Si < lim){ $x_f = x_{f2}$ } else { $x_f = x_{f1}$}
   b. ($x_{f1}$- excretion before Si limitation, $x_{f2}$ –excretion after Si limitation)
3) Increased remineralization of excreted DON in the stationary phase
   a. IF (Si < lim){ rem= $rem_2$ } else { rem = $rem_1$}
   b. ($rem_1$- remineralization rate before limitation, $rem_2$ –remineralization rate after Si limitation)

We extended the model as described above and compared the new model output with the original fit. We changed tuned the new parameter manually until the model output showed substantial differences (approx. > 10% in POC, PON, Chl, or DIN). Eventually, we tested the effects of 100% DOM coagulation ($x_{eps}$ = 1), 1000x higher remineralization rate after Si limitation (rem2 = 10000, rem1=10), 2x higher DOM excretion after Si limitation (xf1 = 0.06, xf2 = 0.12). The order of magnitude of perturbations needed to get changes of 10% of at least 1 output variable gives an indication of the parameters sensitivity. We then tried to tune the new model again to the initial fit by changing the $Si_{PS}$ parameter. For each case, we were overall able to return to the original fit with less  (1 & 2) or equal (3) perturbations of the $Si_{PS}$ parameter than was perturbed in the added parameters. This shows that $Si_{PS}$ is more sensitive and collinear (unidentifiable) with the added parameters, which shows clearly that the 3 suggested model extensions would not improve the model without additional data (e.g. EPS measurements). We added this discussion in a shortened way to the manuscript. More details are given below:

1) The POC can include not only algae biomass, but also EPS that holds the biofilms together. For estimating the potential importance of this POC pool, we added a model run, where all the excreted carbon (given by $x_f$ * POC) is coagulating to EPS and thereby contributing to the POC pool. As shown in Fig 1 the outcome are ca 30% increased POC values in the stationary phase, which is in accordance with our estimate from the sonication treatment. However, it is unlikely that all excreted DOM aggregates to EPS (Schartau et al., 2017) and earlier studies describe much lower proportions of EPS being part of the EPS pool (up to 7% of an biofouling diatom biofilm, Khandeparker and Bhosle, 2009), with the highest fraction after nitrogen and silicate limitation in the stationary phase. A potential model extension to account for EPS aggregation that contributes to the EPS pool would be the approach, described by Schartau et al. (2017) who model carbon excretion (3 different DOC pools) and aggregation to TEP (transparent exopolymeric substances). However, since we did not measure cellular C and extracellular EPS separately, we argue that the extension requiring 11 additional parameters and 3 additional state variables of TEP (EPS) and 3 different DOC pools would i) not be in line with our goal to develop a simple model, and ii) would not be justified by the measured data, making the tuning process rather speculative (overfitting issue). Nevertheless, we acknowledge that this process needs to be discussed and we add the figure below to the supplement to show the maximum potential importance of EPS aggregation (assuming immediate aggregation of excreted DOC).

For estimating the importance of considering EPS aggregation, we also tested an extended model where a fraction of the excreted carbon is coagulated to EPS (xeps), contributing to the POC pool. We added following equation:

$$POC_{EPS} = x_{EPS}\, x_f\, POC$$

with a start value of 50% aggregation and no constraints (values between 0 % and 100 %). The difference between the extreme values of 0 % and 100 % are shown in Fig. 1 below and lead to 30 % difference in maximum POC. This makes the parameter quite insensitive. In fact the SensFUN of the FME package defines the sensitivity of the added xeps parameter close to 0. We also suggest, that the effect of xeps could be compensated by the SIPS term of the EXT model (% reduction in photosynthesis after Si reduction), leading to a very similar fit, indicating collinearity. This is mainly the cause since, EPS aggregation only has a major role with a linear response in the stationary phase when also Si is limiting. Thus, an additional xeps term would be unidentifiable with the current set of measured data.

We will add a more thorough discussion of the approach by Schartau et al. (2017) and the suggestion for a more simplified model extension described above (adding the xeps parameter), for experiments were EPS data are available, but with its limitations for the current model due to collinearity/unidentifiability issues.

[Figure]

Figure 1. POC concentrations of the measured data and model, including a model run showing POC as originally modelled POC + excreted DOC assumed to aggregate immediately and completely to EPS (dashed line).

2) Another finding by Khandeparker and Bhosle (2009) is an increased DOM excretion after Si and N limitation, which is not yet part of our model. Hence, we added a second excretion term xf2 after silicate limitation. A doubling of the excretion rate after silicate limitation leads to slightly reduced POC and PON values (Fi. 2), but no changes in Chl and only small extra NH4, due to the higher importance of the ambient DON for NH4 regeneration. The lower POC values can be completely compensated by doubling the the Photosynthesis after Si limitation (SiPS 0.2 -> 0.4) parameter. The lower PON value can almost be compensated with the same parameter.
This little difference indicate, that a modelling approach with changing xf rates after Si or N limitation is not necessary, at least not in our model system with high refractory DOM concentrations. It may however, become important in open-ocean systems with less terrestrially derived DOM.

[Figure]

Figure 2. Impacts of a 2 times increased DOM excretion after Si limitation (dashed line).

3) The biofilm can also facilitate interactions between bacteria and algae due to the closer proximity. This increased interactions could be represented in an increased remineralization rate of excreted organic matter (rem) after N or/and Si limitation. A potential model extension accounting for it would include a second higher remineralization rate after Si or N limitation. However, the difference between the EXT model where C is excreted as DOC (xf), or simply lost for maintenance respiration (RC) is minor (Fig. 3).

After adding a second remineralization rate of labile DOM (rem2) an increase of 3 orders of magnitude is needed (rem2 = 10e03 rem) to show any visible effect on N assimilation and NH4 regeneration (Fig. 3), showing that this is a highly insensitive parameter. However, about 4 order of magnitude higher rates appear to bring the modelled NH4 concentrations closer to the measured data (while fits to POC, PON, and Chl become very different form the measured data), hinting that the poor model fit to NH4 may not only be related to immobilized NH4 in the measured data (e.g. NH4 adsorbed or trapped in EPS), but may also be related underestimated DON excretion or remineralization in the model.

However, more than 10% DOM excretion and a DON remineralisation rate 3-4 order of magnitude higher than remineralization of the ambient, likely terrestrially derived DOM is rather unlikely. A likely explanation of the low impact of increased remineralization in the biofilm of our experiment is the high ambient DOM concentrations, which are the main DON source for NH4 remineralization (See difference of the extended model with and without excretion in the manuscript). Since our model is supposed to represent coastal systems, we thus argue that only 2 different remineralization rates related to refractory and labile OM is sufficient. In more open ocean setting with less allochthonous DOM input, increased remineralization rate of algae EPS in the stationary phase, may be a useful addition.

We suggest therefore, that a higher remineralization rate is likely, but that a large part of the remineralized NH4 is not available for algae growth due to the biofilm. Thus, the modelled NH4 values represent the available NH4 for algae, which representation is the aim of this study.

[Figure]

Figure 3. Comparison of the original model and a 3 orders of magnitude increased remineralization rate of excreted DON after Si limitation (dashed line).

We add Fig. 3 to the supplement and discuss the potential of increased bacterial remineralization in biofilms and why this is not quantitatively important in our experiment and model.

4) Another effect of the biofilm may be adsorption of ammonium to the EPS or concentration in pockets, not available for algae growth. In fact, this could be one of the explanation for the consistently high NH4 values in the stationary phase, which are poorly represented in the model (See response to Referee #2).

Response to Referee #2
"Ammonium is most likely immobilized in the biofilm via adsorption to the EPS and accumulation in pockets unavailable to diatoms. These immobile NH4 pools are still part of the measured data. With the model assuming all NH4 being available for algae growth, this is a problem."

This could, in particular, explain the high values of measured NH4 compared to the model results as shown in Figure 5c. In addition we could add a small discussion of a potential pH dependence

of NH4+ adsorption to the EPS in terms of the pKa values of NH4+ and carboxylic groups, which belongs to the acidic polysaccharides as a fraction of EPS:

- Carboxylic groups have a pKa < 5, i.e. far away from seawater pH ~ 8, which means that they are always in the deprotonized negatively charged form R-COO- in seawater.
- NH4+ has a pKa ~9 closer to seawater pH.
- Thus, the NH4+/NH3 ratio will be higher in more acidic microenvironments (pH ~7.5-8).
- Thus, a lower pH due to bacterial respiration would increase the concentration of NH4+ in comparison to the bulk medium, which results in a higher immobile NH4 pool due to adsorption to the EPS.
- This could explain the higher discrepancy between modelled and measured NH4+ values in the experiments with bacteria (as seen in Figure 5c).

Since the biofilm formation corresponds with silicate limitation, the reduced photosynthesis might of course be related to either the biofilm or the silicate limitation. But for untangling the effects of biofilm formation and silicate limitation, more experiments or data would be needed. However, only 30% of the culture was part of the biofilm and the best fit of an 80% reduction corresponds very well with an earlier study by Werner (1978), who did not have the issue of biofilm formation. Hence, we suggest that the main cause for the reduction of photosynthesis is related to Si limitation and not the biofilm.

We add this argumentation together with collinearity issues of SiPS with potential model extensions taking the biofilm into account to the discussion. We will also add the figures above showing the impact/sensitivity of potential model extensions to account for the changed dynamics in a biofilm to the supplement.
In the manuscript we add a discussion about the results explaining: i) the potential changes in a biofilm (increased DOM excretion, increased remineralization, trapped NH4), ii) the importance of the biofilm for our model run (POC as EPS instead of algae biomass, differences in fitting and sensitivity for the stationary or exponential phase, considerations of the biofilm being only 30%), iii) we also added a discussion of biofilms or aggregates/marine snow in the environment, which our study aims to represent.

Rakhee DS Khandeparker & Narayan B Bhosle (2001) Extracellular polymeric substances of the marine fouling diatom amphora rostrata Wm.Sm, Biofouling, 17:2, 117-127, DOI:10.1080/08927010109378471

The authors appropriately discuss quota models and their use. A different approach to model celluar nutrient kinetics, that has been argued to be more mechanistic, considers uptake sites for nutrients (Aksnes & Egge, 1991, Mar Ecol Prog Ser. 70:65-72). A good, though slightly technical, paper applying this approach and combining it with variable cellular stoichiometry is Flynn et al., 2018, PLoS Comput Biol 14(4): e1006118. Setting up a model like this for your data could be highly interesting, but beyond the scope of this study. However discussion the approach would provide a very useful context.

We want to thank the reviewer for the interesting suggestion and reference. We added the model by Flynn et al. to Table 1. We also included a discussion of the approach. We argue overall that the model is too complex for the aim of our study, which tries to keep the number of parameters as low as possible allowing scalability (similar to Flynn, 1997; Flynn, 2001), but acknowledge the important role of considering transporter densities, cell size, and mobility. Especially the importance of mobility is an interesting aspect, that we now discussed in the context of diatom sedimentation.

Change:

"The next step to quota based-models is the consideration of more detailed cell based characteristics, such as transporter density, cell size, and mobility, including sedimentation (Aksnes and Egge, 1991). Flynn et al. (2008) discuss a model with detailed uptake kinetics showing that large cells are overall in disfavored over small cells due to higher half saturation constant, but that they may still have competitive advantages due to lower investment in transporter production, and increased sedimentation, increasing the mobility that may offset the disadvantage of a larger size. While this extension is too complex for our aim of a simple model, the dynamics may become important when modelling different algae taxa."

In the introduction (line 46) and in the discussion the authors mention the role of the impact of climate change on coastal phytoplankton succession, including projected increased DOM inputs via river run off. Several studies have found and/or suggested a delayed bloom due to increase turbidity (e.g. Opdal et al. 2019, Glob Change Biol. 2019;00:1–8), which should be mentioned here.

We thank the reviewer for the suggestion, which is an important clarification to the introduction and in particular for the discussion.

Change in Introduction:

"… stratification in coastal Arctic systems is expected to increase… earlier stratified surface layer in spring, which may lead to an earlier spring bloom (Tremblay and Gagnon, 2009).""However, at the same time, brownification and increased sediment resuspension is already leading to light inhibition in spring, which may lead to a delayed spring bloom (Opdal et al., 2019)."

Change in Discussion:

"An earlier temperature driven water column stratification may also lead to an earlier bloom. However, due to increasing river and lake brownification and sediment resuspension, the spring bloom may be delayed (Opdal et al., 2019)." "With decreased light, carbon overconsumption as described by Schartau et al. (2007) may become less important due to decreased photosynthesis. An earlier, or later phytoplankton bloom can lead to a mismatch with zooplankton grazers (Durant et al., 2007;Sommer et al., 2007), which could decrease the fecal pellet driven vertical export and thereby increase the residence time of POM in the euphotic zone and the potential for ammonium regeneration, making the incorporation of bacterial recycling into ecosystem models even more imporatant as also evident from our experimental data and model output."

The authors mention both nitrate and ammonium as nitrogen sources. Additionally, urea can be a relevant nitrogen source in some systems. I am not sure how much of a role this plays in arctic ecosystems, but it should either be discussed or mentioned why it does not play a significant role.

We agree with the reviewer that urea may be another important nitrogen source, especially under nitrate limitation. In some Arctic systems it may reach concentrations of 2 uM. While bacteria may produce urea by ON degradation, the main source of urea is attributed to zooplankton excretion (Conover and Gustavson, 1999). Hence, it does not play a role in our experiment, but may play a role in nature. We added a discussion of urea as potential nitrogen source to the discussion of zooplankton NH4 excretion.

Change in introduction:

"Zooplankton may also release some ammonium and urea after feeding on phytoplankton, but we suggest that this process is likely far less important than bacterial regeneration (e.g. Saiz et al., 2013). Previously measured ammonium excretion of Arctic mesozooplankton is typically low compared to bacterial remineralization (Conover and Gustavson, 1999), with the exception for one study in summer in a more open ocean setting (Alcaraz et al., 2010). In some Arctic systems urea, excreted by zooplanotn may be an important N source for regenerated algae production (Conover and Gustavson, 1999)."

Change in discussion:

"Another potentially important N source for regenerated production may be urea (Harrison et al., 1985), which would lead to an even higher importance of regenerated production as suggested by our study."

Conover, R. J., & Gustavson, K. R. (1999). Sources of urea in arctic seas: zooplankton metabolism. Marine Ecology Progress Series, 179, 41-54.

Harrison, W. G., Head, E. J. H., Conover, R. J., Longhurst, A. R., & Sameoto, D. D. (1985). The distribution and metabolism of urea in the eastern Canadian Arctic. Deep Sea Research Part A. Oceanographic Research Papers, 32(1), 23-42.

Line 168: ": : :but the growth rate can be reduced (Hildebrand, 2002; Gilpin, 2004)". How can the growth rate be reduced? What can lead to this reduction?
We realized that growth rate is not the best term here and changed the sentence as follows:

Change:

"N and Si metabolism have different controls and intracellular dynamics, with N uptake fueled by photosynthesis (as PCref in G98) and Si mainly fueled by heterotrophic respiration (Martin-Jezequel et al., 2000). In general, we assume that nitrogen metabolism is not directly affected by silicate limitation (Hildebrand 2002, Claquin et al., 2002), but we expect cellular ratios to be affected by reduced photosynthesis and chlorophyll synthesis under Si limitation (Hildebrand, 2002; Gilpin, 2004)."

We also suggest to add more recent references on the effects of Si limitation on photosynthesis.

Gilpling (2004) only described the relationships of C,N,Chl production/assimilation under N and Si limition, but didn't give a physiological explanation.
Lippemeier et al. (1999) found a direct inhibition of the PSII reaction centre due to increased photochemical quenching, which is part of the explanation, but still rather descriptive. Our study confirmed lower efficiency of PSII (via Quantum yield measurements) after Si limitation, which is in accordance with Lippemeier et al. (1999) and supports our approach of reduced photosynthesis after Si limitation. Thus, we added a reference to this study in our discussion of the Quantum yield.
Another recent study by Liu et al. (2020) investigated gene expression patterns for C fixation related genes, and found reduced exression under Si limitation, but not under N or P limitation. The most detailed study has probably been done by Thangaraj et al. (2019), who used a metaproteomics approach and found not only downregulated photosynthetic proteins after silicate limitation, but also distracted protein production for mitochondria-chloroplast interactions, chlorophyll synthesis, and mechanisms compensating for disruption in electron transfer.

Lippemeier, S., Hartig, P., & Colijn, F. (1999). Direct impact of silicate on the photosynthetic performance of the diatom Thalassiosira weissflogii assessed by on-and off-line PAM fluorescence measurements. Journal of Plankton Research, 21(2).

Liu, Q., Xing, Y., Li, Y., Wang, H., Mi, T., Zhen, Y., & Yu, Z. (2020). Carbon fixation gene expression in Skeletonema marinoi in nitrogen-, phosphate-, silicate-starvation, and low-temperature stress exposure. Journal of Phycology, 56(2), 310-323.

Thangaraj, S., Shang, X., Sun, J., & Liu, H. (2019). Quantitative proteomic analysis reveals novel insights into intracellular silicate stress-responsive mechanisms in the diatom Skeletonema dohrnii. International Journal of Molecular Sciences, 20(10), 2540.

We added more details and references to the introduction, model description, and discussion.

Figure 6 and figure 7 do not exist.

We corrected the figure references.

Line 660: Table 1 is not the most up-to-date. Especially on the ecosystem model side it would be nice to see more recent developments reflected as well.

For the cultivation based models, we added the study by Flynn et al., 2018 as mentioned above. For the ecosystem scale models, we cited the original reference of the algae growth and potential nutrient regeneration dynamics, which is often rather old, while the full-scale models are mostly updated in terms of mostly physical formulations. We clarified this in the legend and added following more recent references to the ecosystem scale models:

BFM model:_ Smith, K. M., Kern, S., Hamlington, P. E., Zavatarelli, M., Pinardi, N., Klee, E. F., & Niemeyer, K. E. (2020). BFM17 v1. 0: Reduced-Order Biogeochemical Flux Model for Upper Ocean Biophysical Simulations. *Geoscientific Model Development Discussions*, 1-35.

ReCom-2 model: Schourup-Kristensen, V., Wekerle, C., Wolf-Gladrow, D., Völker, C. (2018): Arctic Ocean biogeochemistry in the high resolution FESOM 1.4-REcoM2 model, Progress in Oceanography, 168, 65-81,doi:10.1016/j.pocean.2018.09.006.

MEDUSA model: Henson, S. A., Cole, H. S., Hopkins, J., Martin, A. P., & Yool, A. (2018). Detection of climate change-driven trends in phytoplankton phenology. *Global Change Biology*, *24*(1), e101-e111.

NEMURO model: Anju, M., Sreeush, M. G., Valsala, V., Smitha, B. R., Hamza, F., Bharathi, G., & Naidu, C. V. (2020). Understanding the Role of Nutrient Limitation on Plankton Biomass Over Arabian Sea Via 1-D Coupled Biogeochemical Model and Bio-Argo Observations. *Journal of Geophysical Research: Oceans*, *125*(6), e2019JC015502.

SINMOD model: Alver, M. O., Broch, O. J., Melle, W., Bagøien, E., & Slagstad, D. (2016). Validation of an Eulerian population model for the marine copepod Calanus finmarchicus in the Norwegian Sea. *Journal of Marine Systems*, *160*, 81-93.

NPZD model: Gruber, N., Frenzel, H., Doney, S. C., Marchesiello, P., McWilliams, J. C., Moisan, J. R., Oram, J. J., Plattner, G., and Stolzenbach, K. D.: Eddy-resolving simulation of plankton ecosystem dynamics in the California Current System, Deep Sea Research Part I: Oceanographic Research Papers, 53(9), 1483-1516, 2006.

Especially in the abstract and the introduction there are several long (sometimes convoluted) sentences. To increase readability it would be could to rephrase these (Schachtelsaetze sind im Englischen nicht so hoch angesehen wie im Deutschen ;) ).

We splitted the long sentences into shorter easier to read sentence in the revised version.

---

## Author Response (AR2)

First of all, I really appreciate the effort that the authors went to in responding to my overlong review. On the whole, I'm happy with the changes that they've made to the manuscript. However, I do still have some concerns with a few of their responses. I've listed the main, overarching ones immediately below, but then give more specific comments to particular responses afterwards.

We want to thank the reviewer for the positive feedback and highly appreciate the additional comments and suggestions for clarifications. We addressed all comments as outlined below and believe that the changes improved the manuscript considerably.

Some of the comments pointed to weaknesses of the model itself that we corrected (e.g. How labile and refractory DON are separated, how bacteria biomass is incorporated, Chl degradation after Si limitation). Due to the changes we repeated the model tuning and got slightly different parameters, and fits. Overall, the quality of the model fits stays very similar and the discussion stays overall the same, but the model is now clearer, more realistic and applicable on a wider scale. Some weaknesses of the models that we needed to discuss disappeared (lack of Chl degradation, low sensibility of excretion and remineralization parameters). Because of these changes we updated the parameter, sensitivity analyses and collinearity test tables and updated all plots showing model fits. We also updated all error estimates in the text, and removed weaknesses of the model, that were previously pointed out in the text, but which are now not apparent anymore.

The missing equations and state variables for labile and refractory DON pointed to an assumption in our model that excreted labile DON is quickly converted to refractory DON. We realized that this quick conversion is unrealistic and changed it to 2 separate DON pools (See equations below), where labile DON does not aggregate to refractory DON during the time of the experiment. This change led to a higher sensitivity of the DON degradation parameters and a worse fit of the EXT-exr model. This points to a higher importance of autochthonous DOM degradation compared to allochthonous degradation, which is more realistic. The finding of our previous version was the opposite, which we previously discussed as a potentially problematic simplification in our study.

The higher sensitivity of the excretion and remineralization parameters also lead to a higher importance of EPS aggregation that was previously pointed out by reviewer 3. Thus, we re-evaluated the model extensions of the supplement and can now state that EPS degradation is indeed a potentially important process as suspected by reviewer 3. When considering EPS aggregation we now get a better fit than possible without it. The extension with increased excretion in a biofilm shows the same result as before (Full compensation/collinearity by/with PSSi). We removed the 3rd extension of increased NH4 regeneration due to the problems of the NH4 data pointed out by this review. There may be more NH4 regeneration in the biofilm, but at the same time more immobilization. So the trend can go both ways and a model assuming increased remineralization is unrealistic.

The lack of Chlorophyll degradation of the EXT model points simpy to a missing formulation in one of the model equations. dChl/dt = 0 (under Si limitation), which should be dChl/dt = 0 − RChl * Chl (RChl, being Chlorophyll degradation), which is modelled in all other Chl synthesis related equations of the model. After the correction, the Chl degradation of the EXT model is now well represented. In the previous manuscript, this absence of degradation was pointed out as a weakness of the model. Hence, the correction strengthened the model and we could remove the statement that Chl degradation is not well modelled.

The confusion about the units of the bacterial carrying capacity pointed us to a more realistic approach of giving bacteria biomass (and carrying capacity) in Carbon units, which makes their abundances directly comparable with algae POC and biomass and the model easier to understand and replicate. We also added the logistic growth curve fit to the bacteria biomass in the supplement.

The starting conditions (the state variables) were indeed not clear. We now clarified that the experiments start at day 1 (excluding day 0 where we assume artifacts directly after the transfer of cultures). For NOx and Si we do not have measurements for day 1, but only for day 0 and 2. We now clarified that we use the mean data between day 0 and day 2 as starting condition. After adding the information to the methods, with the experimental data in the plots of the manuscript and the raw data being publically available on Dataverse, the process is now transparent, clear and repeatable.

Some of the explanations for the experimental data still do not make sense (e.g. occurrence of a

PO4 spike). Explanations should be consistent between the axenic and non-axenic cultures, and the authors should not be afraid to note where a feature in the data appears inexplicable (i.e. is more likely an experimental / measurement error).

Concerning the high experimental NH4 values, we explained in-depth, why we suggest that the issue lies with immobilized ammonium that is released via filtration (and thereby part of the measured NH4), while being unavailable for the diatoms (immobilized in EPS). This is indeed an experimental issue as we also mention in the revised manuscript.

Concerning the phosphate peak we agree that the sudden increase of 100% can not solely be explained by reduced Diatom uptake if PO4 while bacterial PO4 remineralisation increases. We also do not have an explanation for the sudden drop at the last day. Since PO4 is not part of the model, we do not consider these unexplained variations to be a problem for the model and manuscript. However, we now acknowledge these patterns as mostly inexplicable in the manuscript as suggested by the reviewer (See below). We agree that it is better to not data/measurement issues that we cannot fully explain, than only discussing potential problems that may explain part of these inexplicable PO4 changes.

We also clarified the outliers in the figures more clearly by excluding them from the polygons and simply marking them as asterisks. Some of the concerns are related to single outlier values (e.g. sudden Chl drop at day 8, large variance of phosphate at day 14). The figures are now much clearer and less misleading (outliers are apparent without the need to read the legend).

The model equations are still not complete as far as I can see. There are also some cosmetic issues around the use of (unexplained) constants in the equations – it would be better to assign (fixed) conversion parameters.

We are very grateful for the effort of RW2 to check our model equations in detail. Some equations were indeed not complete and/or unclear.

We now added the missing DON equations separating labile and refractory DON and the conversion of DOC (which we measured) to DON (which we used in the model). The constants are now parameterized before the equations (Redfield ratio, Molar mass of nitrogen).

We also corrected the equation about logistic growth of bacteria and now use Carbon units instead of cells, which makes it more quantitative and comparable to other model parameters. For Chl degradation after Si limitation, we added the term dChl/dt = - RChl * Chl, which now allows us to model the degradation in the stationary phase.

The explanation of the tuning method is much appreciated. However, as it stands, it's very dense text that's been included. It might be better to add a supplementary section that breaks this up into bullet-pointed steps – that would be ugly for the main body, but would probably be very helpful as a supplement interested readers could consult.

We thank the reviewer for the suggestion and added such a supplementary section with bullet point in the Supplement.

While the authors have rephrased their hypotheses, they still make no use of them, nor even refer to them, elsewhere in the manuscript. Plus, and especially because of the way they're used, these read more like conclusions of the work than hypotheses that the paper actively tests.

We do return to the hypothesis but we realized that we did it in a very intertwined way with no clear references back to the specific hypothesis. We agree that it should be more clear and suggest additional references to the hypotheses in the discussion. We now added clear references back to the hypotheses were it was useful. (See details below)

Overall, I think that the manuscript is now close to being acceptable once these remaining details are addressed. I would advise accept after minor revisions.
* * *
Ln. 161, "The strong spike of phosphate after day 8 corresponds with the end of the exponential phase for algal growth and a spike of ammonium. At the same time bacteria abundances start increasing considerably. Thus, we explain the phosphate peak by increased bacterial regeneration (source of phosphate) and decreased algal uptake (sink of phosphate) at the same day." - *Qualitatively* I can see this line of argument, but it doesn't really stack up *quantitatively*. First, the timing's not quite right, with ambient PO4 jumping ~50% on day 8, when phytoplankton are at their peak and bacteria have only just got moving. - It would be more convincing to me if there was an attempt to budget the elements to make this shift between the medium, the phytoplankton and the bacteria clearer; it remains difficult for me to shake the impression that something has gone wrong in the experiment; and this should be done for both experiments and models; among other things, if it can't be done for whatever reason, it does tend to suggest that, contrary to the authors "hypothesis 3", experiments do not provide a good testbed for ecosystem functionality

Concerning the phosphate peak we agree that the sudden increase of 100% can quantitatively not solely be explained by reduced Diatom uptake if PO4 while bacterial PO4 remineralisation increases. We also do not have an explanation for the sudden drop at the last day. We can not budget the etire P pool within bacteria P, algae P and PO4 since a large part is within the DOM pool. Mineralization processes may however be another explanation. Since PO4 is not part of the model, we do not consider these unexplained variations to be a problem for the manuscript or model. We now acknowledge these patterns as mostly inexplicable in the manuscript.

The mismatch of NH4 and PO4 remineralisation may be caused by N being more limiting (more N retained by bacteria than released as NH4) and PO4 not (less PO4 retained and more excreted via remineralization), but our data do not allow to answer this hypothesis conclusively.

Changes: L472f in the revised MS: "After NO3 depletion at day 15, also PO4 concentrations drop, indicating a coupling of NO3:P metabolism, but not of NH4:P metabolism. Thus, the sudden drop may also indicate dynamics of bottle experiments, not accounted for, showing potential limitations of these experiments. "

Changes: L467f in the revised MS: "With the start of the stationary phase, NH4 and PO4 concentrations doubled, presumably due to decreased assimilation by the silicate starved diatoms and increased regeneration by bacteria, supplied with increasing labile DOM (doubled remineralisation rate in EXT) excreted by the stressed algae. However, NH4 concentrations double in 4 days, while PO4 concentrations double in only 1 day, indicating some unexplained internal dynamics, potentially via different bacterial uptake and release of N and P."

Changes: L511f in the revised MS: "Overall, our cultivation experiment is powerful to represent some major spring bloom dynamics, but has its limitations, thereby confirming our third hypothesis only to some extent."

Ln. 183, "Excretion of organic phosphate by diatoms is also common in cultures with surplus orthophosphate (Admiraal and Werner, 1983), which can be another explanation of the phosphate peak after silicate becomes limiting." - but it only happens in the case with bacteria, so this doesn't make sense; if anything the case without bacteria should be more stressed because there's no nutrient regeneration

We do see a spike in both treatments. However, we agree that the spike in the axenic culture is much less pronounced and removed the argument for streamlining the manuscript and to avoid confusion.

Ln. 193, "Figure 4, 5, B1) "...Chlorophyll a concentration in experimental cultures with a potential outlier at day 8, presumably due to photodegradation, causing a negative spike." - "photodegradation"? it looks like a straight error to me as it bounces right back; if you are going with photodegradation, expand on this

We assume indeed that the outlier is a measurement error. We think that this specific sample might have been exposed to light during the extraction, which degrades Chlorophyll and leads to lower values. So, yes, it is a straight error. We avoided simply removing outlier values of measured data from plots, simpy because they do not fit to our (or the models) expectations, which we consider

data manipulation. However, we acknowledge that this data point appears misleading. In fact, it is not clear that it is based on an outlier without reading the entire legend. We show all outliers now as an asterisk in the plot, while excluding it from the area of the polygon.

Changed legend: "Figure 4, 5, B1) "…Chlorophyll a concentration in experimental cultures (the asterisk indicates a presumed measurement error)."

[Figure]

**(d)**

Ln. 202, "We added the missing equation and double-checked for any other incomplete model descriptions." - the equations for DOM are still inadequate; background DOM is listed as a state variable but its equation is omitted

We did not list DOM as state variable, but assume the reviewer refers to the DON state variable. We are indeed lacking the conversion from DOC (measured variable) to DON, which we estimated via the Redfield ratio: DON = DOC / 16.

We did following changes in the appendix tables:

- We added the state variable DOC in table A1
- We defined the redfield ratio (RR) C:N = 106/16 in as parameter in table A3
- We added following equation to table A7: DONr = DOC /RR

We also realized that we did not differ between DONr and DONl in the equations in table A7 and corrected the equations in the following way (Replacing the number 14 with the parameter molar mass of N (MN) as suggested below). The previous model also had an assumption of labile excreted DON becoming refractory shortly after it is released to the medium, which we realized is not realistic without including a 3rd DON pool. Thus, we adjusted the equations in order to keep all excreted DON labile throughout the experiment. The model changes required a new model fitting, plotting, and tuning. The results are nearly the same.

Changes in table A7:

| 8) | Ammonium uptake and production | $IF\left(\dfrac{C}{N} < 10\right)$ |
|---|---|---|
| | *(Threshhold after Tezuka 1989, and Gilpin 2004)* | $\dfrac{dNH4}{dt} = \dfrac{-\left(\dfrac{V_{NH4}^{C}}{Q}\right)N + Bact\ DONl\ rem + Bact\ DONr\ rem_{d})}{M_{N}\ 10^{3}}$ |
| | | $ELSE\ IF\ (NH4 > nh4_{thresj}$ |

$$\frac{dNH4}{dt} = \frac{-\left(\frac{V_{NH4}^C}{Q}\right)N - \frac{dBact}{dt\,RR})}{M_N\,10^3}$$

$$ELSE$$

$$\frac{dNH4}{dt} = \frac{-\left(\frac{V_{NH4}^C}{Q}\right)N)}{M_N\,10^3}$$

9)   DON uptake and production

$$IF\left(\frac{C}{N} < 10\right)$$

$$\frac{dDONl}{dt} = \frac{-Bact\,DONl\,rem + xf\,N - \frac{dBact}{dt\,RR}}{M_N\,10^3}$$

$$\frac{dDONr}{dt} = \frac{-\,Bact\,DONr\,rem_d}{M_N\,10^3}$$

$$ELSE$$

$$\frac{dDONl}{dt} = \frac{xf\,N}{M_N\,10^3}$$

$$\frac{dDONr}{dt} = 0$$

Ln. 206, "Equation 7a) Silicate uptake" - working through the units in this equation, using the given units for state variables and parameters, I find that there's a discrepancy of 3 orders of magnitude; I think one of the sets of units must be listed incorrectly

Vmax = (mol Si / d) / (mg C)
Sid = umol / L
Smin = umol / L
Ksi = umol / L
C = mg C / m3

dSid / dt = (Vmax * Sid * ((Sid - Smin) / (Ksi * Smin)) * C
= ((mol Si / d) / (mg C)) * umol / L * (umol / L) / (umol / L * umol / L) * mg C / m3
= ((mol Si / d) / (mg C)) * ((umol / L)^2) / ((umol / L)^2) * (mg C / m3)
= ((mol Si / d) / (mg C)) * (mg C / m3)
= mol Si / d / m3
= mmol Si / L / d
≠ Sid ≠ umol / L

We are thankful for the reviewer found this error and corrected the unit for Vmax to: (umol Si/ d) / (mg C). as mentioned below we also changed the units of C,N, and Chl to mg L-1

dSid / dt = (Vmax * Sid * ((Sid - Smin) / (Ksi * Smin)) * C
= ((umol Si / d) / (mg C)) * umol / L * (umol / L) / (umol / L * umol / L) * mg C / L
= ((umol Si / d) / (mg C)) * ((umol / L)^2) / ((umol / L)^2) * (mg C / L)
= ((umol Si / d) / (mg C)) * (mg C / L)
= umol Si / d / L
= umol Si / L / d
= Sid / dt = umol / L / d

Ln. 207, "Equation 7b)" - the 14 comes from nowhere here; you should make it clear that it's a molar mass

We agree and added the term $M_{Si}$ as parameter in table A3 and replaced 14 in eq 7b with $M_{Si}$

Ln. 235, "Figure 1"
- what is meant by control here?; these lines in particular make this diagram difficult to follow; when I suggested a diagram it was to (a) make clear what the state variables are, and (b) indicate the relationships between them so that the simple food web in the experiment vessels was; this diagram is overcomplicated, while also seeming to omit detail (e.g. bacterial metabolism)
- in this diagram the bacteria - which are meant to be heterotrophic - consume only nitrate; that doesn't make sense biogeochemically

We agree that the diagram is too complex with the lines indicating the controls and suggest removing the controls and simplifying the diagram by simply showing the state variables and how they are connected. Regarding bacteria we add an unmodelled DOC pool to make the diagram biogeochemically more meaningful.

[Figure]

Ln. 258, "2.3 Modelling fitting"
- while thorough, this is very difficult to parse. I would suggest writing it out clearly as a bullet pointed step-by-step process (you've actually begun this in your response elsewhere), and adding this description to the supplementary material. Make clear which models, which parameters, which target datasets and whether tuning was manual or automatic
- On a separate note, I don't think initial conditions used are covered; and some of the plots suggest this is not done as one might expect; e.g. Figure 6 seems to show the model starting at values quite different from the experiments

We agree that the "Modelling fitting" chapter became quite detailed after considering all suggestions from the first review round and added a step-by-step process as bullet points to the Supplement. Since the bullet points do not include the justifications for the different approaches, we do not think that they can replace Ch. 2.3, but we agree that it helps to understand the approach more clearly.

We added following chapter to the Supplement:

Model tuning protocol (short version)

1. Fitting of the G98 model to the BAC- experiment
    1.1. Model programming (R)
    1.2. Model solving (ode function, Runge-Kutta method, deSolve package)

Concerning the separate note, the model starts at day 1 in order to avoid artifacts caused by the stress of transfer. For POC,Chl,PON and NH4 the values were measured at day 1 and this was straightforward. NO3 values were measured at day 0 and day 2, but not directly at day 1. Hence, we used the mean between the values of day 0 and 2 as start NO3 values for the model (Separate calculations for BAC- and BAC+ -> different starting conditions). Fig 6 still shows the values at day 0 of the measured data, since we argue that it helps to evaluate the overall fit more thoroughly. We added the information (including why we start the model at day 1) to chapter 2.3.

Change: L270f in the revised MS: "The model fitting started at day 1 in order to avoid artifacts during acclimation of the cultures after transfer to a new medium. Si and NOX were not measured at day 1 and the mean of day 0 and day 2 was used."

Ln. 381, "2. Diatoms continue photosynthesis under silicate limitation at a reduced rate if DIN is available"
- what does this mean for cell division?; if not here, you should certainly comment on this somewhere

We added the information more specifically in the discussion:

Change: Line 526f in the revised MS: "Si is only needed for frustule formation and cell division, mostly during a specific time in the cell cycle (G2 and M phase,…)…"

Ln. 389, "Ammonium is most likely immobilized in the biofilm via adsorption to the EPS and accumulation in pockets unavailable to diatoms"
- this needs a bit of expansion - the model has less NH4 than the observations, and if it was being immobilised in the experiments within the biofilm (something that's absent in the model), one might instead expect lower concentrations, no?

We suggest that the immobilized ammonium is not available for diatoms, but that it will be part of the measurements after release of the ammonium during filtration and pH changes. Thus, we measure ammonium which is not available for diatoms, while the model only considers bioavailable ammonium. In the model, ammonium is taken up preferably and quickly after its release keeping the ammonium levels low, while we suggest that this is not possible for immobilized ammonium. Consequently, the model will assume more ammonium uptake and less NO3 uptake.

- also, it's not clear which of this text has made it into the revised manuscript

We added this information of the revised manuscript submitted already after the first round of reviews in the following sections. We also added a few more details as suggested in this review (marked in yellow below):

Ch 2.3: "The model was only fitted to total DIN, due to the potential uncertainties related to ammonium immobilization in the biofilm, which is released during filtration and part of the measured data."…" The main effect of the biofilm that we could not model with the available data appears to be ammonium immobilization in the biofilm, either due to adsorption, accumulation in pockets, or conversion to ammonia due to the locally reduced pH caused by increased bacterial respiration."

Ch 3.1: "While not all ammonium measured is also available for algae growth, discussion of the dynamics (decrease in the start, increase with the onset of the stationary phase), especially if also shown in the EXT model, are still useful to understand multinutrient dynamics (e.g. regeneration)"

Ch 3.2: "This indicates that the problem lies with the ammonium data, which include immobilized ammonium in the biofilm, unavailable for diatoms growth, while the model assumes that all ammonium is available."

Ln. 520, "In previous cultivation experiments, no efforts for obtaining axenic cultures were mentioned, which hints to bacteria contaminated cultures."
- I take the point; from my own experience of the tedious nature of maintaining axenic cultures, it's not unreasonable to assume that a study where this isn't mentioned is probably not working with axenic cultures

We thank the reviewer for the acknowledgement and personal experience as support for our suggestion.

Ln. 601, "Microscopy showed bacteria attached to the diatoms (mostly in the stationary phase), but mostly free-living."
- as the audience of this paper may be unclear on this point, it would be good to add a sentence noting your microscopy results so that readers understand you are largely referring to free-living bacteria

We added the information to the text: Line 120 in the revised MS: "…inoculation with mostly free-living bacteria cultures, isolated beforehand from the non-axenic culture."

Ln. 621, "As mentioned above, we changed the hypotheses in the following way:"
- you still don't return to them!; normally, hypotheses are stated ahead of work then reassessed at the end of the work; these are almost conclusions that are appearing in the introduction!

We do return to the hypothesis but we realized that we did it in a very intertwined way with no clear references back to the specific hypothesis. We agree that it should be more clear and suggest following additions in the discussion, which mainly add a clear reference back to the hypotheses were it was useful.

In the previous version we summarized how each hypothesis was confirmed in the study as introduction to the Discussion: "The experimental incubations represented typical spring bloom dynamics for coastal Arctic systems, including an initial exponential growth phase terminated by N and Si limitation and the potential for an extended growth period via regenerated production."

We changed this introduction in the following way to make it more clear that it refers back to the hypotheses: "The experimental incubations showed that in the presence of bacteria both the growth period and gross carbon fixation can be extended (Hypothesis I). The diatoms were able to continue photosynthesis under silicate limitation at a reduced rate as long as silicate was present (Hypothesis II). Overall, the incubations represented typical spring bloom dynamics for coastal Arctic systems, including an initial exponential growth phase terminated by N and Si limitation (Hypothesis III) and the potential for an extended growth period via regenerated production."

For making the red line and relevance of the hypothesis even clearer we also reference back to the hypothesis in the more detailed discussion.

L. 459-462: "==As suggested by our second hypothesis, p==hotosynthesis was reduced by approx. 70% after silicate became limiting, which is comparable to earlier experimental studies (Tezuka, 1989). However, ==as suggested by our first hypothesis==, the secondary bloom was extended in time by bacterial regeneration of ammonium, allowing regenerated production to contribute about 69% of the total production (f-ratio=0.31) …"

L469f: "The presence of bacteria and thus regenerated production allowed diatom growth to continue 8 days after silicate became limiting (Figs. 2, 3 & 4), nearly doubling the growth period similar to observations in the field (e.g. Legendre and Rassoulzadegan, 1995; Johnson et al., 2007==), which supports our Hypotheses I and III==."

L481f: "While we do not expect the f-ratio in our bottle experiment to be directly comparable to open ocean system, which does include a variety of algal taxa beyond C. socialis, a comparison can aid to identify limitations in our experiment and model. ==The f-ratio also allows a discussion of how representative the cultivation study is for typical spring bloom dynamics (Hypothesis III)==."

L500f: "Hence, ecosystem scale models will need to consider these dynamics regarding bacterial abundances, microbial networks and particle export in addition to bacterial remineralization in order to model realistic ammonium regeneration in the euphotic zone. ==Overall, our cultivation experiment is powerful to represent some major spring bloom dynamics, but has its limitations and thereby confirming our third hypothesis partly.=="

Ln. 754, "Thus, we implemented a labile (DONl) and refractory (DONr) DON pool with different remineralization rates (rem, remd)."
- as noted the equations for the DONr pool are not included

We added the equations as mentioned above.

Ln. 760, "We do not suggest a complete stop of remineralisation …"
- OK - this sounds more reasonable

Thanks for the confirmation.

Ln. 766, "DOM C/N mass ratio"
- why use "mass ratio" when "molar ratio" is more common?

We changed the definition to ==molar ratio== including line 775.

Ln. 775, "We refer to DOM as substrate for bacteria and clarified it: DOM C/N ratios…."
- Please be clear if C:N is mass or molar throughout if you're going to chop and change

We changed the manuscript in a way that all ratios relate to molar ratios, unless given otherwise (gC:gN). gC:gN ratios are mostly needed for the modelling and kept for comparability of parameters with earlier fits of models relating to the G98 construct. In the methods we still mention "molar C/N ratios" (e.g. molar C/N ratio <10) while we do not add this information, when we are writing about ratios in general. A "higher" molar C/N ratio has the same implications as a "higher" mass C/N ratio. What matters and what needs to be clear are ratios with values given. We proof-read the text and clarified it where needed.

Ln. 884, "Collinearity is a measure for the parameter identifiability …"
- this is good, but has it made it into the text?

Yes it is part of ch 2.3: "Prior to the automated fitting, parameters were tested for local sensitivity (SensFun) and collinearity, or parameter identifiability (collin; e.g. Wu et al., 290 2014)."…" The sensFun output is further used as input for the collinearity, or parameter identifiability analyses. Parameters were considered collinear and not identifiable in combination with a collinearity index higher than 20 (Brun et al., 2001). In this case, only the more sensitive parameter was used for 295 further                                                                                                                                  tuning."

Ln. 975, "due to the potential uncertainties related to immobilized ammonium"

- as noted above the immobilisation is curious given that the measurements include this extra NH4; is the NH4 only effectively immobilised for the phytoplankton such that when the vessel is stirred, it becomes apparent?

Yes, as mentioned above, we suggest that the ammonium is only immobilized for the algae, but released during filtration (vacuum disintegrating EPS structure, change in temperature and pH). As mentioned above this information is now given more clearly in the text as well.

Ln. 1129, "The R code is now available at github"
- if you want to create a permanent record of the model with a DOI, consider using the Zenodo link from Github; this will preserve a version of the exact version used here
- if you do this, make sure to add the DOI to your final manuscript revision

We than the reviewer for the valuable suggestion and archived the code permanently on zenodo. The doi number is given under data availability in the manuscript.

Ln. 1154, "We argue that the large range is plausible since it is i) …"
- on point (i), outliers are usually excluded because they are suspected of being false (e.g. measurement error); on point (ii), you're talking about the size of the range of PO4 on day 14, when there's literally zero range of bacterial variability at day 14 - this doesn't make sense

We now excluded outliers from the polygons and marked them as an asterisk to avoid confusion. (See comment on Chl above).

Regarding point 2 we overall agree and did not mention it in the manuscript. Instead we acknowledge some inexplicable variation pointing to limitations of the experiment as suggested earlier by the reviewer.

Ln. 1165, "As mentioned in the results, the bacteria growing towards the end are still in so low …"
- that's OK - it's tough keeping things axenic

We thank the reviewer for the confirmation

Ln. 1192, "we added the output of a prolongated model run in the supplement"
- thanks!

We are thankful for the helpful suggestion.

Ln. 1280, "We shortened the table slightly and explained all columns in detail in the corrected version"
- the mean, min and max columns are still not properly explained; for instance, why are there negative values?

Since the information about the sensitivity ranges is not needed for the manuscript we removed the last columns (min, mean max). Negative or positive values indicate the direction in which the output variable changes after parameter perturbations (decrease or increase).

Ln. 1313, "We changed the form of eq 14 to the same format as in eq 15"
- OK; but the use of (unexplained) numbers in these equations rather than parameters remains unhelpful

As mentioned above we replaced the numbers with parameters that we explain now in table A4 (e.g. RR (Redfield C:N ratio) = 16; MN (Molar mass of nitrogen) = 14)

Ln. 1345, "Table A8"
- this table is still confusing; do the rows need identifying labels?

The rows indicate which set of two parameters were tested for collinearity. For consistency with other tables in this paper, we replaced 0 with X (for absence) and 1 with V (for presence). We also added

a vertical line before the last column to make clear that the first columns are part of a matrix and the last column is the result (Collinearity index).

- also, what's with the strange number more than 1 million?

A collinearity of more than 1 million means that these two parameters are highly collinear and clearly unidentifiable. The parameters are I (light) and alphaChl (C assimilation per Chl and light), which shows already in their units that a change in light can be almost fully compensated by a change in alphaChl. With the threshold of 20 given by Brun et al. we argue that the table header is sufficient for interpreting this high value as unidentifiable parameter combination.

Changed table:

| Parameter combinations | | | | | | | | | | |
|---|---|---|---|---|---|---|---|---|---|---|
| $\zeta$ | $R^C$ | $\theta^N_{max}$ | $Q_{min}$ | $Q_{max}$ | $\alpha^{Chl}$ | I | n | $K_{no3}$ | $P^C_{ref}$ | collinearity |
| V | X | V | X | X | X | X | X | X | X | 27 |
| V | X | X | X | V | X | X | X | X | X | 98 |
| V | X | X | X | X | V | X | X | X | X | 31 |
| V | X | X | X | X | X | V | X | X | X | 31 |
| V | X | X | X | X | X | X | V | X | X | 93 |
| X | V | X | X | X | X | X | X | X | V | 25 |
| X | X | V | X | V | X | X | X | X | X | 34 |
| X | X | V | X | X | V | X | X | X | X | 101 |
| X | X | V | X | X | X | V | X | X | X | 101 |
| X | X | V | X | X | X | X | V | X | X | 38 |
| X | X | V | X | X | X | X | X | X | V | 32 |
| X | X | X | X | V | V | X | X | X | X | 40 |
| X | X | X | X | V | X | V | X | X | X | 40 |
| X | X | X | X | V | X | X | V | X | X | 146 |
| X | X | X | X | X | V | V | X | X | X | 455473 |
| X | X | X | X | X | V | X | V | X | X | 46 |
| X | X | X | X | X | V | X | X | X | V | 28 |
| X | X | X | X | X | X | V | V | X | X | 46 |
| X | X | X | X | X | X | V | X | X | V | 28 |

Figure 2

- I'd not really paid attention before, but the experiments have a large span of initial nutrient concentrations - shouldn't this be properly controlled?; or am I misunderstanding the plots?

We agree that the variation of the nutrient concentrations are quite high, but that are the data we measured. However, since our model starts at day 1 and not day 0 (Due to expected stress responses in the 1st day), we argue that these variations do not have strong implications for the discussion of the manuscript.

- (and thinking back the point made previously) not only does PO4 increase inexplicably between days 6-10 (+100%), it also drops off a cliff between days 14-15 (-80%)

See comments above were we acknowledge some inexplicable internal dynamics in the manuscript.

- your plots consistently run out of x-axis; the experiments are at least 15 days long, but your axis stops at 14 days

We extended the x axis to 16 days

- Panel d runs out of y-axis (as well as x-axis)

We also extended the y-axes were needed

Other changes based on the comments above include that outlier are excluded from the polygons and marked with an asterisk

Figure 5
- nothing about bacterial biomass and how it fares when excretion is included / omitted

Modelling bacteria biomass is not the main objective of the paper. As described earlier we modelled bacteria biomass production via a logistic growth curve. DON does not directly affect the fitted growth curve. As described in our previous response, we do not think that an added complexity of bacterial growth modelling is justified by the data or aim of the paper. Thus, we do not think that adding the effect of DON excretion on bacterial growth would add value to the manuscript, but add complexity with rather poor support in the model. See also our response to the first review:

 "The main improvement of the model is to include a remineralisation rate controlled by: i) bacteria biomass, ii) substrate (DOM) C:N ratios, and iii) substrate origin (autochthonous, allochthonous). Other models typically have a fixed remineralisation rate either only dependent on the DOM/POM, or not controlled by any environmental variable. Thus, we still see our extension as a considerable improvement and consider a simple logistic growth estimate sufficient.

We could of course model bacteria growth via Michaelis-Menten kinetics based on 2 DOM pools, but this would not have any effect on the parameterization or modelling of algae physiology, which is the main goal of the paper, while increasing the number of parameters and computational costs, which we 790 tried to keep low. Since, the aim of the model is not to model bacteria growth, but algae growth and intracellular C:N:Chl ratios we do not see that a more accurate and more complex model of bacteria growth would improve the manuscript."

However, we acknowledge that the fit of the logistic growth curve may be of interest and added the plot to the supplement.

- chlorophyll doesn't decline here but it should; and it does in a later supplementary plot (B1c); something doesn't seem quite right.

We thank the reviewer for the observation. In fact, the formulation of a stop in Chl production after Si limitation was the problem. In the previous version the formulation was $dChl/dt = 0$, while it should be $dCh/dt = - RChl\ Chl$ (maintenance loss of Chl). We corrected the corresponding equation in the manuscript and modelling code and rerun the fitting routine for EXT.

Figure 6
- why do the model lines start at day = 1 and not day = 0?; the observations seem to start at day = 0 by contrast (and unlike in other plots) also, in the BAC- experiment, the model seems to start at a lower value than it did in the experiments

See comments above: "the model starts at day 1 in order to avoid artifacts caused by the stress of transfer. For POC,Chl,PON and NH4 the values were measured at day 1 and this was straightforward. NO3 values were measured at day 0 and day 2, but not directly at day 1. Hence, we used the mean between the values of day 0 and 2 as start NO3 values for the model. Fig 6 still shows the values at day 0 of the measured data. We added the information (including why we start the model at day 1) to chapter 2.3.".

Fig 6 in the previous version also included only NH4 data for the days where NO3 data are available, which lets the start of the model appear out of place. We now added all measured NH4 data points to the plot for clarification.

We also double-checked all starting values with our measured data.

- the behaviour of modelled ammonium is still confusing; it seems to have very little to do with what the experiments did;

Considering the overall poor fit to measured NH4 values, we discussed the effect of NH4 immobilization and release via filtration in detail in the responses above. In summary, we suggest Nh4 immobilization in the biofilm (unavailable for diatoms) and release during filtration (available for the nutrient analyzer). We added this information as described above (including a straight forward acknowledgement of the data limitations).

Table A2
- what does "mio." mean?; does it mean "million"?; it's not an abbreviation I'm familiar with; and, if so, it looks like bacteria reach concentrations of 60 million cells per mL, but the value for bact_max ranges 0.005-0.1 million cells per mL; this is confusing

The reviewer pointed out an error in the units that we missed and now corrected in the manuscript. We also double-checked all other units for consistency and solvability in the given equations. For consistency, we now adjusted the bacteria cell numbers, and carrying capacity for the model into carbon units (mgC L-1) in order to make them comparable to the C,N, and Chl units of the diatoms (20 fg C per cell). We did following changes in the manuscript:

- All units (except inorganic nutrients) are now given in mg L-1.
- We add a figure of the bacterial growth curve fit to the supplement
- We add a methods description of the bacteria cell to carbon conversion
- We adjusted the remineralization rates which are now based on bacterial C instead of cells.

We also double checked all equations in the manuscript and modelling code for consistent units and repeated the fitting routing of the EXT model (Now using bacteria converted to bacterial carbon, and as mentioned above the corrected formulations of labile and refractory DON remineralization).